# Interannual Variations of Terrestrial Water Storage in the East African Rift Region

Eva Boergens[1], Andreas Güntner[1,2], Mike Sips[1], Christian Schwatke[3], and Henryk Dobslaw[1]

[1]GFZ German Research Centre for Geosciences, Telegrafenberg, 14473 Potsdam, Germany
[2]University of Potsdam, Institute of Environmental Sciences and Geography, 14469 Potsdam, Germany
[3]Technical University of Munich, School of Engineering & Design, Department of Aerospace & Geodesy, Deutsches Geodätisches Forschungsinstitut (DGFI-TUM), Arcisstraße 21, 80333 München, Germany

**Correspondence:** Eva Boergens (boergens@gfz-potsdam.de)

**Abstract.** The US-German GRACE (Gravity Recovery and Climate Experiment, 2002-2017) and GRACE-FO (GRACE-Follow-On, since 2018) satellite missions observe terrestrial water storage (TWS) variations. Over twenty years of data allow for investigating interannual variations beyond linear trends and seasonal signals. However, the origin of observed TWS changes cannot be determined solely with GRACE and GRACE-FO observations. This study focuses on the northern part of

the East African Rift around the lakes Turkana, Victoria, and Tanganyika. It aims to characterise and analyse the interannual TWS variations compared to meteorological and geodetic observations of the water storage compartments (surface water, soil moisture, and groundwater).

We apply the STL method (Seasonal Trend decomposition based on Loess) to decompose the signal into a seasonal signal, an interannual signal, and residuals. By clustering the interannual TWS dynamics for the African continent, we define the exact

outline of the study region.

We observe a TWS decrease until 2006, followed by a steady rise until 2016, and the most significant TWS gain of Africa in 2019 and 2020. Besides meteorological variability, surface water storage variations in the lakes explain large parts of the TWS decrease before 2006. Storage dynamics of Lake Victoria alone contribute up to 50% of these TWS changes. On the other hand, the significant TWS increase around 2020 can be attributed to nearly equal raise in groundwater and surface water

storage, which coincide with a substantial precipitation surplus. Soil moisture explains most of the seasonal variability but does not influence the interannual variations.

As Lake Victoria dominates the surface water storage variations in the region, we further investigate the lake and the downstream Nile River. The Nalubaale Dam regulates Lake Victoria's outflow. Water level observations from satellite altimetry reveal the impact of dam operation on downstream discharge and on TWS decrease in the drought years before 2006. On the

other hand, we do not find evidence for an impact of the Nalubaale Dam regulations on the the strong TWS increase after 2019.

# 1 Introduction

Satellite gravimetry, as realised with the Gravity Recovery And Climate Experiment (GRACE, 2002-2017) satellite mission and its successor GRACE-Follow-On (GRACE-FO, since 2018), is the only remote sensing technique available today that provides quantitative estimates of water storage changes of regional to global scales. These observations represent changes in all hydrological storages, including all surface water bodies, soil moisture, snow, ice, and groundwater. That makes the GRACE and GRACE-FO data an unique observation type for hydrology.

The GRACE and GRACE-FO (hereafter only GRACE) satellite missions measure changes in the distance between the two twin satellites, one following the other in a polar orbit at very low (500 km) altitudes (Tapley et al., 2004; Landerer et al., 2020). A global gravity field can be derived from collecting these inter-satellite range variations along the satellite orbits over a certain period, usually a month. The underlying mass deviations can be inferred from the spatial and temporal changes in the monthly gravity fields.

The applications of GRACE data in hydrology are manifold and include, for instance, assessing water balance closure at regional to global scales (Lehmann et al., 2022), groundwater storage changes (Frappart and Ramillien, 2018), water storage capacity and flood potential (Reager and Famiglietti, 2009), or drought effects (Gerdener et al., 2019). Tapley et al. (2019) gives a comprehensive summary of state-of-the-art applications of GRACE data for climate and hydrological research.

Quantifying large-scale terrestrial water storage (TWS) variations on the continents has been one of the primary fields of application of GRACE data. Several publications investigated TWS in Africa at regional to continental scales. For example, Frappart (2020) analysed the groundwater storage in the Sahara aquifer systems, while Ferreira et al. (2018) combined TWS and surface water storage in the Volta Basin in West Africa. Scanlon et al. (2022) investigated hydrological extremes in Africa by considering climatic teleconnections. Of all continents, only Africa had an overall positive linear TWS trend of about $2\,\mathrm{mm/year}$ (corresponding to $62\,\mathrm{Gt/year}$) over the last 21 years (Figure 1a). Thus, the region has been gaining water during the previous two decades, although the magnitude of the signal is smaller than for most other continents. The spatial trend patterns, though, are heterogeneous (Figure 1b). Two regions stand out with positive TWS trends for the GRACE period: the Niger River Basin and the East African Rift. This study focuses on the northern part of the East African Rift which exhibits the most distinct trend.

The East African Rift is characterised by large lakes, including Lake Victoria, the second largest freshwater lake in the world (by area), Lake Tanganyika, and Lake Turkana. The lakes of the region have been named in the Global 200 eco-regions for conservation by the World Wide Fund for Nature (WWF), emphasising their importance for hydrology and ecosystems (Olson and Dinerstein, 2002). The shores are one of the most populous regions in the world (Salvatore et al., 2005; Center For International Earth Science Information Network-CIESIN-Columbia University, 2018). Local societies rely intensely on their water for industrial and domestic use (Juma et al., 2014).

Earlier studies on TWS changes in the East African Rift include Becker et al. (2010), who used GRACE data together with altimetric water level observations and found TWS influenced by the Indian Ocean Dipole via the regional precipitation regime, and surface water dynamics via the lake retention effects. Anyah et al. (2018) confirmed the former result by investigating

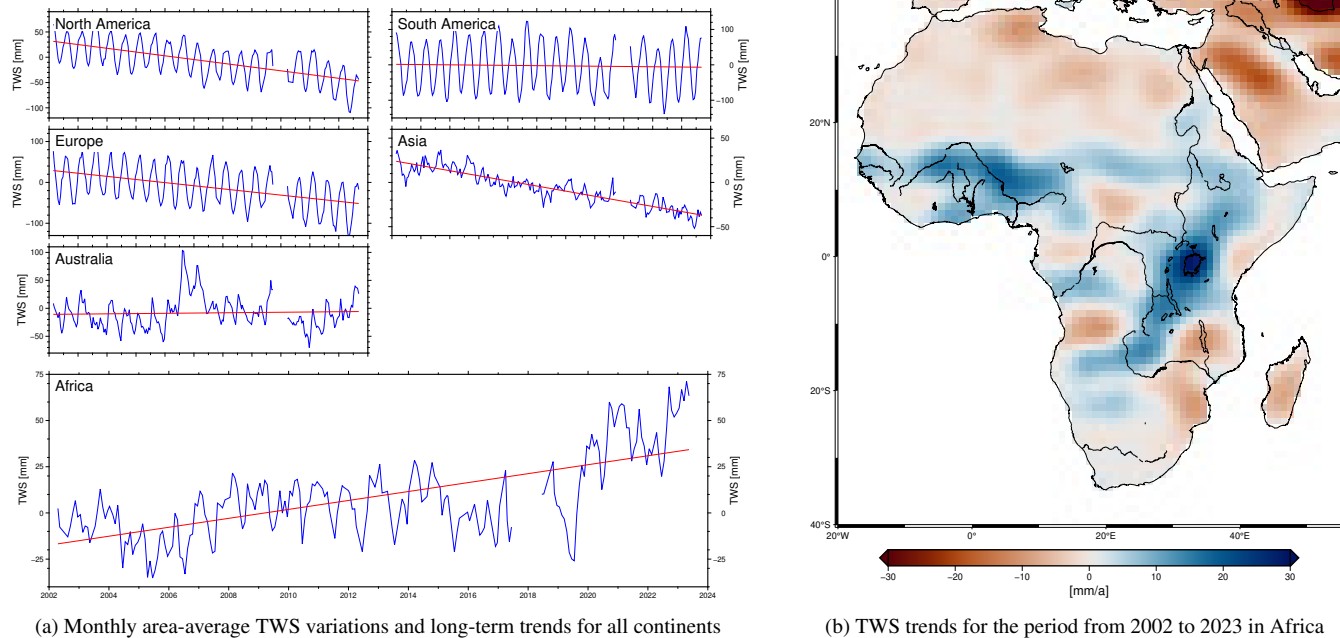

(a) Monthly area-average TWS variations and long-term trends for all continents

(b) TWS trends for the period from 2002 to 2023 in Africa

**Figure 1.** Linear trends of Terrestrial water storage (TWS) from GRACE satellite gravimetry (COST-G/GravIS data set, see subsection 3.1)

connections between climate indices and TWS and found a strong influence of the Indian Ocean Dipole, too. Kvas et al. (2023) analysed the water mass gain in Lake Victoria with a spatial high-resolution long-term TWS trend product against water mass estimations from satellite altimetry. Due to the spatial high resolution of the data, they could restrict the study area only to Lake Victoria and found a very high agreement between the GRACE and altimetry-observed water mass gain.

60    Monsoon precipitation governs the hydrology of the region (Palmer et al., 2023). To investigate long-term interannual variations in precipitation and evapotranspiration, especially in the context of drought monitoring, well-known indices such as the Standardised Precipitation Index (SPI) (McKee et al., 1993) and Standardised Precipitation-Evapotranspiration Index (SPEI) (Vicente Serrano, S.M., Beguiria, S. & Lopez-Moreno, 2010) have been used extensively. Both indices have been applied in regional studies in East Africa. For example, Ayugi et al. (2020) employed the SPEI to examine droughts and floods in Kenya,

65 or Uwimbabazi et al. (2022) used both SPI and SPEI for their investigation of changes in droughts over Rwanda.

Due to the large lakes, surface water storage is an essential contributor to TWS in the regions. Altimetry measures operationally the water levels of lakes, rivers, or wetlands (e. g. Schwatke et al., 2015) while surface water body extent can be monitored by optical remote sensing satellites (e. g. Pekel et al., 2016; Schwatke et al., 2019). With these two observations, storage variations can be estimated. Tong et al. (2016) combined altimetric water levels and surface area extent to investigate

70 surface water storage in East Africa, while Herrnegger et al. (2021) employed water level data of smaller lakes in Kenya to analyse the hydroclimatic conditions of the region. The large lakes of the East African rift have been the subject of earlier

research with multiple sensors and a particular focus on Lake Victoria (e. g. Swenson and Wahr, 2009; Velpuri et al., 2012; Hassan and Jin, 2014).

Besides surface water storage, TWS contains storage changes in soil moisture and groundwater. Soil moisture is now opera-
tionally monitored from space by the ESA CCI Soil Moisture product (Dorigo et al., 2017; Pasik et al., 2023). Liu et al. (2022)
used this data set to investigate droughts in East Africa. They identified the 2005/2006 drought, also visible in TWS data.

Unfortunately, in-situ groundwater data in East Africa are scarce (see data sets available at https://ggis.un-igrac.org/view/ggmn/). Thus, only satellite-based observations could cover this data gap. However, groundwater storage cannot be measured individually from space, but only as part of TWS measured with satellite gravimetry. To this end, groundwater storage es-
timations can be gained from TWS by subtracting all other water storage compartments observed by satellites or provided by hydrological models. Werth et al. (2017) estimated groundwater variations from TWS together with hydrological models in the Niger River Basin and found that groundwater increase plays an essential role for TWS there. Nanteza et al. (2016) investigated GRACE-based groundwater storage changes in East Africa with satellite observations and found high agreements with in-situ data. The Horizon2020 EU project Global Gravity-Based Groundwater Product (G3P) led to the development of a
global groundwater data set based on satellite data (Güntner et al., 2024).

The dense population of the study region influences the hydrology and, thus, TWS through human interventions such as the construction of large dams along the major rivers (Getirana et al., 2020). Most notably, Lake Victoria, which exhibited strong fluctuations in its water levels in the last decades, has been regulated since the 1950s by the Nalubaale Dam (formerly known as Owen Falls Dam). In the years between 2003 and 2006, the region exhibited a drought which naturally lowered the water levels
of Lake Victoria, while at the same time, the water extraction at the dam was increased, which further declined the water levels (Sutcliffe and Petersen, 2007; Kull, 2006; Awange et al., 2008). The disproportional water release was also made public by an independent hydrologic engineer, after which the dam operators returned to the previously agreed discharge curve. Intense rainfall events in 2019/2022 led to a rapid rise of water levels, causing massive floods along the shores (Khaki and Awange, 2021).

Vishwakarma et al. (2021) assessed global TWS trends and found the signal around Lake Victoria to be an "extreme gain". Rodell et al. (2018) globally investigated and categorised TWS trends as well and labelled the observation around Lake Victoria as "probable natural variability". In contrast, Zhong et al. (2023) found the TWS gain of the region to be non-precipitation-driven, indicating that it is presumably caused by anthropogenic actions. Several recent studies found with hydrological mod-
elling that a large part of the observed storage variation of Lake Victoria is due to human intervention and not naturally
occurring (Vanderkelen et al., 2018; Getirana et al., 2020). Whether and to which extend the observed TWS trends in the region are natural or anthropogenic is still under debate.

In this study, we investigate TWS signals in the northern part of the East African Rift. The variations show a distinct and significant interannual variability but no substantial changes in the seasonal component. Accordingly, we focus on interannual signals in this study. A clustering algorithm identifies the exact region outline in subsection 5.1. We also compare these in-
terannual TWS variations not only against meteorological data (subsection 5.2) but to observations of all other relevant water storage compartments (surface water, soil moisture, and groundwater) in subsection 5.3. That allows a more comprehensive

view of the different drivers of storage changes in the regions. This study closes with a more detailed investigation into the Lake Victoria and Nile River Basin and its storages as we strive to give additional evidence of whether the observed TWS trends are of climatic/natural or anthropogenic origin (subsection 5.4).

## 2 Study Region

We focus on the northern part of the East African Rift as outlined in Figure 2. It encompasses the high plateau of the African Rift system between the eastern (Gregory Rift) and western (Albertine Rift) branches of the rift. The Ethiopian highlands and Lake Malawi mark the northern and southern end of the region. We will abbreviate the study region as NEAR (Northern East African Rift).

The climate of NEAR is mainly tropical, with both an annual and semiannual precipitation signal in different parts (Palmer et al., 2023). The primary rainy season is from March to May (both annual and semiannual precipitation signal), and the secondary rainy season is from October to December (only seminannual) (Yang et al., 2015). Still, substantial interannual variability in precipitation and evapotranspiration have been observed in the past (Ummenhofer et al., 2018). However, the study region also includes arid regions in the North around Lake Turkana.

Some of the largest freshwater lakes of the world dominate the hydrology of the region. Namely, Lake Victoria is the second largest (by area, ninth by volume), and Lake Tanganyika is the sixth largest (by area, second by volume) freshwater lake. Lake Turkana, located at the northern end of the study region, is one of the largest endorheic lakes and the largest permanent desert lake in the world. All lakes that are accessible with satellite altimetry are included in this study. They account for 94% of the surface water bodies, by area, of the region according to the Global Lake and Wetland Database (GLWD, World Wildlife Fund).

Most importantly, Lake Victoria cannot be regarded as a natural lake any more. Uganda's Nalubaale Dam regulate the water level and outflow of lake since 1954. The reservoir on top of Lake Victoria was filled in the 1960s. This enlarged the lake volume by about $200 \, \mathrm{km^3}$ and raised the water level by about $2 \, \mathrm{m}$ (Okungu et al., 2005). It was agreed between the operators of the dam and the downstream riparians of the Nile River that the outflow should mimic a natural discharge curve (after the water level increase in the 1960s).

The agreed rating curve follows the equation:

$$Q = 66.6(WL - 7.96)^{2.01}, \tag{1}$$

where $Q \, [\mathrm{m^3/day}]$ is the water discharge and $WL[m]$ the water level at the gauge in Jinja (near the outflow) (Sene, 2000; Vanderkelen et al., 2018). However, the gauge datum is not publicly defined relative to meters above sea level. Thus, we cannot use the agreed rating curve with water level observations by satellite altimetry to estimate discharge directly.

In 2006, a second hydroelectric power plant named Kiira Power Station was inaugurated $1 \, \mathrm{km}$ downstream of the Nalubaale Dam.

Lake Victoria's outflow strongly governs the water levels of the downstream lakes in the Nile River Basin. Sutcliffe and Parks (1999, chapt. 4) showed with historic discharge observations that the outflow of Lake Victoria almost completely determines

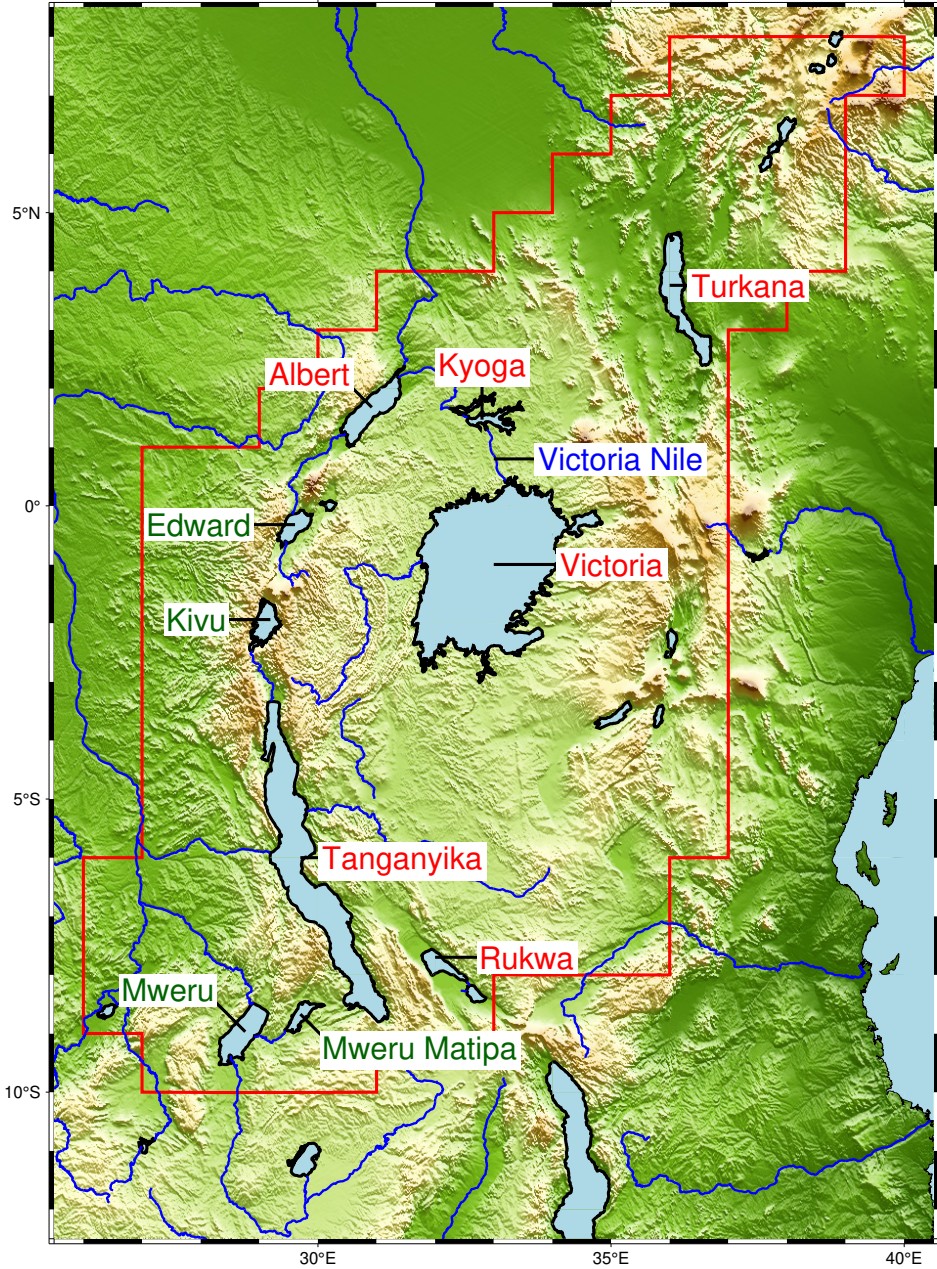

**Figure 2.** Northern East African Rift (NEAR) Region: Study area and major lakes. The red outline delineates the study area considered here. For lakes labelled in red, we analyse SWS variability individually; lakes labelled in green are summarised later as small lakes (see Figure 10).

140   the inflow and water level of Lake Kyoga. Lake Kyoga's outflow, in turn, almost completely determines the water levels of Lake Albert.

## 3 Data

### 3.1 Terrestrial Water Storage Data

We use 221 monthly gravity fields from the COST-G RL01 (GRACE) and RL02 (GRACE-FO) data set (Jäggi et al., 2020;
Meyer et al., 2023). They result from the IAG (International Association of Geodesy) service International Combination Service
for Time-variable Gravity Fields (COST-G) in which seven different Level-2 solutions of GRACE data processing centres are
combined.

These monthly gravity fields are filtered with the time-variable anisotropic VDK filter (Horvath et al., 2018) and subsequently
synthesised to a global $1°$ TWS grid. More processing steps are detailed in Dobslaw and Boergens (2023). The grids are freely
and publicly available from the GravIS portal (gravis.gfz-potsdam.de, TWS V.0005 (Boergens et al., 2020a)). The de-facto
spatial resolution of the TWS data is roughly $300\,\mathrm{km}$.

To assess the uncertainties, we employ the covariance model developed by Boergens et al. (2020b, 2022). With this model,
we compute the standard deviations of the regional mean TWS time series and uncertainties for each grid cell.

### 3.2 Precipitation Data and Precipitation Indices

This study analyses precipitation and a standardised meteorological drought index. We use the monthly Global Precipitation
Climatology Centre (GPCC) precipitation data set, given on a $1°$ grid until the end of 2019 (Schneider et al., 2022). After
January 2020, we use the First Guess GPCC monthly data, given in the same spatial resolution (Ziese et al., 2014). As the latter
contains data since 2004, we verify the consistency between the two in the overlapping period. Instead of monthly precipitation,
we consider time series of accumulated precipitation. To this end, the accumulated precipitation value for each month is the
sum over the preceding $n$ months, with $n$ taking integer values between 1 and 48.

Considering only precipitation omits another essential hydro-meteorological flux component in humid tropical climates:
evapotranspiration. For precipitation minus (potential) evapotranspiration ($P - ET$), we do not use direct observations but
the Standardised Precipitation-Evapotranspiration Index (SPEI) (Vicente Serrano, S.M., Beguiria, S. & Lopez-Moreno, 2010).
This index relates current $P - ET$ observations to the long-term mean since 1955. For the index, $P - ET$ at time step $t$,
accumulated over a fixed period of $n$ months $\overline{P - ET}_n$, is compared to the statistical distribution of this quantity in the same
month $j$ over the whole time series. From this distribution, which is not necessarily Gaussian, the mean value $\mu_j$ and the
standard deviation $\sigma_j$ are calculated. The index $SPEI(t,j)$ is then computed with

$$SPEI(t,j) = \frac{\overline{P - ET}_n(t,j) - \mu_j}{\sigma_j}. \tag{2}$$

SPEI values between -1 and 1 imply near normal conditions, while values below -1 indicate drier and above 1 wetter conditions
than usual.

The Instituto Pirenaico de Ecología, Zaragoza, Spain (https://spei.csic.es/database.html) provides two different pre-computed
SPEI data sets. The first, the SPEI Global Drought Monitor, is based on the GPCC First Guess precipitation data. The poten-
tial evapotranspiration is computed via the Thornthwaite equation for which the mean temperature is taken from the NOAA

NCEP CPC_GHCN CAMS dataset (Fan and van den Dool, 2008). This SPEI realisation is recommended for near-real-time applications. The second, the Global SPEI database (SPEIbase, v2.9), uses the CRU TS 4.03 precipitation data and the FAO-56 Penman-Monteith estimation for potential evapotranspiration (Vicente Serrano, S.M., Beguiria, S. & Lopez-Moreno, 2010; Beguería et al., 2010, 2014). This SPEI realisation offers more long-term, robust information. In this study, we employ both SPEI variants (called SPEI (GPCC-based) and SPEI (CRU-based) hereafter).

### 3.3 Surface Water Storage Data

In order to analyse surface water storage (SWS) variations, we employ altimetry data for water level (WL) time series together with water occurrence maps. Figure 2 shows the location of the lakes and the Victoria Nile River observed with altimetry in this study.

The WL time series are based on multi-mission satellite altimetry. They are freely available from the Database for Hydrological Time Series of Inland Waters web portal (DAHITI, https://dahiti.dgfi.tum.de). WL time series are based on a Kalman filtering approach and an extended outlier rejection, described in detail in Schwatke et al. (2015). All applied geophysical corrections and models are identical for all altimeter missions, including a multi-mission cross-over analysis to derive homogeneous WL time series from various satellites. The time series length varies and falls between September 1992 and September 2023, depending on the available data. The temporal resolution depends on the number of altimeter crossings and the repeat cycle of the used mission and can vary between a few days and about a month.

To assess the water surface area (WSA) of the lakes, we analysed the Global Surface Water Occurrence maps provided by Pekel et al. (2016) via the Global Surface Water Explorer (https://global-surface-water.appspot.com/). The data set is based on 36 years (1984-2020) of global remote sensing observations of water surfaces, classified to a water occurrence probability for each pixel. Permanent water bodies have a water occurrence probability of 100%, while lake shores have a probability between 0% and 100% due to varying water levels. However, as a result of cloud cover or similar effects in the remote sensing data, some permanent water pixels do not reach 100% (see Figure A1 in the Appendix for Lake Albert as an example). Further, the histogram of values inside Lake Victoria within a $20 \, km$ margin (see Figure A2 in the Appendix) showed, that all values above 95% should be considered as permanent water.

We can assess the summed pixel area for each water occurrence probability from these maps. As the lake outlines provided by GLDW do not perfectly coincide with the maps (see again Figure A1 in the Appendix), we collect the data inside the lake polygon plus a buffer of $20 \, km$. We then estimate an empirical cumulative distribution function (ECDF) of WSA.

Similar to the ECDF of WSA, we derive the ECDF of water levels of each lake. Assuming a monotone and continuous relationship between WL and WSA, the two ECDFs can map a WSA for each WL observation. Figure 3 illustrates the procedure on the example of Lake Tanganyika. For a given WL, its ECDF quantile is read (step 1), i.e., the percentage of observed WL equal or lower. The same quantile is looked up in the WSA ECDF (step 2); thus, the corresponding WSA can be determined (step 3). With this procedure, we get an associated WSA for every WL value.

Here, we assume a monotonic but non-parametric relationship between WL and WSA. Thus, the ECDF method is more flexible for complicated terrains than methods fitting a parametric curve through the WL – WSA relationship (e. g. Wang et al.,

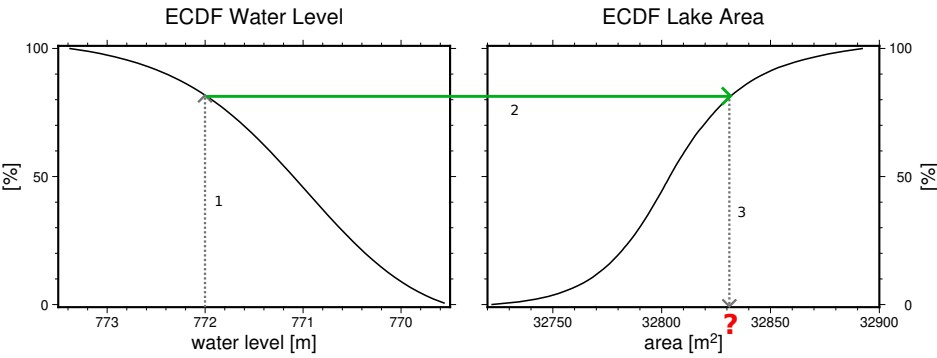

**Figure 3.** Example of WL and WSA empirical cumulative distribution function (ECDF) for Lake Tanganyika. Procedure to get surface area from a given WL: Step 1 - for WL, read the quantile of ECDF; Step 2 - look up the same quantile in WSA ECDF; Step 3 - get WSA to this quantile.

2011). However, the ECDF approach expects that the WL – WSA relationship does not show a hysteresis, i.e., the relationship does not depend on rising or falling water levels (Zhang and Werner, 2015). This assumption holds in this study as we do not
investigate lakes with extensive wetlands.

From the time series of WL and WSA, the water volume change $\Delta V_i$ between the time steps $t_{i-1}$ and $t_i$ can be calculated with a truncated pyramid formula (Abileah et al., 2011):

$$\Delta V_i = \frac{1}{3}\left(WL_i - WL_{i-1}\right)\left(WSA_i + WSA_{i-1} + \sqrt{WSA_i WSA_{i-1}}\right). \tag{3}$$

To get the time series of lake volume $V$ relative to the first time step, all $\Delta V_i$ are cumulated. Following, the lake volume (in
[m$^3$]) is converted to lake storage (in [Gt]).

The pyramid formula is based on linear lake profiles between the two WL observations. In most cases, the differences between consecutive height observations are as small as a few centimetres, where this simplified profile is reasonable. Nevertheless, the differences can be as large as one metre due to data gaps or rapid changes in the water levels observed with temporally sparse altimetry. Thus, we tested the assumption by artificially removing WL observations. We found only very
minor differences in the resulting volume time series with and without data gaps. Especially given the uncertainties of the WSA and WL data (see below), these are negligible.

In order to compare the lake storage variations to TWS, we linearly interpolate each time series to the GRACE time steps and distribute the mass uniformly over the lake surface yielding equivalent water heights. Next, we employ a spatial Gaussian filter with a half-width of $250\,\mathrm{km}$ to mimic the resolution of TWS. The half-width has been found by comparing the empirical
spatial correlation function of TWS and other water storage compartments smoothed with different Gaussian filters (Güntner et al., 2023).

No direct uncertainty estimates are available for WL and WSA and, thus, SWS. Although the DAHITI WL time series are provided with a field labelled "errors", these estimates only describe the internal error of the Kalman filter. They should only be used to compare different time series against each other. Thus, we rely on literature data, where altimetric WL time series

have been externally validated against in-situ gauge data. Schwatke et al. (2015) found RMSE values around $5\,cm$ for lakes which we take for WL uncertainty in our study.

We employ the value of 5% misclassification for the water occurrence maps to estimate the uncertainty of WSA (Pekel et al., 2016). The uncertainties of WSA and WL are variance propagated to the volume time series and SWS. We know our uncertainty assumptions are conservative, probably leading to overestimating uncertainties. However, the resulting uncertainties are in the

same order of magnitude as the TWS uncertainties.

### 3.4  Soil Moisture and w

We evaluate root-zone soil moisture storage (RZSM) variations based on the data product available in Güntner et al. (2024) until September 2023. The data set is based on the ESA CCI soil moisture product (Pasik et al., 2023) but spatially smoothed with a Gaussian filter, half-width $250\,km$. An uncertainty assessment in the form of gridded standard deviations accompanies

the data set.

While the RZSM of the Güntner et al. (2024) data are satellite-based, SWS is based on simulation results of the hydrological model LISFLOOD (van der Knijff et al., 2010). Unfortunately, these SWS have to be considered unreliable in the study region according to Prudhomme et al. (2024) as modelled runoff do not agree with in-situ observations. Thus, the groundwater storage (GWS) data employed in this study is estimated from TWS, RZSM and the altimetry-based SWS (cf. subsection 3.3). The

SWS estimations do not contain river mass variations. However, we assume that river water storage does not exhibit significant interannual variability, only seasonal. The uncertainty of GWS is variance propagated from the uncertainties of TWS, SWS, and RZSM.

## 4  Methods

### 4.1  Time Series Analysis

A time series decomposition into deterministic periodic (e. g. annual and semiannual) signals and a linear trend is not well suited to characterise the temporal variations of TWS in Africa over the available 21 years. It cannot describe the substantial interannual variability and possible change in the seasonal amplitude due to climate change. Thus, we employed the loss-free Seasonal-Trend Decomposition with Loess (STL) method to separate the TWS signals into an annual, trend, and residual signal (Cleveland et al., 1990). The so-called trend component of STL contains not only a linear trend but all interannual variations. In

order to avoid confusion, we will continue to call the STL trend "interannual signal". The second advantage of STL compared to a conventional parametric decomposition is its ability to capture changing seasonal amplitudes over the time series.

The results of the STL decomposition depend on several parameters that govern the smoothness of the interannual and the annual signal. Cleveland et al. (1990) provide guidelines for choosing them, which we used together with empirical testing and visual inspection. That results in the following parameter values. $n_p$: Length of annual signal, 12 in our case; $n_i$ and $n_o$: the

number of passes through the inner and outer loop, set to 1 and 10, respectively; $n_l$: the width of the low-pass filter, to be set to

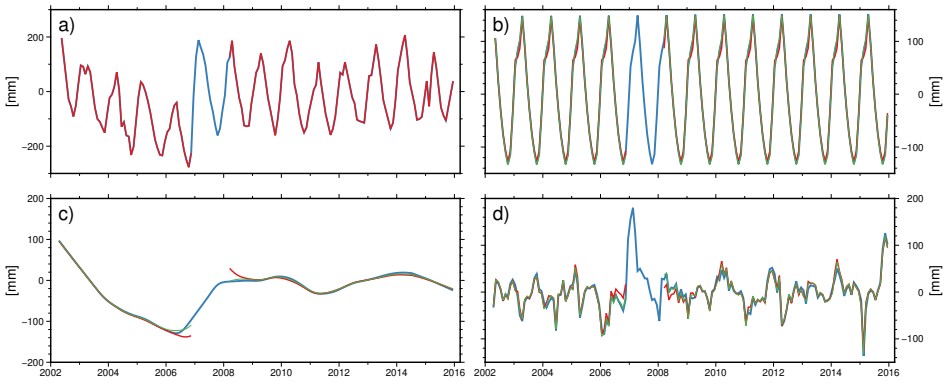

**Figure 4.** STL decomposition results of TWS time series (GRACE-only) at grid point 31.5°E, 5.5°S without data gap, with synthetically added data gap, and with filled data gap.

a) original TWS time series (blue) and with data gap in 2007 (red); b) annual seasonal signal; c) interannual signal; d) residual.

Blue lines: without data gap; red lines: standard STL gap filling; green lines: adjusted gap filling including seasonality.

the least odd integer larger than $n_p$, thus set to 13; and $n_t$ and $n_s$: the trend and seasonal signal smoothing parameter, both set to 35. While the former four parameters are straightforward, $n_s$ requires more considerations. We chose the value $n_s = 35$ in such a way that we consider the interannual variability of the seasonal signal no longer governed by noise. $n_t$ depends on the value of $n_s$. However, we found that the value provided by the rationale given in Cleveland et al. (1990) ($n_t = 19$) produced

a trend component containing still too many short-term variations. Finding the value $n_t = 35$ was done with empirical testing and visual inspection.

STL struggles to determine the interannual signal component around data gaps, such as the period after the end of GRACE and the launch of GRACE-FO in 2017/2018. The standard STL approach linearly interpolates across missing data, which is inappropriate here due to the seasonality. Figure 4 displays the problem based on a synthetic gap (Dec. 2006 to Feb. 2008). In

blue, the resulting separated signals of the original signal are displayed, and red shows the results in the presence of a data gap. While the annual signal is barely affected, the interannual signal exhibits unexpected behaviour before and after the missing months.

To overcome this problem, we implemented a more sophisticated gap-filling approach. We took the STL described above in the first step to identify the annual signal which was then removed from the original time series. Across the missing months

the resulting residual time series is linearly interpolated, and subsequently the annual signal is added back. This time series is then, in turn, used as input to the STL decomposition. Finally, the time steps of the data gaps were masked out again for further analysis and presentation. Returning to the example above, Figure 4 shows in green the resulting separated signals after we filled the synthetic gap in 2007. The interannual signal is significantly closer to the original interannual signal, and the unexpected peaks have vanished.

As the data sets are provided with uncertainties, these must be propagated to the STL decomposed time series. Instead of analytical uncertainty propagation through the iterative process, we employ a Monte Carlo simulation. To this end, we

added Gaussian-distributed noise to the input time series prior to the STL decomposition. By realising 100 differently noisy decompositions, we gained an estimate of the spread of the seasonal and interannual signals.

We applied the STL decomposition with gap filling and uncertainty estimation to the TWS, SWS, RZSM, and GWS data sets.

## 4.2 Clustering Algorithm

The visual inspection of the interannual TWS signals revealed similar temporal patterns in regions often incongruent with river basins or climate zones. Thus, we employed a cluster analysis to identify regions of similar temporal TWS dynamics.

Clustering aims at grouping data items into subsets such that the elements within each set have a high degree of similarity among themselves and are relatively distinct from elements assigned to other clusters (see, e.g. (Hastie et al., 2009) for an overview of cluster analysis algorithms). We applied a hierarchical approach (Ward, 1963) for which no assumptions are needed. Hierarchical approaches produce a tree of clusters, where subsets at higher levels are created by merging two clusters from the next lower level. Here, we measure the similarity of two grid points with the pairwise Euclidean distances of their time series. We also considered the connectivity graph, which represents the k-nearest neighbours, to avoid a disjunct distribution of the resulting regions across the continent.

## 4.3 Validation and Assessment Metrics

This study uses different validation and assessment metrics to compare different observations.

We employ two correlation coefficients, which evaluate temporal similarities of time series regardless of amplitude difference. The first one is the well-known Pearson's correlation coefficient $\rho$, which is defined as

$$\rho = \frac{1}{n} \frac{\sum_{i=1}^{n}(x_i - \overline{x})(y_i - \overline{y})}{std(x)std(y)} \tag{4}$$

with $x$ and $y$ the two time series with the length $n$ and their standard deviations $std(.)$.

The Pearson correlation coefficient measures a linear relationship between the two time series, whereas Spearman's rank correlation coefficient $\rho_s$ only assumes an (unknown) monotonic relationship. It is defined as

$$\rho_s = \frac{1}{n} \frac{\sum_{i=1}^{n} R(x_i)R(y_i)}{std(R(x))std(R(y))}, \tag{5}$$

where $R(x)$ is the rank variable of $x$. Thus, Spearman's rank correlation is the Pearson's correlation coefficient applied to the rank variables. Both correlation coefficients range between -1 and 1, with 1 indicating a perfect linear (or monotonic, respectively) relationship.

Further, we employ the percentage of explained variance ($PEV$) to evaluate the relationship of the amplitudes. $PEV$ is defined as

$$PEV = \left(1 - \frac{var(x-y)}{var(x)}\right)100\%. \tag{6}$$

Here, $var(.)$ denotes the variance of the time series.

## 5 Results and Discussion

### 5.1 Clustering of Interannual TWS Variations

We applied the clustering method described in subsection 4.2 to the interannual TWS signals of Africa. That resulted in eight regions of similar interannual TWS dynamics (see Figure 5 for the spatial distribution and the mean TWS time series for each cluster). Cluster 7 contains the whole of Madagascar Island. Cluster 2 encompasses most of the Sahara desert and lacks significant TWS signals. Cluster 6 covers most of the Niger River Basin and is one of the two African regions with a strong positive trend. Large parts of the tropical rain forest (climate A according to Köppen-Geiger classification) are in Cluster 3, which does not show a positive trend unlike the other central African regions (Clusters 0, 4, 6). The western part of southern subtropical Africa is subsumed into Cluster 5, which was wetting until 2012 and subsequently drying again. Clusters 1 and 4 encompass large parts of Africa and summarise regions with and without trends, respectively. Cluster 0, the NEAR region, shows a distinct temporal dynamic compared to the other African regions with substantial non-linear interannual variability, i.e., a TWS decline until 2006 and an overall increase afterwards, culminating in a steep TWS rise in 2019 and 2020 and again a subsequent decline.

The numbering of the clusters does not have any further meaning. However, by step-wise increasing the number of regions $m$ the dissimilarity of a found cluster to all others can be investigated. If we assume only two clusters ($m = 2$), the algorithm first separates Madagascar Island from mainland Africa due to its spatial disconnection. With $m = 3$, Cluster 0 is already separated from all other regions in mainland Africa, indicating that here the TWS signals are most distinct. The separation between the clusters can be measured by the Euclidean distance, on which the algorithm is based, between the mean time series. We found that Cluster 0 has the largest Euclidean distance to all other clusters.

The following two regions to be split off are Cluster 5 and Cluster 6, which both have distinct interannual variations, too. As $m$ further increases, the split-off clusters become less distinct and have a larger signal spread within.

We decided on the final value for $m$ based on the results, especially the size and shape of the regions. We sought the largest number of clusters while keeping them reasonable for GRACE data interpretation. Here, we found $m = 8$ to be the optimal number. The ninth cluster would be ring-shaped and only about $100\,\mathrm{km}$ across in the narrowest place. Thus, such a region is no longer meaningfully interpretable with GRACE data.

### 5.2 Comparison between TWS Signals and Precipitation

In this section, we compare the interannual TWS variations with the GPCC precipitation data set and the meteorological drought index SPEI. TWS and precipitation are not directly comparable but linked to each other via the water budget equation: $\frac{d(TWS)}{dt} = P - ET - R$, with $R$ being the runoff. We decided against differentiating TWS for the comparison but to temporally integrate $P$ and $P - ET$ to evaluate similarities. We call the temporally integrated precipitation "accumulated precipitation". The temporally integrated $P - ET$ is investigated with the SPEI. To this end, we employ the two SPEI variants introduced above.

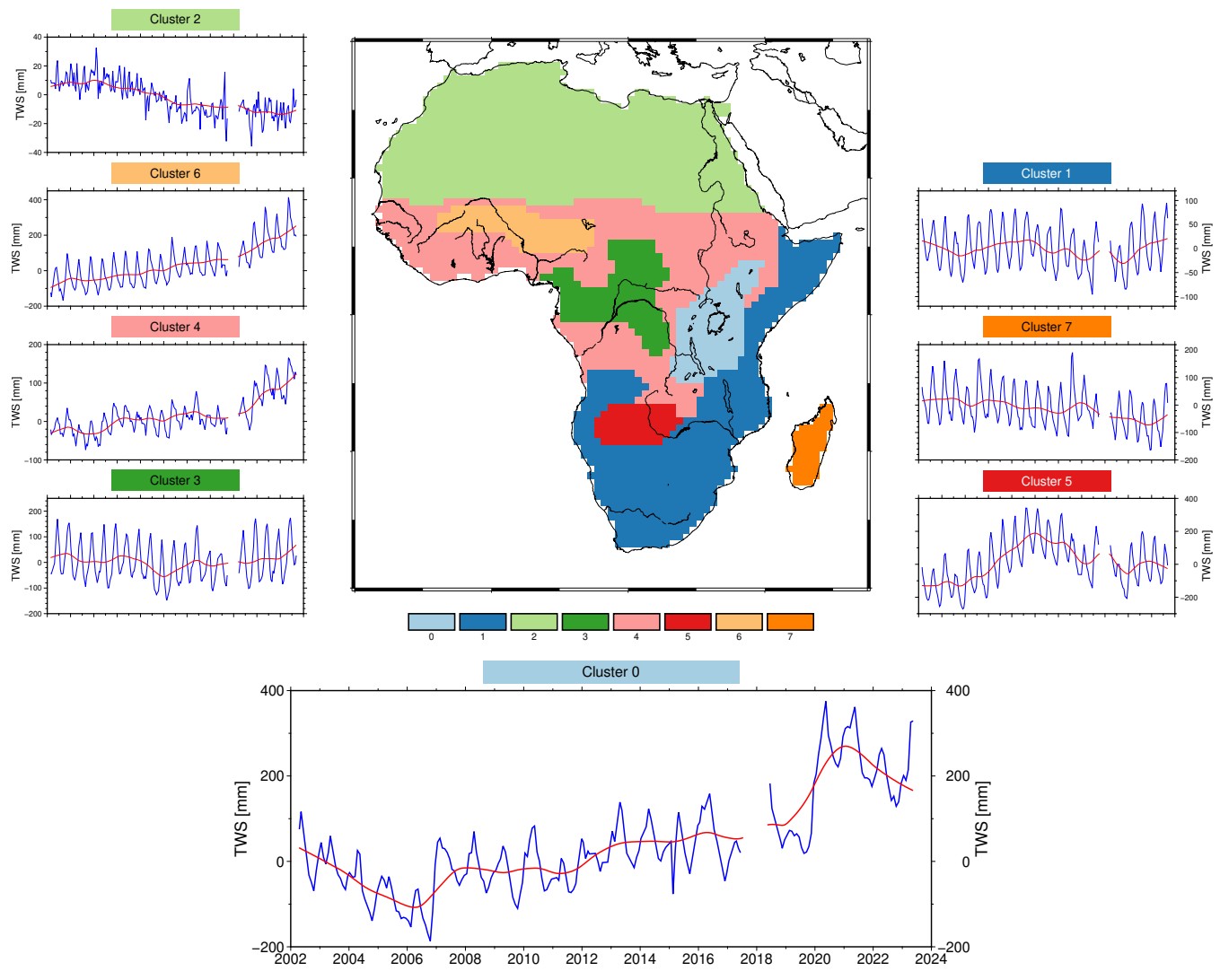

**Figure 5.** Cluster results for interannual TWS variability over Africa (centre). The mean TWS time series (blue) and the interannual trend component (red) from the STL analysis are shown for all clusters. Cluster 0 is the region of interest in the East African Rift.

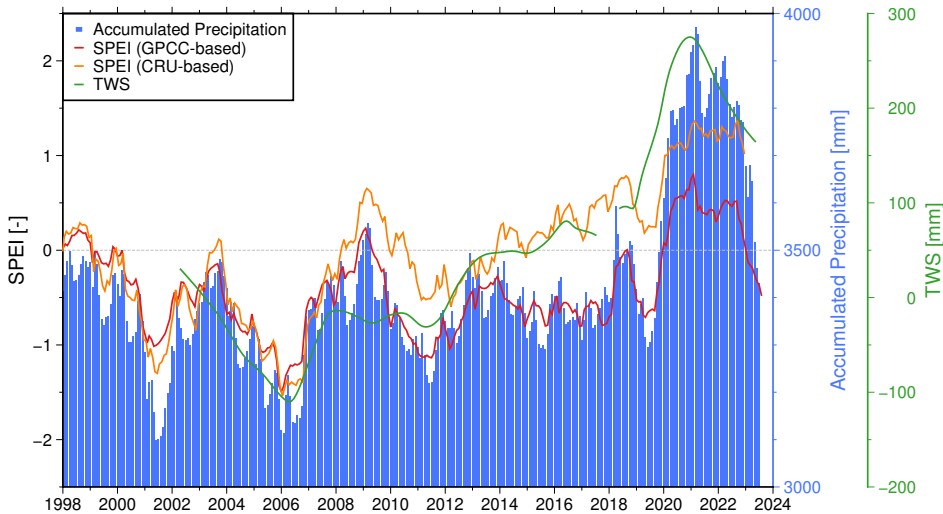

**Figure 6.** Comparison between SPEI (GPCC-based), SPEI (CRU-based), accumulated precipitation (accumulation period of 36 months), and TWS. Please note the different y-axes.

As we only investigate the interannual variability of TWS, the accumulation period for both precipitation and SPEI should be an integer multiple of twelve months to remove the seasonality. Following the rationale of the STL parameter choices (see subsection 4.1), we chose an accumulation period of 36 months.

Figure 6 presents the interannual time series of TWS, the accumulated precipitation, and the two variants of SPEI. The relative shortage of precipitation from 2003 until 2006 matches the decline of TWS in these years. In the early years before 2008, TWS fit well with both SPEIs. However, after 2008, TWS continuously rose while the accumulated precipitation stayed nearly constant until its rapid increase following 2020.

SPEI (CRU-based) shows a positive trend from 2008 to 2016 but does not display a step around 2020. On the other hand, SPEI (GPCC-based) does not show the intermediate increase but only the distinct in 2020.

Overall, both SPEI indices strongly correspond with TWS with a correlation coefficient of $\rho = 0.88$ (CRU-based) and $\rho = 0.79$ (GPCC-based). TWS and precipitation correlates with $\rho = 0.87$. However, the values of $\rho$ are governed by the two extreme points of the time series in 2006 and 2020. That explains that despite SPEI (CRU-based) being better able to describe the changes between the two extremes, precipitation and SPEI(GPCC-based) have a similarly high correlation to TWS.

To further investigate the differences between the two variants of SPEI, we looked into their input precipitation. The comparison of these two data sets revealed significant differences in the overall volumes of accumulated precipitation but similar interannual dynamics (see Figure B1 in the Appendix). Increasing rainfall trends since 2008 are slightly larger for CRU (4.3 mm/year) than for GPCC (4.0 mm/year) which may partly explain the diverging patterns of the two SPEI data sets after 2008. Nevertheless, differences in the (potential) ET data used may also contribute to the differences, but the (potential) ET data itself were not available to us.

Before 2006, the shortage of accumulated precipitation can at least partly describe the TWS drought in these years. Similarly, the excess thereof can account for the TWS increase in 2020-2022. However, meteorological data alone cannot explain the TWS gain between 2008 and 2016. Thus, precipitation or precipitation minus evapotranspiration is an essential driver of TWS but insufficient to justify all variations observed. Further, the discussed differences between the two SPEI data sets and their input precipitation data sets limit the explanatory power of the comparison between SPEI and TWS. Hence, we will investigate the different storage compartments of TWS in the next section to identify further drivers.

### 5.3   Comparison between TWS Signals and Water Storage Compartments

We analyse the contributions of the water storage compartments (WSC) soil moisture, surface water storage and groundwater storage to the internannual TWS variations in the following. Water storage in snow and ice can be neglected for the present study.

Figure 7 a) shows the regional mean time series and their uncertainties of the WSCs and TWS, the interannual signals are displayed in Figure 7 b), and the seasonal signals in Figure 7 c). RZSM has the largest uncertainties that propagate into GWS. The annual variability observed by TWS originates mainly from RZSM, while the interannual variability originates both from SWS and GWS.

To look into the temporal agreement of the WSCs and TWS dynamics, we show the percentage of explained variance ($PEV$) and Pearson's correlation coefficient ($\rho$) in Figure 8. For these analyses, we used the full-time series. $PEV$ helps to evaluate to which extent the amplitudes of one WSC can explain the variations of TWS. On the other hand, $\rho$ describes the temporal similarities between the respective time series and is insensitive to amplitude differences. In the centre of the region around Lake Victoria, about 60% of TWS can be explained by SWS (Figure 8 a)). We observe high temporal correlations between TWS and SWS throughout the study region, with the highest values close to one again around Lake Victoria (Figure 8 d)). In the centre part of the region, only around 20% of the variations of TWS can be explained by RZSM and the temporal correlations are only around 0.5 (Figure 8 b) and e)). GWS is the dominant compartment according to $PEV$ in the northern part of the region and east and west of Lake Victoria (Figure 8c)). There, GWS also shows a high temporal similarity to TWS (Figure 8 f)).

We found in Figure 7 two periods of significant change in TWS, before 2008 and after 2018. Thus, we look closer at the $PEV$ for these shorter time spans. The results are in the Appendix in Figure C1. Here, the influence of RZSM significantly increases due to the higher influence of the annual variability. In 2002-2008, GWS has only minor contributions to the interannual variability of TWS according to $PEV$ while the SWS influence is even more focused around Lake Victoria. The TWS changes after 2018 are more evenly distributed across the three different WSCs.

Next, we investigate the yearly storage change contributions of the different WSCs compared to TWS. To this end, we employ the annual storage change of each WSC and TWS computed from the the interannual signal. The results are shown in Figure 9. Please be aware that the storage change of 2002 only comprised the months of April to December, 2017, January to July, and 2018, May to December. Due to using the interannual signal, the WSCs do not always sum up to TWS.

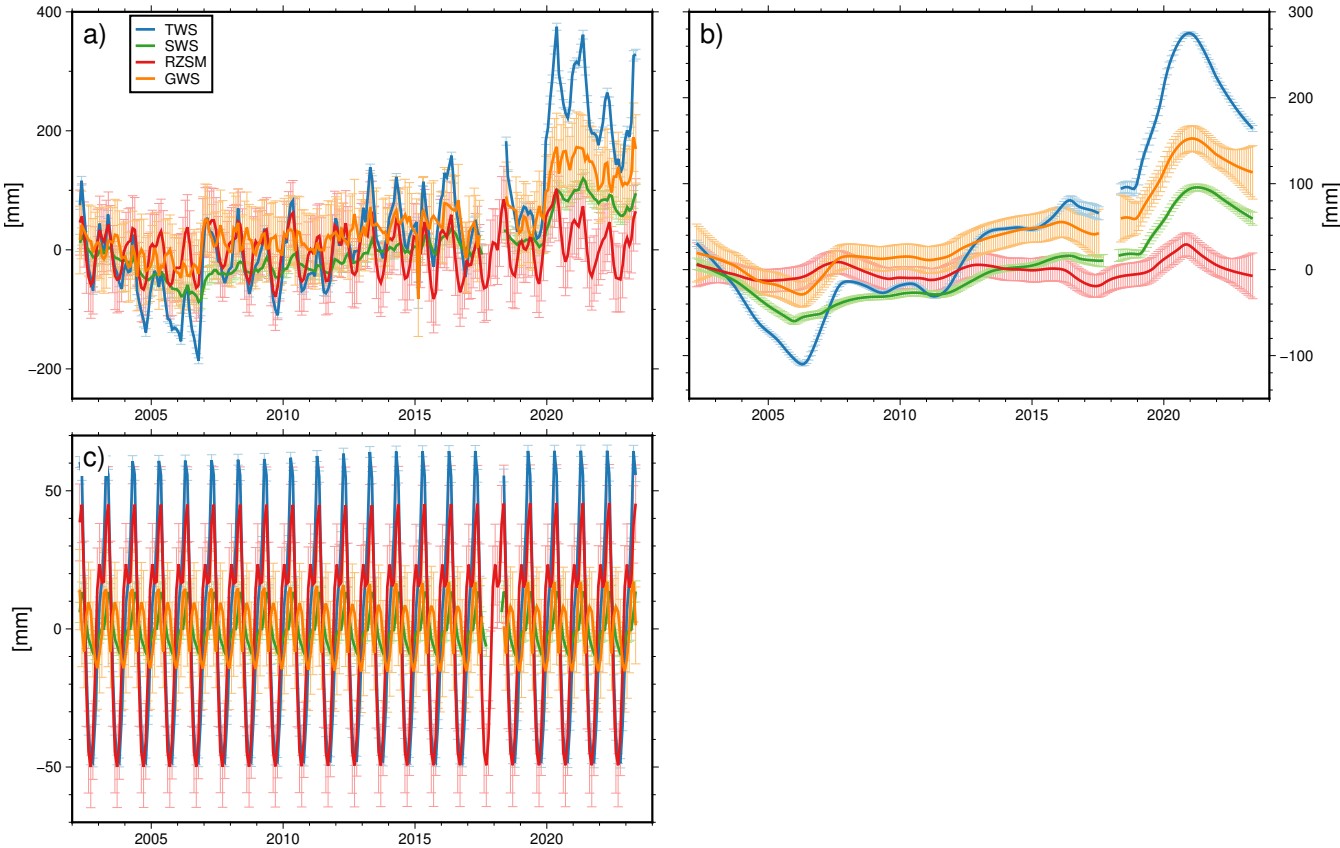

**Figure 7.** Time series of the mean water storage compartments with their uncertainties. a) Full signal; b) interannual signal; c) seasonal signal.

Again, we observe that RZSM contributes comparably little to the interannual variability. Especially in years with small storage changes (2008, 2009, 2010, 2011, 2013, 2014, 2017, 2018), the uncertainty of the RZSM change is larger than the overall TWS signal. During the early years 2002-2005, the loss of storage is governed by the negative storage change in SWS. On the other hand, the TWS deficit in 2021 and 2022 is instead governed by GWS depletion. The significant TWS increases in 2019 and 2020 originate equally in SWS and GWS. Similarly, in 2012, SWS and GWS observe nearly equal storage increases. In 2006 and 2007, GWS contributed more strongly to the TWS changes than SWS. In the median, both SWS and GWS can explain  35% of the annual change of TWS.

The change in TWS in 2016 is close to zero, while the WSCs observe significant changes. Looking into the time series in Figure 7, this might be caused by persisting problems of the STL gap filling.

To investigate the SWS contributions of the different lakes further, we examine the annual changes per lake compared to TWS, as above. Here, we do not employ the spatially filtered SWS data set but the individual lake mass changes. We only consider the mass change of a lake if less than two months are missing at the beginning or end of the year. In order to account

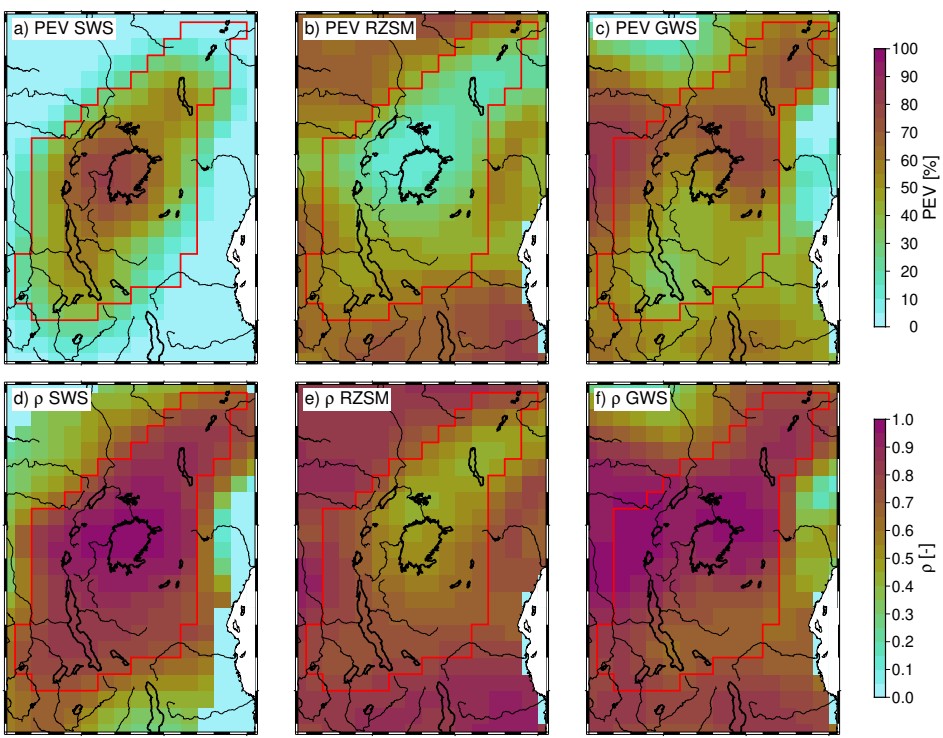

**Figure 8.** Upper row (a)-c)): Percentage of explained variance $PEV$ between spatially filtered WSC and TWS; lower row (d)-f)): Pearson's correlation coefficient $\rho$ between spatially filtered WSC and TWS.

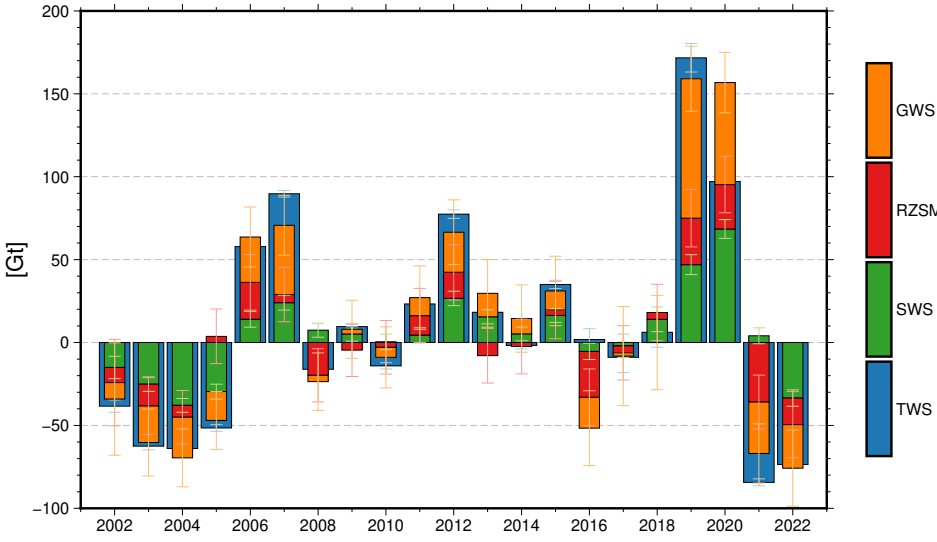

**Figure 9.** Annual changes in the storages of TWS and the WSCs.

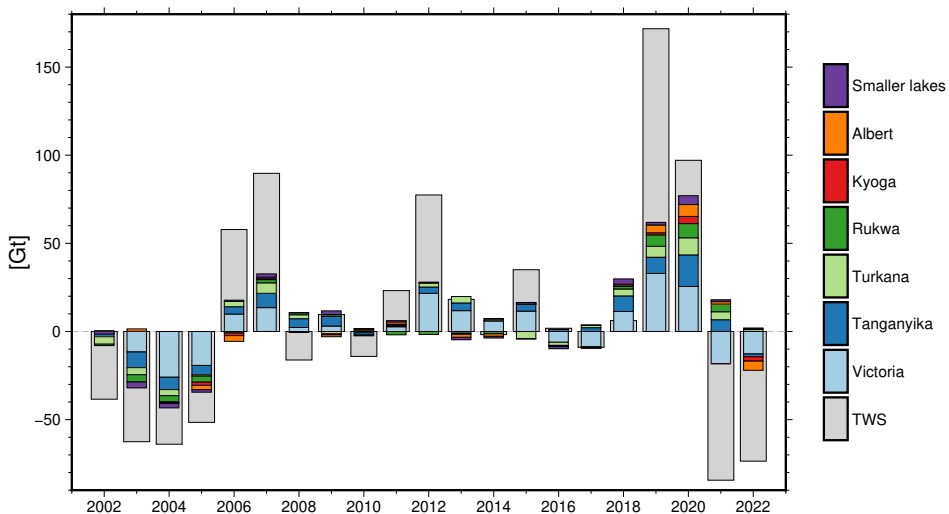

**Figure 10.** Annual TWS and lake water storage change.

for these months, the mass change is upscaled in these years. That is the case for Lake Albert in 2013, Lake Kivu in 2010 and 2013, Lake Mweru Matipa in 2010 and 2016, and Lake Edward in 2010 and 2013.

Figure 10 compares the annual TWS and lake mass changes; for the sake of readability, we refrained from showing uncertainties here. The lakes Mweru, Mweru Matipa, Kivu, and Edward (see Figure 2 green labelled lakes) are summarises as small lakes as their signals are too small to be distinguishable from each other in the figure. In years with a substantial TWS change, the lake masses usually agree with the direction of change. Especially, Lake Victoria exhibits the same direction as TWS, except in 2008 and 2014, which are years with minor signal. The median contribution of Lake Victoria to SWS is 63% but no clear pattern emerged here concerning wetting or drying years. It is notable that prior to 2006, the influence of Lake Victoria on SWS is considerably higher compared to Lake Turkana and Lake Tanganyika than in the years after 2019, considering their different sizes.

These results show that during periods covered by the investigated time series, different WSCs have a governing influence on TWS changes. In the drought years before 2006, SWS has the most significant influence, which, in turn, is governed by the changes of water storage in Lake Victoria. We found that SWS determines TWS the central region around Lake Victoria. That is also in line with earlier studies focusing on Lake Victoria alone, which found that SWS of Lake Victoria can clearly explain the majority of TWS (e. g. Kvas et al., 2023; Getirana et al., 2020). When looking into the two periods of large change in TWS (2002-2008 and 2018-2023), we see significant differences in the contribution of the individual WSCs.. During the drought and subsequent recharge before 2008, SWS around Lake Victoria contributed most strongly, while during the floods after in 2019 and 2020, all WSCs account similarly for TWS. We assume that the anthropogenic influence of Lake Victoria through the Nalubaale Dam can explain the pre-2008 behaviour. Thus, in the next section, we will further investigate Lake Victoria and the lakes of the Nile River Basin.

## 5.4 Dynamics of Lake Victoria and of Downstream Water Bodies in the Nile River Basin

Here, we investigate the relationship between the dynamics of Lake Victoria and the Victoria Nile River, Lake Kyoga and Lake Albert, located downstream in the Nile River Basin.

While we do not have access to the Victoria Nile River discharge data downstream of the Nalubaale Dam, we use altimetric water level observations as a proxy (Figure 11). Compared to the WL of the lakes, the quality of the time series of the Victoria Nile River is poorer due to the comparatively small size and the challenging topography of the river for satellite altimetry. Literature values for the uncertainty of altimetric WL for lakes are widely available, but the uncertainty values for rivers show a significant larger spread. Thus, we do not provide an uncertainty estimate for the WL of the Victoria Nile River. As the outflow of Lake Victoria strongly governs the WL of Lake Kyoga, we use them additionally to direct observations at the Victoria Nile River. We include Lake Albert to illustrate the natural flow between Lake Kyoga and Lake Albert.

The WL dynamics of Lake Victoria versus Victoria Nile River and Lake Kyoga are evident between 2006 and 2020 (Figure 11). While the former experienced a relatively steady rise in WL during the period, the WL of the latter remained stable. In 2020, the WL of all lakes quickly rose. The WL of Lakes Kyoga and Albert show high similarities, not only between each other but also to the temporal pattern of the Victoria Nile River.

The change in the relation between the WL over time is further illustrated in Figure 12 where the WL of Lake Victoria are plotted against the WL of the Victoria Nile River and Lake Kyoga. For this purpose, we interpolate the time series to common time steps, usually the coarser temporal resolution. Under natural conditions, the WL time series of the three lakes should have a close, albeit not necessarily linear, relationship. We thus employ Spearman's rank correlation coefficient $\rho_s$ to quantify the similarity. Only a weak relationship between the WL dynamics was found, with $\rho_s$ in the range of 0.37 to 0.41 (Figure 12). The poor data quality of the Victoria Nile River time series can partly explain the weaker Spearman's correlation between Lake Victoria and Victoria Nile River compared to the Lake Kyoga correlation. In contrast, with $\rho_s$ equal to 0.88, the WL dynamics of Lake Kyoga and Lake Albert are much more similar.

If the outflow of Lake Victoria were mainly governed by its WL, hardly any temporal variation in the relationship between the WL of Lake Victoria on the one hand and Lake Kyoga and Victoria Nile River on the other hand would be visible. However, according to Figure 12, the relationships changed: before 2006, more water was released at the dam as expected from the WL, as shown in previous studies. The dam operators discharged a surplus of water to help inaugurate the Kiira Power Station (Sutcliffe and Petersen, 2007; Kull, 2006; Awange et al., 2008). Between 2006 and 2019, the WL of Lake Victoria was rising while both the WL of Lake Kyoga and the Victoria Nile River stayed rather constant. That might indicates that less water was released from the dam than was supposed to by the agreed rating curve. The high amounts of precipitation can explain the sharp increase in all lakes in 2020.

All these observations agree with the modelled results of Vanderkelen et al. (2018) and Getirana et al. (2020), who found that the storage variations of Lake Victoria are influenced both naturally and anthropogenically.

Strictly speaking, we should consider the travel time of water between the lakes in the correlation analysis above. The Victoria Nile River between Lake Victoria and Lake Kyoga is about 120 km long (estimated on a map along the river path)

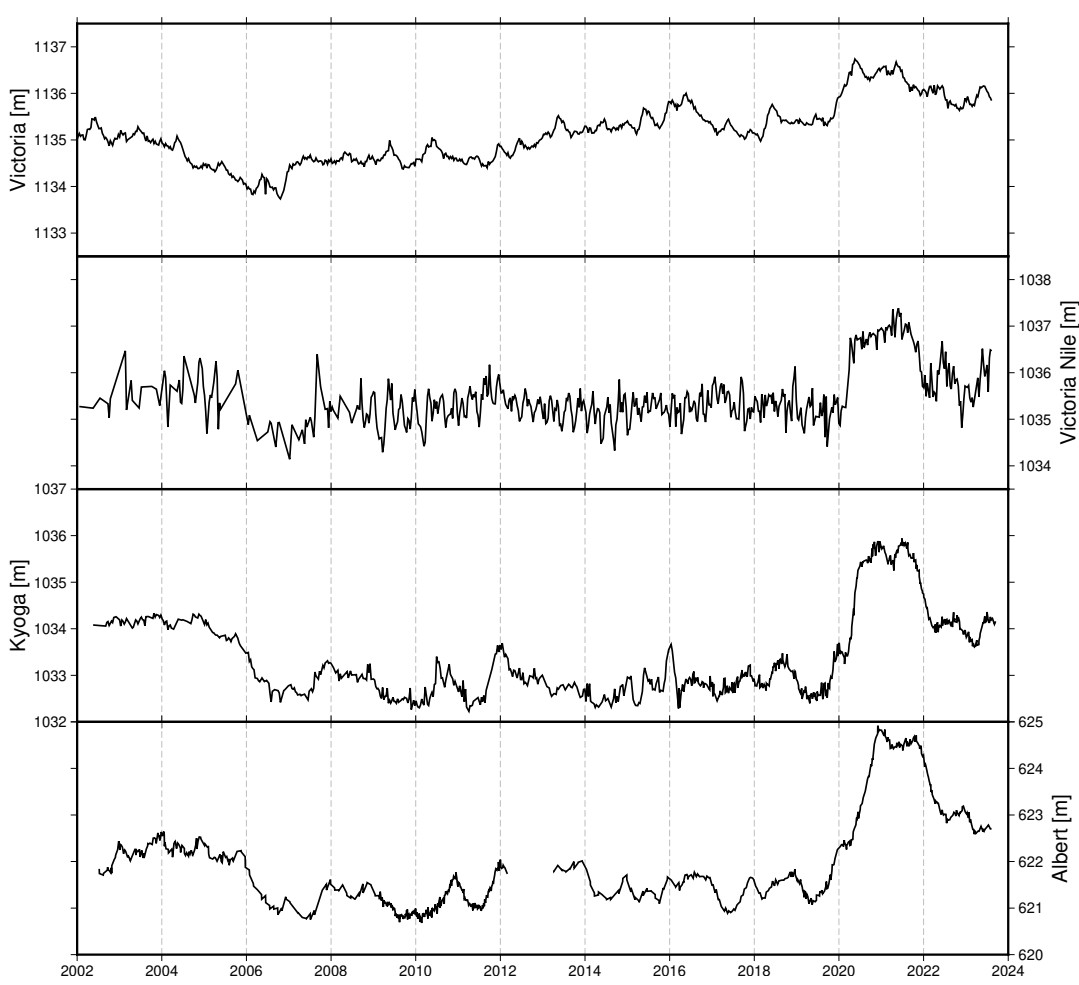

**Figure 11.** Water levels a.s.l. of Lakes Victoria, Victoria Nile River, Lake Kyoga, and Lake Albert.

and has an elevation difference of $100\,\mathrm{m}$ (measured by altimetry). We roughly estimate the flow velocity with the Gaukler-Manning-Strickler equation (Strickler, 1981) to $4.5\,\mathrm{m/sec}$ (assumption of constant river depth of $10\,\mathrm{m}$, river width of $500\,\mathrm{m}$, and literature value for large rivers for the Strickler coefficient of $35\,\frac{m^{\frac{1}{3}}}{s}$). Therefore, the travel time is only about $7.5\,\mathrm{h}$, which
465 is too fast for altimetry to observe a time shift.

Instead of employing WL to assess the lake discharge, we could also investigate volume changes as a proxy for the flow estimation. Investigations into volumes instead of water levels revealed no new information, so we refrained from presenting them here. Further, with volume change, we could only investigate the lakes, not the Victoria Nile River.

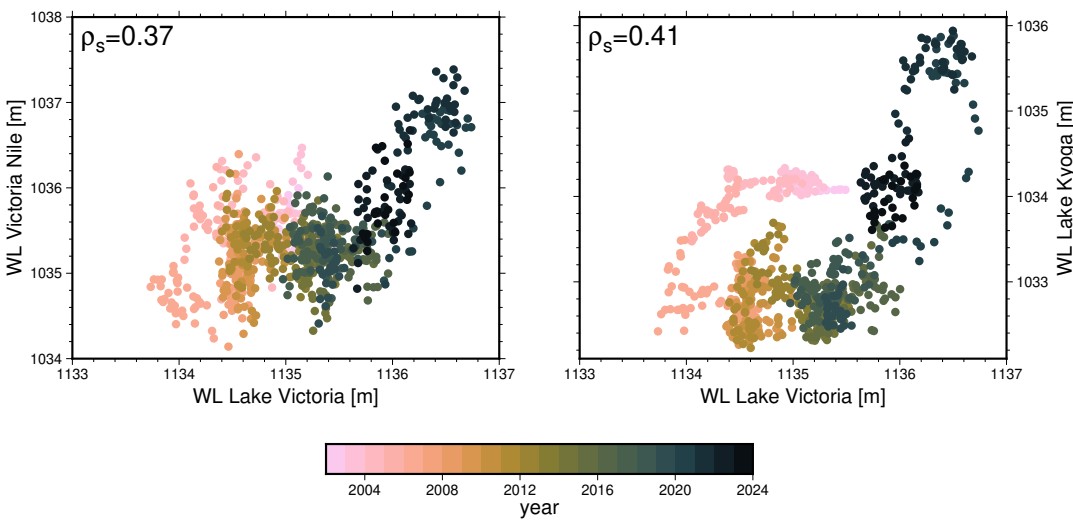

**Figure 12.** Relationships between the water levels a.s.l. of Lake Victoria and Victoria Nile River (left plot) and Lake Victoria and Lake Kyoga (right plot).

## 6 Conclusions

Unlike other world regions with clearly positive TWS trends over the last 20 years (e. g. the Caspian Sea region), the Northern East African Rift (NEAR), as well as most of Africa, shows a complex interannual behaviour.

A linear trend plus annual and semiannual seasonal signals describe the temporal patterns only insufficiently. To better investigate the interannual variations of TWS in Africa, the TWS signal was separated into an annual component and an interannual component with the help of the Seasonal-Trend Decomposition with Loess (STL) method. To further analyse the spatial patterns, a geographical clustering algorithm was applied to the interannual TWS signals to identify similar regions. The method is based on hierarchic trees but with the extension of ensuring geographically connected regions. With this method, the NEAR region was identified. It encompasses the East African highlands, from Lake Turkana in the North to Lake Tanganyika in the South, including Lake Victoria. The mean TWS signal of the study region shows a decline in water storage prior to 2006, linked to a documented natural drought period. Afterwards, TWS steadily increased until 2019. An even more substantial TWS rise occurred in 2019 and 2020 due to excess precipitation.

The first investigation focused on the comparison between the interannual TWS signal and GPCC precipitation data and the drought index SPEI, provided in two variants based on GPCC and CRU precipitation data, respectively. All three meteorological data sets detect a meteorological drought prior to 2006. However, only SPEI (CRU-based) could explain the steady increase in TWS between 2008 and 2018, while the precipitation data set and SPEI(GPCC-based) were better able to explain the substantial TWS gain in 2020. Nevertheless, the two precipitation data sets based on interpolated in-situ observations could not sufficiently well explain the observed TWS changes.

In a second step, the TWS compartments of surface water storage (SWS), groundwater storage (GWS) and soil moisture (root zone soil moisture - RZSM) have been analysed. RZSM is the driving storage of the seasonal TWS variability but add only slightly to the interannual variations. During the meteorological drought years prior to 2006, SWS had the most significant influence on TWS. However, in the exceptionally precipitation-rich years after 2019, SWS and GWS contributed similarly to TWS. Between 2008 and 2016, no clear driver for the steady TWS increase could be identified.

Further research into the impact of the lakes making up SWS revealed differences between the periods before 2006 and after 2019. Prior to 2006, SWS was strongly influenced by the mass variations of Lake Victoria, with only smaller contributions from Lake Tanganyika and Lake Turkana (the second and third largest lakes in the region). On the other hand, in 2019 and 2020, the storage changes of these three lakes were more balanced, considering their different sizes.

Finally, Lake Victoria, which is regulated by the Nalubaale Dam, and the downstream Nile River Basin with Lake Kyoga and Lake Albert were further studied. The water levels of these lakes are controlled by the dam's outflow. Satellite altimetry provided evidence that prior to 2006, the discharge was significantly higher than the agreed rating curve. Combining these results with the previous findings from meteorological, SWS, and GWS data sets, it can be concluded that the natural drought before 2006 was exacerbated in TWS by human decisions at the Nalubaale Dam. However, no clear evidence could be found that the natural precipitation surplus after 2019, leading to a storage surplus, was amplified by human activities.

Returning to the ongoing scientific debate on whether the TWS trend in the East African Rift is anthropogenic or natural, we conclude that it is a combination of both. Our research provides evidence that the interannual TWS variations of the African Rift region are influenced by a blend of natural precipitation and evapotranspiration variability, along with human interventions.

*Code and data availability.* The TWS data of COST-G used in this study have been published by Boergens et al. (2020a) and are available at ftp://isdcftp.gfz-potsdam.de/grace/GravIS/COST-G/Level-3/TWS. The data are published under the "CC BY 4.0" licence.

The water occurence map is available at https://global-surface-water.appspot.com and documented by Pekel et al. (2016). It is produced under the Copernicus Programme and is provided free of charge, without restriction of use.

The surface water storage and ground water storage data sets, including water levels, water surface extent, volume change, and filtered maps have been published by Boergens and Schwatke (2024) and are licensed under "CC BY 4.0".

The root zone soil moisture data are part of the data published by Güntner et al. (2024) and are available at https://datapub.gfz-potsdam.de/download/10.5880.G3P.2024.001-CWAniu/. The data are published under the "CC BY 4.0" licence.

Both SPEI data sets have been documented by Vicente Serrano, S.M., Beguiria, S. & Lopez-Moreno (2010); Beguería et al. (2010, 2014). SPEIbase can be downloaded at https://spei.csic.es/spei_database/; SPEI Global drought monitor can be downloaded at https://spei.csic.es/map/maps.html. Both are published under the ODbL 1.0 license.

The Python Code used for the clustering are published under the EUPL-1.2 licence and available at https://git.gfz-potsdam.de/big_data_analytics/hc-viz.

## Appendix A: Water Occurance Maps

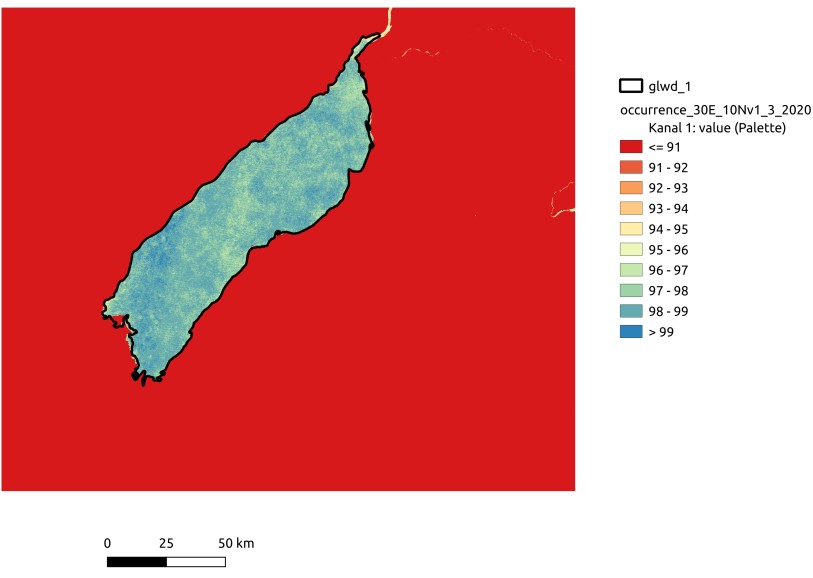

**Figure A1.** Water occurrence map of Lake Albert as an example of the below 100% occurrence probability in the middle of the permanent lake. Further the mismatch between lake outline from GLWD and the water occurrence maps are visible.

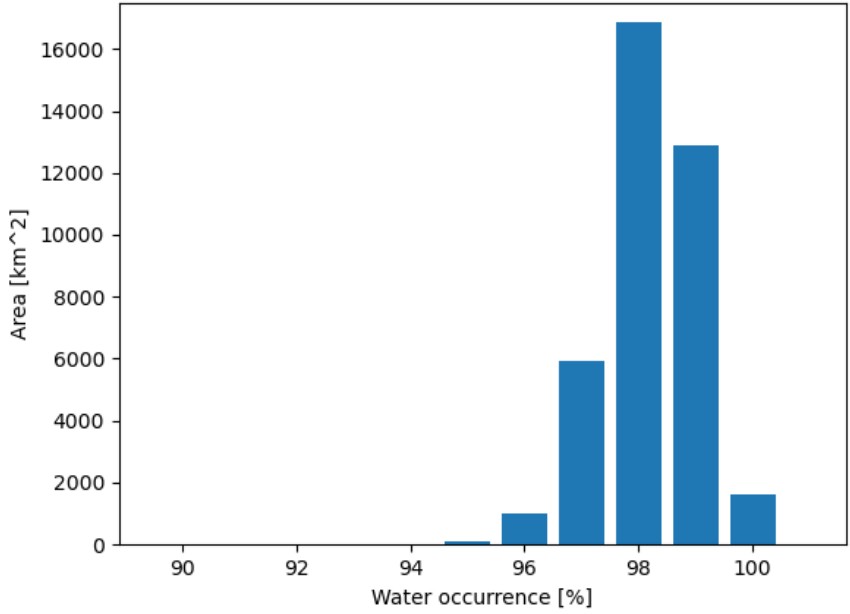

**Figure A2.** Histogram of the water occurrences inside a 20 km margin of Lake Victoria.

## Appendix B: Comparison of GPCC and CRU Precipitation Data Sets

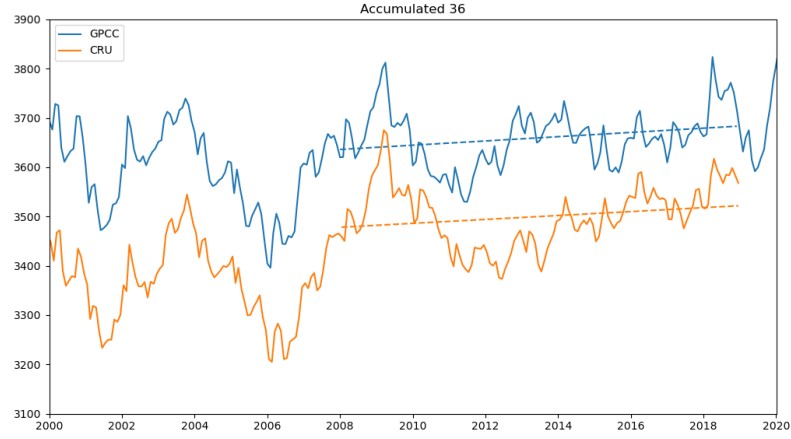

**Figure B1.** Comparison between accumulated precipitation (36 months) based on the GPCC and CRU data sets.

 **Appendix C: Percentage of Explained Variance for Time Periods 2002-2008 and 2018-2023**

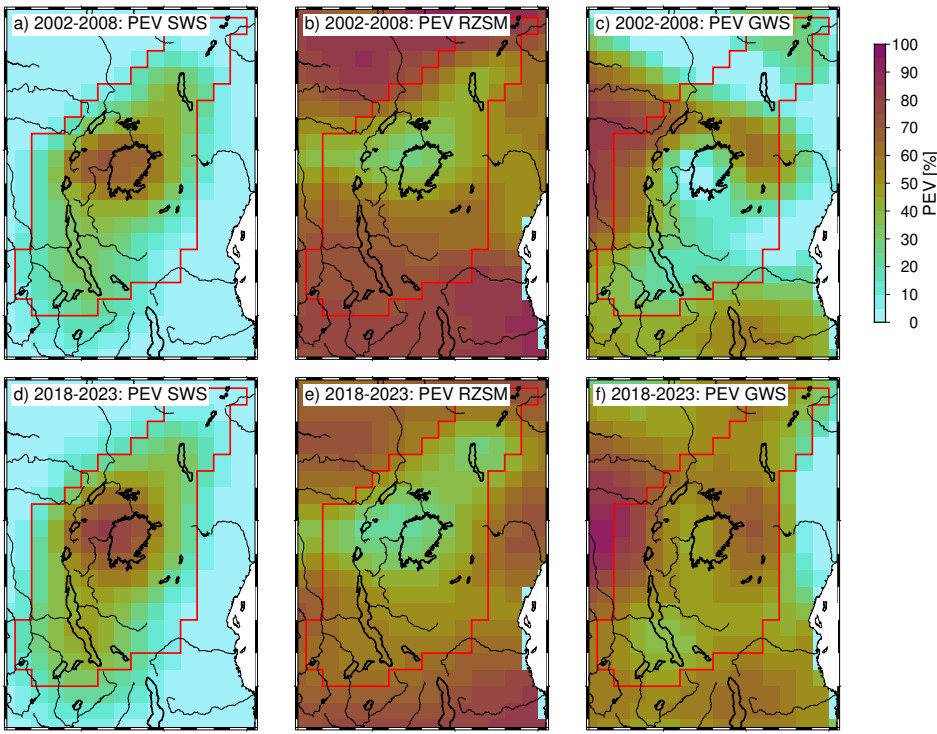

**Figure C1.** $PEV$ for the time period 2002-2008 and 2018-2023

*Author contributions.* EB and AG designed the study concept and discussion of the results with contributions of HD. EB did the implementation and lead of manuscript writing, including figure compilation. MS provided the clustering algorithm, and CS the altimetric water level time series. All authors contributed to the manuscript writing.

*Competing interests.* The authors declare that no competing interests are present.

*Acknowledgements.* We would like to thank Ulrich Meyer, University of Bern, for the computation of the COST-G gravity field product and Christoph Dahle, GFZ German Research Centre for Geosciences, for his contribution in the TWS processing. Qianheng Chen and Peter Morstein contributed to the development of the clustering algorithm. We thank the three reviewers for their very constructive comments, which enabled us to significantly improve this study. Language improvements were made with the help of Grammarly.

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
