# Peer review of "Interannual Variations of Terrestrial Water Storage in the East African Rift Region"

_EGUsphere, 2024_

## Author Comment (AC1)

Dear Vagner,

Thank you very much for your valuable comments. In this first rebuttal letter, we will only address the more significant concerns and changes to the study. Please find them below in red.

Minor text changes, figures, or language-related comments will not all be answered individually here, but we will consider them in the revised version of the manuscript.

With kind regards,

Eva Boergens (on behalf of the authors)

The study "Interannual Variations of Terrestrial Water Storage in the East African Rift Region" addresses an interesting topic and provides some valuable insights. However, several issues need to be addressed before the manuscript can be considered for publication. I recommend major revisions based on the following main points and some other minor comments presented below.

Main points:

1.   The authors state that "human intervention in the form of dam management at Lake Victoria substantially contributes to the TWS variability" (lines 15-16); however, they didn't provide a clear estimation of the magnitude of this contribution. It would be interesting to see the relative contribution of natural variability and human interventions to the observed TWS fluctuations.

Thank you very much for this comment. Assessing the human versus natural contributions from the available observations alone is difficult due to the integrative nature of the gravity observation in particular, which hardly allows for separating different impacts. Instead, hydrological model simulation results could be used to compare simulations with and without human intervention. However, hydrological models explicitly adapted to the study region would be needed for such an investigation, which is unavailable to us. Following your comments and suggestions from the other reviewers, we will focus more strongly on the observed dynamics in the revised version of the manuscript and reduce the focus on their interpretation regarding the anthropogenic influence.

2.   The study proposes a clustering approach to identify the East African Rift region as having similar interannual TWS dynamics. However, the justification for focusing on this specific region could be further improved by providing a stronger rationale for selecting this study area. The manuscript could highlight the East African Rift region's unique hydrological characteristics, ecological significance, or socio-economic importance.

Africa is the only continent with a net positive TWS trend over the last 22 years. We applied the cluster analysis to better identify the patterns of TWS variations at the sub-continental scale. TWS variations in Africa, in general, are not well covered in the GRACE-related literature so far. The observation that the region around Lake Victoria stands out with the strongest positive linear trend within Africa is causing our interest and provoking further investigations. Partly, this study also continues the work of Kvas et al. (2023) (to which some of the authors of this manuscript contributed), where long-term TWS trends with a higher spatial resolution were investigated, featuring the Lake Victoria region as one prominent example. In the study by Kvas et al., we can attribute the long-term trend nearly entirely to the SWS change of the lake.

We will make our reasoning clearer in the revised manuscript and add further evidence for the particular relevance of the study area for both wildlife and humans, e.g., the region's lakes have been named one of the Global 200 eco-regions for conservation by the World Wide Fund for Nature (WWF), emphasising their importance for hydrology and ecosystems (Olson and Dinerstein, 2002). The shores of Lake Victoria are densely populated (one of the world's most populated areas, according to Gridded Population of the World, GPWv4.11). The population heavily relies on the water of lakes for domestic and industrial purposes (Juma et al., 2014).

A Kvas, E Boergens, H Dobslaw, A Eicker, T Mayer-Guerr, A Güntner, Evaluating long-term water storage trends in small catchments and aquifers from a joint inversion of 20 years of GRACE/GRACE-FO mission data, *Geophysical Journal International*, Volume 236, Issue 2, February 2024, Pages 1002–1012, https://doi.org/10.1093/gji/ggad468

Olson, David M., and Eric Dinerstein. "The Global 200: Priority Ecoregions for Global Conservation." Annals of the Missouri Botanical Garden, vol. 89, no. 2, 2002, pp. 199–224. JSTOR, https://doi.org/10.2307/3298564. Accessed 17 June 2024.

Center for International Earth Science Information Network - CIESIN - Columbia University. 2018. Gridded Population of the World, Version 4.11 (GPWv4): Population Count, Revision 11. Palisades, NY: NASA Socioeconomic Data and Applications Center (SEDAC). https://doi.org/10.7927/H4JW8BX5. Accessed 17 06 2024

Juma, D.W., Wang, H. & Li, F. Impacts of population growth and economic development on water quality of a lake: case study of Lake Victoria Kenya water. Environ Sci Pollut Res 21, 5737–5746 (2014). https://doi.org/10.1007/s11356-014-2524-5

3.   Although the study compares TWS variations with precipitation, evapotranspiration, and surface water storage in the major lakes, the analysis of the underlying drivers remains somewhat superficial. The study could provide more information about the potential mechanisms that link these factors to TWS variability in the region (e.g., land use/land cover changes, soil moisture dynamics, groundwater recharge, and human water abstractions). A more comprehensive discussion of these drivers would beef up the interpretations and conclusions of the study.

Thank you for this valuable suggestion, which aligns with the other reviewers' comments. A full analysis of the drivers of TWS variability in the region would go beyond the scope of this paper as it would need to involve comprehensive hydrological modelling, e.g., assessing the effects of land use/land cover changes. Also, in this study, we focus on geodetic observations to analyse the water storage dynamics in the region. This focus will be made more explicit in the revised version of the manuscript. Nevertheless, to investigate the drivers and contributions of the TWS changes in more detail, we performed further analyses of soil moisture and groundwater storage dynamics in relation to TWS variability and will include further results in the revised manuscript.

The authors of this manuscript have been part of the international consortium of the "Global Gravity-based Groundwater Product" (G3P) project funded by the EU as a Horizon 2020 project (https://www.g3p.eu/), joining several leading experts in Europe for satellite-based remote sensing of soil moisture (W. Dorigo, TU Wien), glaciers (M. Zemp, Uni Zurich), snow (K. Luojus, FMI) and mass changes with GRACE (A. Güntner, F. Flechtner, GFZ, T. Mayer-Gürr, TU Graz, A. Jäggi, Uni Bern). G3P provides groundwater storage changes as the difference between TWS and surface water storage (SWS), root zone soil moisture (RZSM), snow, and ice. The latest data set version, including all individual storage compartments, is available until 09/2023 (Güntner et al., 2024. While RZSM is satellite-based, the SWS variations are based on simulation results of the hydrological model LISFLOOD (Van der Knijff et al., 2008). However, LISFLOOD simulations of surface water storage changes are considered unreliable in the study region (cf. Prudhomme et al., 2024). In particular, despite similar dynamics and shorter time scales, the modelled SWS does not show the distinct and strong interannual variability we see in the altimetry-derived SWS of the study (see the following figure).

[Figure]

Thus, we computed groundwater storage variations for the present study based on our altimetry-based SWS results as GWS = TWS - RZSM - SWS(altimetry). Snow and ice can be neglected in the study region.

Side note: Our altimetry-based SWS does not include river storage variations. Based on the model's different SWS components of rivers, lakes and reservoirs, we estimate that river SWS explains roughly 10% of the seasonal SWS variations in the study area and does not show large interannual trends.

The following figure shows the area-average time series of TWS, SWS, RZSM, and GWS for the study area.

[Figure]

These additional data sets will allow us to discuss more on the contributions to the observed TWS signals.

Güntner, Andreas; Sharifi, Ehsan; Haas, Julian; Boergens, Eva; Dahle, Christoph; Dobslaw, Henryk; Dorigo, Wouter; Dussailant, Inés; Flechtner, Frank; Jäggi, Adrian; Kosmale, Miriam; Luojus, Kari; Mayer-Gürr, Torsten; Meyer, Ulrich; Preimesberger, Wolfgang; Ruz Vargas, Claudia; Zemp, Michael (2024): Global Gravity-based Groundwater Product (G3P). V. 1.12. GFZ Data Services. https://doi.org/10.5880/g3p.2024.001

Van Der Knijff, J. M., Younis, J., & De Roo, A. P. J. (2008). LISFLOOD: a GIS-based distributed model for river basin scale water balance and flood simulation. International Journal of Geographical Information Science, 24(2), 189–212. https://doi.org/10.1080/13658810802549154

Prudhomme, Christel, et al. "Global hydrological reanalyses: The value of river discharge information for world-wide downstream applications—The example of the Global Flood Awareness System GloFAS." Meteorological Applications 31.2 (2024): e2192.

4.    The study cites some relevant literature; however, it could improve by discussing how the proposed study's findings compare to or advance previous research on TWS variability in the East African Rift region. The paper would benefit from a more thorough synthesis of the existing knowledge and a clearer articulation of this study's novel contributions.

We will add a more detailed discussion of earlier studies to the introduction and discussion part of the manuscript.

5.    The study lacks a thorough assessment of the uncertainties associated with the GRACE/GRACE-FO data, the precipitation and evapotranspiration datasets, and the surface water storage estimates. It would be interesting to see a more detailed description of the potential sources of error and their implications for the results. Also, the authors could elaborate more on the limitations, such as the coarse spatial resolution of GRACE data and the lack of ground-based validation data. These limitations could be explicitly acknowledged and discussed.

We agree with the reviewer that we should add more uncertainty discussion to the manuscript. The other reviewers have also raised a similar concern. We will add more details on the data uncertainties and discuss their implications.

Unfortunately, we only have reliable uncertainties for the GRACE data set. Boergens et al. (2020, 2022) developed a covariance model for TWS data to assess the uncertainties of this study's used TWS data set. From these, the uncertainties of the STL-derived time series components can be derived via variance propagation.

Although the altimetric water level time series come with an error, these are only formal errors from the Kalman filter estimation. They can only be used for an internal comparison between different time series but not as a measure of uncertainty of the water level observations. In the manuscript, we will discuss the differences between estimates based on time-variable lake area and constant lake area for the uncertainty of the derived surface water storage.

The precipitation and evaporation data sets are provided without uncertainty assessments.

Further, we will add a more thorough discussion of the different data sets' limitations in terms of spatial and temporal resolution.

Boergens, Eva, et al. "Modelling spatial covariances for terrestrial water storage variations verified with synthetic GRACE-FO data." GEM-International Journal on Geomathematics 11.1 (2020): 24.

Boergens, Eva, et al. "Uncertainties of GRACE-Based Terrestrial Water Storage Anomalies for Arbitrary Averaging Regions." Journal of Geophysical Research: Solid Earth 127.2 (2022): e2021JB022081.

6.    The current conclusion section is somewhat vague and does not fully address the broader implications of the findings for water resources management, ecosystem conservation, or climate change adaptation in the region (conditioned to the rationale for selecting the study area as per comment 2). The authors could elaborate on the potential applications of the study's findings.

After the manuscript's revision, considering the reviewers' comments, we will thoroughly revise the conclusion section.

Minor comments:

7.    Between lines 35-40, where it is "Niger Basin in West Africa," it should be "Volta Basin in West Africa" in the context of the sentence.

8.    Lines 134-137: The description of the water occurrence map processing is unclear. Please provide more details on how the 95% occurrence threshold was determined and how it affects the estimation of lake surface areas.

By visually inspecting the lake's water occurrence maps and polygon outlines, we realised that even in the middle of the lake, pixels show a water occurrence of less than 100% but above 95%. See the following figure for Lake Albert. By ignoring this effect, the surface area would be underestimated.

[Figure]

Also, Pekel et al. (2016) documented that their water detection algorithm misses up to 5% of water pixels. That is due to partial cloud cover during the lake remote sensing observations, a particular problem in regions with regular cloud cover (such as the study region). The 95% threshold was found via a histogram of the water occurrences of pixels located clearly inside Lake Victoria (margin of 5km inside the polygon outline). In the manuscript, we will clarify the water surface processing.

Pekel, Jean-François, et al. "High-resolution mapping of global surface water and its long-term changes." Nature 540.7633 (2016): 418-422.

9.    Lines 139-143: Please discuss the limitations of the surface water storage analysis based on a simplified relationship between lake level and area changes based on empirical cumulative distribution functions (ECDF). What could be the potential uncertainties it introduces in the storage estimates? For example, the monotonic and continuous relationship between lake level and area might not always be the case in reality. Lakes with complex bathymetry or irregular shorelines may exhibit non-monotonic or discontinuous relationships between level and area. However, the ECDF approach can handle outliers or anomalies in the input data more robustly than a linear regression used by Ferreira et al. (2018).

Following this comment and comments in a similar direction raised by the other two reviewers, we will add more discussion about the ECDF approach to the manuscript.

The differences in SWS for using time variable surface area or constant area are very small for the large lakes. Non-monotonic relationships between area and water level or relationships with different behaviours for rising or sinking water levels are usually only to be expected around lakes with extensive wetlands. Unfortunately, we cannot further investigate the assumption of a monotonic and continuous relationship between area and water level as we would need time-dependent area data for the lakes. However, the computation of such time series is not feasible as the mosaicking of the remote sensing scenes necessary for such big lakes is very computationally expensive. At the same time, the area change is very small compared to the total lake size.

10.   Lines 240-247: The discussion of the differences between the two SPEI datasets seems speculative. Please provide more evidence to support the claim that the divergence after 2008 is caused by differences in precipitation data rather than PET estimation methods.

We compared the two precipitation data sets in the figure below (monthly time series of area-average precipitation over the study region).

Differences between the two data sets are clearly visible: monthly maxima tend to be higher in GPCC than in CRU, and in some years, the phase of the annual signal is slightly shifted (CRU being later than GPCC).

[Figure]

Further, we investigate the accumulated time series (following figure) with an accumulation period of 48 months (the same period used for SPEI).

Both GPCC and CRU precipitation data sets have a positive trend since 2008 (when the two SPEI time series started to differ), but the increase in precipitation is larger for CRU (GPCC:

5.5mm/year; CRU: 7.3mm/year). This agrees well with the observed increase of SPEI (CRU-based) after 2008 and can be a reason why both SPEI data sets diverge afterwards. We cannot rule out that differences in the PET data used for the two SPEI data sets may also contribute to the SPEI differences. However, the PET data were unavailable for a direct comparison.

[Figure]

We will add this analysis and discussion to the manuscript.

11.    Lines 290-295: The description of the Nalubaale Dam and its impact on Lake Victoria's water levels is incomplete. Please provide more information on the characteristics of the dam (e.g., operating rules) and downstream effects on the Victoria Nile and other water bodies. A study area section presenting the East African Rift Region would be useful.

After the introduction, we will add a section collecting information about the study region, including more on Lake Victoria and the Nalubaale Dam.

12.    Lines 314-315: Please provide a more rigorous assessment of the data quality and its impact on the correlation analysis.

As written above, we will include an uncertainty assessment in the manuscript.

13.    Lines 367-368: That concluding statement seems too broad and not fully supported by the analysis. Please refine this conclusion and provide a more nuanced interpretation of the relative contributions of natural and anthropogenic factors to TWS variability.

We will rewrite the conclusion.

14.   Please revise the English since there are several issues (e.g., Line 5 shows "region region", Line 92 shows "We analyses…")

A thorough English language check will be done.

---

## Author Comment (AC2)

Dear Susanna,

Thank you very much for your valuable comments. This first rebuttal letter will only address the more significant concerns and changes to the study. Minor changes related to the text, figures, or language-related comments will not all be answered individually here. Still, we will consider them all in the revised version of the manuscript.

Please see our answers below in red.

With kind regards,

Eva Boergens (on behalf of the authors)

General Comments

The study presents an analysis of long-term variations in GRACE total water storage variations (TWS) over the last 22 years for Africa and compares the data to surface water storage (SWS) variations in major lakes derived from satellite altimetry in central Africa. The authors compare the datasets also with meteorological/drought data via time series analysis and statistical methods. They discuss the influence of human and climate on the variability in TWS and surface SWS in Central Africa. They also provide some novel insight into the TWS dataset through a cluster analysis for the continent Africa. The authors have conducted a good work. It provides detailed information on data and methods used and provides very interesting insight into the water storage variations in the study area around Lake Victoria in Africa. I do think, however, that a more structured organization of the manuscript; a quantification of uncertainty of the surface water storage estimates; and, following from that, a more comprehensive discussion and concise conclusion of the results would make the work clearer and more significant.

On organization of the work: In Section 3-7, the authors cover certain topics. For Section 3-5 they combine respective methods, results and discussions into one section. For Section 6, some of the relevant methods are explained in Section 2, then some more method description is added after results and discussion in this section. The authors often jump between results presentations along various figures and corresponding piece-wise discussion and conclusions. It makes it harder for the reader to discriminate objective facts from opinion or suggestions by the authors. Some of this becomes especially a problem in Section 5 and even more so Section 7, which are presented the least clear. A clearer structure should be introduced to the entire manuscript, for example, for example, either the methods, or the discussions should be split off in some way. Also, some of the figure organization need some improvements, for example some legends are incomplete. Introducing panel letters might help to address results in figures with more than two panels more clearly. Some figures might be better suited for a supplement. Further suggestions are given below.

Following your advice, we will restructure the manuscript as follows (abbreviated section titles):

1. Introduction
2. Study Region East Africa
3. Data (with the existing subsection but without the methods for surface water storage)
4. Methods:
    a. Time series analysis
    b. Clustering
    c. Surface Water Storage computation
    d. Validation measures
5. Results and Discussion
    a. Clustering
    b. TWS vs Meteo data
    c. TWS vs Surface Storage (new "TWS vs Storage Compartments")
    d. Lake Victoria system
6. Conclusion

The subsections of section 5 will each be subdivided into results and discussion (in the ordering of text, not with subsubsections). Following your suggestions, we will revise the figures and decide which could be moved to the supplement or even left out altogether. We plan to rename section 5 c to "TWS vs. Storage compartments" as we plan to include further storage data sets (see comments below).

On uncertainty of the results: estimates of TWS from GRACE as well as SWS from altimetry and subsequent modeling includes several sources of uncertainty, e.g. measurement errors, parameter uncertainty. These uncertainties should be discussed. But since the authors are quantifying percentage of explained signal variance, a quantification is also suggested, especially for SWS. The conclusions need to be put into perspective of the uncertainties (see also next section).

Thank you very much for this remark, which was raised similarly by the other reviewers. We will add more uncertainty discussions to the manuscript and will discuss their implications.

Unfortunately, we only have reliable uncertainties for the GRACE data set. Boergens et al. (2020, 2022) developed a covariance model for TWS data, which will be used to assess the uncertainties of this study's TWS data set. From these, the uncertainties of the STL-derived time series components can be derived via variance propagation.

Although the provided altimetric water level time series contains the field "error", these errors are only formal errors from the estimation. They can only be used for an internal comparison between different time series but not as a measure of uncertainty of the water level observation. In the manuscript, we will discuss the differences between estimates

based on time-variable lake areas and constant lake areas for the derived surface water storage uncertainty.

The precipitation and evaporation data sets are provided without uncertainty assessments.

Boergens, Eva, et al. "Modelling spatial covariances for terrestrial water storage variations verified with synthetic GRACE-FO data." GEM-International Journal on Geomathematics 11.1 (2020): 24.

Boergens, Eva, et al. "Uncertainties of GRACE-Based Terrestrial Water Storage Anomalies for Arbitrary Averaging Regions." Journal of Geophysical Research: Solid Earth 127.2 (2022): e2021JB022081.

On the conclusions: The study's conclusion on the nature of the driver of TWS variations, i.e. whether it is either human or climate during certain temporal periods, is not fully supported by the results and analysis provided. First, this statement is mainly directly addressed in Section 7, where water levels of various lakes and river level are compared, and the impact of dam management is highlighted. There is no direct comparison with SWS and TWS variations provided. Second, a correlation of TWS to drought indicators is not an explanation or proof of climate dominance, as stated in the conclusion (L357-358), because human water use (e.g. of surface or groundwater) itself is typically also heavily influenced by drought conditions and might therefore similarly impact TWS. In addition, in the rest of the manuscript, the authors only analyze the SWS portion of TWS variation but no soil moisture or groundwater, hence, a large portion of TWS variation remains unexplained, and therefore a conclusion on human or climate dominance in TWS remains very speculative. I am also wondering if such a conclusion is even relevant to emphasize on the importance of the work, but rather may take away from the actual interesting quantitative and qualitative findings of the work on the importance of SWS in the region. This could be more highlighted by slightly altering the discussion of the findings.

Following your comment, as well as similar comments by the other reviewers, we decided to shift the focus of the manuscript away from the anthropogenic influence and towards the (geodetic) observations of the variability of the hydrological storage compartments. We will carefully revise the conclusion accordingly.

In addition, the authors do not comprehensively quantify and discuss why TWS may be rising overall in the Central Africa/Lake Victoria region over the last two decades (they did so only for specific sections of the TWS time series or in relation to P and SWS). It was shown that precipitation plays an important role. However, the P increase (or change in ET) does not indicate if and where the water is stored. (Here, the authors could also make the role of the hydrological processes - flux versus storage - more clear in the work.) Then, is the overall TWS increase mostly due to the accumulation of water in the lakes/reservoirs, or may other storages also play a role? The results the authors show, do suggest that quite some of the increase sources from the lakes. However, since up to 50% of annual variations occur only during very specific times, e.g., dry years (further comments below), and the size of the linear trend is quite different (Figure 10, additional numeric quantification of this overall

increase might be helpful to compare SWS and TWS) a large part of the interannual increase is still unexplained by SWS. However, the correlation between the time series (TWS and SWS in Figure 10, bottom) is striking and the overall rise over the last decades very congruent, just the amplitudes are not matching. So, the question is, does the uncertainty of the SWS amplitudes (from sensors and model parameter) (or from TWS) play a role here? Or are maybe other storage components besides SWS equally important for explaining TWS rise in the region? Just as an example (no need to cite), Werth et al. (2017) have suggested groundwater storage increase may play a role for the storage increase in the Niger basin, and the argument was supported by reports of increasing groundwater levels in the region. Since the cluster for Niger and Lake Victoria have some similarity, maybe groundwater might be relevant in your study area as well. Such or similar thoughts could be included in the discussion and conclusions of the work.

Thank you for this very valid and valuable suggestion. We decided to introduce further analyses of soil moisture and groundwater storage change in the manuscript.

The authors of this manuscript have been part of the international consortium of the "Global Gravity-based Groundwater Product" (G3P) project funded by the EU as a Horizon 2020 project (https://www.g3p.eu/), joining several leading experts in Europe for satellite-based remote sensing of soil moisture (W. Dorigo, TU Wien), glaciers (M. Zemp, Uni Zurich), snow (K. Luojus, FMI) and mass changes with GRACE (A. Güntner, F. Flechtner, GFZ, T. Mayer-Gürr, TU Graz, A. Jäggi, Uni Bern). G3P provides groundwater storage changes as the difference between TWS and surface water storage (SWS), root zone soil moisture (RZSM), snow, and ice. The latest data set version, including all individual storage compartments, is available until 09/2023 (Güntner et al., 2024. While RZSM is satellite-based, the SWS variations are based on simulation results of the hydrological model LISFLOOD (Van der Knijff et al., 2008). However, LISFLOOD simulations of surface water storage changes are considered unreliable in the study region (cf. Prudhomme et al., 2024). In particular, despite similar dynamics and shorter time scales, the modelled SWS does not show the distinct and strong interannual variability we see in the altimetry-derived SWS of the study (see the following figure).

[Figure]

Thus, we computed groundwater storage variations for the present study based on our altimetry-based SWS results as GWS = TWS - RZSM - SWS(altimetry). Snow and ice can be neglected in the study region.

Side note: Our altimetry-based SWS does not include river storage variations. Based on the model's different SWS components of rivers, lakes and reservoirs, we estimate that river SWS explains roughly 10% of the seasonal SWS variations in the study area and does not show large interannual trends.

The following figure shows the area-average time series of TWS, SWS, RZSM, and GWS for the study area.

[Figure]

This data set can assess the yearwise storage change in TWS, SWS, and GWS. From a first look at the time series of the figure, SWS shows more substantial interannual variability, while most of the annual variation originates from RZSM. In further investigations, we will also include uncertainties from both TWS and RZSM.

In addition to the gravity-based groundwater storage estimation, we will further investigate in situ groundwater observations provided by the Global Groundwater Monitoring Network (GGNM, https://ggis.un-igrac.org/view/ggmn/). This network has some groundwater time series available in the study region. However, their time frame is limited.

Güntner, Andreas; Sharifi, Ehsan; Haas, Julian; Boergens, Eva; Dahle, Christoph; Dobslaw, Henryk; Dorigo, Wouter; Dussailant, Inés; Flechtner, Frank; Jäggi, Adrian; Kosmale, Miriam; Luojus, Kari; Mayer-Gürr, Torsten; Meyer, Ulrich; Preimesberger, Wolfgang; Ruz Vargas, Claudia; Zemp, Michael (2024): Global Gravity-based Groundwater Product (G3P). V. 1.12. GFZ Data Services. https://doi.org/10.5880/g3p.2024.001

Van Der Knijff, J. M., Younis, J., & De Roo, A. P. J. (2008). LISFLOOD: a GIS-based distributed model for river basin scale water balance and flood simulation. International Journal of Geographical Information Science, 24(2), 189–212. https://doi.org/10.1080/13658810802549154

Prudhomme, Christel, et al. "Global hydrological reanalyses: The value of river discharge information for world-wide downstream applications–The example of the Global Flood Awareness System GloFAS." Meteorological Applications 31.2 (2024): e2192.

In addition, a few clarifications on the methods and discussions are requested in specific comments further below.

Specific Comments

Abstract: The authors state that the study's main objective "determine whether natural variability or human interventions caused these changes" in TWS variations. However, based on the presented results, the authors can only discuss this for SWS, not for TWS, since they do not analyze other storage components (see comment above).

The abstract will be thoroughly revised after incorporating the changes from the revision.

Introduction: Clarify why were specifically the interannual variations analyzed and not (also) the seasonal variations?

Our interest in the study region was initially triggered by the strong long-term positive trend of TWS as in the findings of Kvas et al. (2024) for the Lake Victoria region. The time series decomposition of STL also allowed us to investigate changes in the amplitude of the seasonal signal. There, we found only minimal changes in the amplitude over time. Thus, we decided to focus on the interannual variations.

A Kvas, E Boergens, H Dobslaw, A Eicker, T Mayer-Guerr, A Güntner, Evaluating long-term water storage trends in small catchments and aquifers from a joint inversion of 20 years of GRACE/GRACE-FO mission data, *Geophysical Journal International*, Volume 236, Issue 2, February 2024, Pages 1002–1012, https://doi.org/10.1093/gji/ggad468

L91: SPEI is typically labeled a drought index. On the data website they define it as follows: "The SPEI is a multiscalar drought index based on climatic data."

L130: Approach to estimate water area bases on optical data. How would the uncertainty of the water occurrence probability due to weather conditions affect the final SWS estimate of the study? Also, this drawback of visual light imagery has been solved by other studies that rely on radar data to detect surface water occurrence, with the advantage that they are not weather-dependent. The authors could include in the discussion, why they have not referred to such data instead, or how application of radar instead of visible light remote sensing images might enhance the accuracy of the method.

We agree that estimating water surface extents from radar imagery is an alternative method, especially during the rainy season in the tropics. However, deriving surface extents from SAR imagery is also challenging.
One limitation is the number of bands available. At least 6 different bands (red, green, blue, near-infrared and 2 shortwave infrared) can be used from optical imagery. SAR imagery

provides only one band but with different polarisations (vertical/horizontal). The number of bands available from optical imagery affects the quality of land-water masks. The estimation of land-water masks from optical imagery is more accurate than that of radar images because the processing of SAR images requires speckle filtering of the speckle noise to reduce the noise. The applied thresholding based on a combination of different water indices (MNDWI, NDWI, AWEI, etc.) derived from different optical bands makes the result more accurate. The applied thresholding used for SAR imagery is based on a single band and is, therefore, not as robust as optical imagery.

It should be noted that the JRC water occurrence mask also uses optical imagery containing data gaps caused by clouds, ice cover, etc. However, the use of several hundred images (even with partial data gaps) since 1984 to estimate the water occurrence mask leads to robust results of the water occurrence mask, even if not all periods can be covered. The increase in temporal resolution over the last few decades has also improved the ability to monitor flood events in our study area.

Finally, we agree that radar imagery could improve the estimation of the land-water mask, especially in the rainy season, but to our knowledge, no datasets equivalent to the JRC dataset based on radar imagery exist. Furthermore, the processing of land-water masks from SAR imagery is beyond the scope of this paper.

Based on the number of comments raised regarding the SWS estimation, we will discuss the limitations of the used approaches and used data sets to the manuscript. See also the further comments and answers below and from the other reviewers.

L137: cululative > culmulative

L145: Add a statement to further spell out what your assumption on the lake profile shape for the volume estimation is, e.g. how steep is the pyramid wall inclined?

The pyramid formula assumes linear lake profiles between the heights of two observations. In most cases, the differences between consecutive height observations are as small as a few centimetres, where this assumption is reasonable. However, due to data gaps or rapid changes in the water level observed only with sparse altimetry coverage, the differences can be as large as one metre. Here, the assumption is indeed questionable. To investigate the effect, we tested the linear interpolation between the two different height values to get closer to the assumption of a linear lake profile between height observations. The following figure shows both the original SWS time series and one computed with interpolated water levels in larger data gaps for Lake Victoria and Lake Bangweulu.

[Figure]

[Figure]

Lake Bangweulu has the most complicated profiles of all lakes in the study area and also the most significant surface extent variations (compared to size). Here, we can observe an effect between interpolating across data gaps or not. But even for this lake, the effect is small. For Lake Victoria, on the other end of the spectrum, no change at all is visible.

We will add a discussion about this to the manuscript.

L30ff/L147: Please clarify, if all lakes in the region were included? Or to what percentage are smaller lakes neglected?

No, only those lakes were considered where surface elevation data from altimetry are available. However, this accounts for 94% of the lake area in the study region (according to the outlines of GLWD). This information will be added to the manuscript.

Equation 2) How representative is such a profile for the lakes? This approach probably has some uncertainty because the lake wall angle is likely heterogeneity inclined, for example, shallower near the shore. Can this introduce a significant error to the total surface water storage estimate? And how large is the uncertainty? It would help to provide a reasonable range of uncertainty for this.

See also the answer above. The method's assumption will introduce no significant error.

All three reviewers have questioned or commented on the ECDF methods for estimating surface water volume so that we will add more discussion and information about this to the manuscript.

L151: I appreciate that the authors spatially filter the surface water data to mimic the sensitivity of the GRACE observations to water mass changes. The author's did not, however, clearly state if the applied gaussian filter width of 350 km is comparable to that applied during the GRACE data processing as conducted for the COST-G dataset. A different filter width can significantly alter the amplitude in storage variations. Since the GRACE dataset used is a unified from various datasets, this might be a bit more complex to evaluate. However, a discussion of it is missing. Optionally, this could be included as another source of uncertainty in the surface water storage time series.

The TWS data set is filtered with the time variable anisotropic VDK filter (Horvath et al. 2018). The different data sets in COST-G are combined on an L2 basis; thus, the combined L2 solution is subsequently filtered as one.

In the meantime, for the G3P groundwater product mentioned above, we investigated which filter width of the Gaussian filter best fits the spatial resolution of GRACE-based TWS filtered with VDK. The related publication of Sharifi et al. is under preparation and we hope to be able to cite it upon publication of this manuscript. According to Sharifi et al., a Gaussian filter width of 250km is best suited for data sets of water storage compartments to make them comparable in spatial resolution to VDK-filtered GRACE-based TWS data. We will thus change the SWS filtering of this study to this value.

We will add more information and discuss the filters used in the manuscript.

Horvath, A., Murböck, M., Pail, R., Horwath, M., 2018. Decorrelation of GRACE time variable gravity field solutions using full covariance information. Geosciences 8, 323. https://doi.org/10.3390/geosciences8090323

151: Please indicate how the filtering was conducted, e.g. in the spatial or frequency (spherical harmonic) domain.

L154: the term "simple" is vague here. I assume you are referring an assumption for stationarity of the temporal components in the time series, as stated further below in L159? Different approaches available (e.g. fourier based, or others) are not more or less simple, but instead they are potentially better applicable to climate processes. Also, the non-stationarity of climate signals is not only present in seasonal components but also in the inter-annual/trend components, hence, why STL is better applicable for both. Please rephrase to make this clearer.

True, the term "simple" here does not fit. We will remove it and add the possible change in the amplitude of the seasonal signal due to climate change.

L160ff: how does the smoothing parameter affect the signal decomposition? What was the criteria for choosing them. I understand this is a trial and error approach, and requires some empirical decision making. However, it would be good to try to write down what you were aiming for, when choosing the parameter.

Please also see our answer to Reviewer #3 on this question. The choice of n_i, n_o, and n_l is straightforward. The most difficult to choose is n_s (smoothing length of the seasonal signal), on which n_t (smoothing length of the trend/interannual signal) depends. We chose n_s such that the seasonal sub-cycle (collection of all values of a given month) no longer contains variations that can be interpreted as noise. N_s=35 was the smallest value where we considered the seasonal sub-cycles noise-free. Our reasoning for choosing the parameter values will be included in the manuscript.

L156/Section 3: Please indicate if the STL is loss-free or not.

L171/Figure 4: If I understand this correctly, the black time series (original in a) is corresponding to the blue long-term signal in b (no-data gap)? I wonder if it makes sense to match the color (same in c and d)?

You are correct! We changed the colours in the figure.

Figure 5&6: The clusters are coded two ways, once by colors and once by numbers. It would be easier to if this is limited to either one. Or also add the colors in the titles, behind numbers in figure 5, e.g. cluster 5 (red) and add numbers to colored dendogram in Figure 6.

We will change the figures following your suggestion.

Figure 6: I was wondering, if it would be sufficient to have this in a supplement. The additional information is minor, as the time series in Figure 5 already show degree of similarity.

L206: I suggest to add brief explanation: regions with overall positive trend are those located in Central Africa (including blue, yellow, dark green, pink).

We will add this to the text.

L207ff: Here, the authors shift from a 7-cluster analysis to an 8-cluster analysis without a more detailed explanation. This should either be a new paragraph, to make that shift more clear.  Alternatively, I am wondering if Figures 5-7 could be combined. For example, why is cluster 8 not also shown in Figure 5?

L207: if I understand it correctly, the sub-clusters in Figure 7 are also appearing in the cluster tree in Figure 6, as the authors emphasis on that here. However, in Figure 6 they are colored all light blue. I was wondering, if it makes sense to mark the purple cluster 8 also in Figure 6, to be more clear.

For both comments above: Our original intention in this part of the manuscript was to discuss the choice of cluster number in more detail and interpret the dendrogram shown in Fig. 6 more. However, after rereading it now, we realised that it is more confusing than helpful for the overall purpose of the manuscript. Thus, we will remove it.

L209: … has even larger TWS amplitudes than … > … has a larger TWS amplitude than ...

L210-211: change the word "marked" to "significant","distinct", or "fast"

L214/Figure 9: The graph in Figure 9 does not look like the values are accumulated, but rather filtered with some kind of moving-window filter of certain width (or accumulated within a moving window). In case of only accumulating, you would have only values every n months, with n being the accumulation period. Please clarify.

Throughout the manuscript, we use accumulation with a moving window, which is standard for the published SPI and SPEI. We will clarify this in the text.

L214-215: You compare accumulated precipitation with SST filtered TWS. The two time series are treated with different methods. Are they really comparable this way? Why do the authors not also apply an SST filter (using the same parameter as for TWS) to the precipitation data instead? This would also save them from estimating the correct filter-width for P.

We tested your very interesting suggestion and present the results for the seasonal and trend/interannual signal in the figure below.

[Figure]

The interannual component of the STL decomposition of precipitation (blue line in the trend plot above) is mainly similar to the accumulated precipitation shown in Fig 9 of the manuscript. TWS has only an annual seasonal signal at the seasonal scale, but precipitation also has a semi-annual seasonal signal (lower plot).

If we were only investigating TWS vs. precipitation, using the STL decomposed precipitation time series would indeed be a good idea for further investigation. However, we also investigated the drought indices SPI and SPEI, where we had to decide on an accumulation period. Thus, we decided to keep the processing of all meteorological data sets (precipitation, SPI, and SPEI) as in the original version of the manuscript, as the same procedure could not be applied to all of them.

Figure 8 might also be ok for a supplement, instead of the main manuscript?

L220-2029: I am wondering if this can be shortened, as P becomes less relevant given their concluding that E is missing to better compared to TWS. However, this conclusion is rather trivial from a hydrological perspective.

L218: Maybe add a sentence explaining the purpose of the violin plot. Does the change in width of the blue areas (violins) have any meaning?

Answers to the three comments above. Following the suggestion of Reviewer #3, we will use the parameters chosen for the TWS STL decomposition as a guiding light for the correct accumulation period. We found that we need at least a 3-year period to estimate the annual and trend components reliably (see also our answer above to your question regarding the parameters). Thus, we should also choose at least an accumulation period of 3 years for the meteorological data sets. Unfortunately, SPI is not published for 36 months. Thus, we decided for the sake of comparability between the accumulated precipitation, SPI, and SPEI to use 48 months. With the investigations shown in the manuscript (so far), we reached a similar conclusion. But as you correctly state, it is relatively trivial from the hydrological perspective and will be significantly shortened in the revised manuscript.

We will change the manuscript accordingly and remove Fig. 8.

L232-233: unclear formulations, please rephrase a bit simpler.

L233-234: unclear formulation, rephrase. "…longterm observation of ?"; also you do not put P-E in relation to TWS, but SPEI

Figure 9: add precipitation to the legend.

L243: do > does

L253: for the names > for their names

L256: I cannot see the 50% in Figure 10, the color bar is kind of vague. The top left Figure 10 colors seem saturated given the color bar. What are the maximum value in Figure 10 top row? It looks to me more like 30%, given the time series in Figure 10 bottom.

Figure 10: The red polygon shown in the upper three panels is neither labeled the legend, nor in the caption. I assume it is outline for cluster 7? Please add.

L261: space missing

Figure 11 caption: correct spelling of de-sesonalized

Figure 11: compares PEV and correlation for de-sesonalized SWS and TWS. It would be useful to show the deseasonalized time series somewhere, e.g. add to Figure 11 or Figure 10 bottom?

We will add the time series to Fig. 11.

L285: the 50% occur only for years with very low TWS, but not for wetter years. Hence, this feels like an overstatement (also in the abstract). Maybe it would be more representative to also estimate the median or mean of the explained percentage over the years? Or it would

be more transparent to discriminate between dry and wet years (see also comment for abstract above)?

We will add similar investigations with RZSM and GWS (as explained above) to the manuscript and, thus, this figure. We will add to the discussion the discrimination between dry and wet years.

L289: Victoria Nile > Nictoria Nile River

L291-295: this information might be better suited already in Section 2.3 to provide more detail on the surface water bodies in the region and how they are managed. It would already help for understanding previous sections.

L311: Can you provide a reference to support this statement?

L235: govern > governed

L235-235: sentence unclear, reformulate

Figure 13: This is not compiled well to support the discussion in Section 7. Maybe presenting the time series in a single or stacked panels and/or in comparison to TWS and/or SWS time series would help the purpose more?

L363: reformulate sentence, a lake cannot lead, rather results for the lake.

L360-362: I disagree, SWS does not fully explain the steady increase of TWS, as shown in Figure 10, only partially. The value of this multi-year TWS/SWS rise was also not quantified in the manuscript, maybe it would help to add this?

L366: The connection between dam discharge and TWS is not clearly shown in the manuscript.

References

Werth, S., White, D., & Bliss, D. W. (2017). GRACE Detected Rise of Groundwater in the Sahelian Niger River Basin. Journal of Geophysical Research: Solid Earth, 122(12), 10,459-10,477. https://doi.org/10.1002/2017JB014845

---

## Author Comment (AC3)

Dear Bramha,

Thank you very much for your valuable comments. In this first rebuttal letter, we will only address the big concerns and changes to the study. Minor text changes or language-related comments will not all be answered individually here, but we will consider them all in the revised version of the manuscript.

Please see our answers below in red.

With kind regards,

Eva Boergens (on behalf of the authors)

Summary: The manuscript uses GRACE(-FO) along with Altimetry, precipitation, and Evaporation datasets to analyse the spatiotemporal behaviour of the East African rift region. In terms of tools, STL and clustering algorithms were used first and then comparisons were made between several variables (lake storage, SPEI, and TWS) to draw conclusions.

General comments: the application of a clustering algorithm to identify regions with similar behaviour is one of the most interesting part of the manuscript, but this is not fully explored. The article has numerous language and grammar errors (from spelling mistakes to redundant and incorrect sentence formations). Authors indicate that they have investigated human vs climate signals, but the analysis in that direction is also weak. They found a good agreement between Altimetry and GRACE & GRACE-FO in general and that remains the most convincing part.

Thank you for this comment. We decided to shift the focus of the manuscript more towards the (geodetic) observations of the water storage compartments in this region and their interpretation, rather than the analysis of human vs climate signals.

Here are some recommendations/concerns/suggestions:

1. Line 5: here the study claims that it will characterize and analyze the interannual TWS variations over the East African rift region to provide a categorical classification: natural or human. Several important hydrological aspects that represent human and climate have been missed in the analysis: for example, groundwater is not accounted for in the whole analysis. The African Monsoon system has a huge impact on the decadal water resource availability in Eastern Africa, which has not been included in the discussion. Even the Monsson system is evolving with climate (see https://www.nature.com/articles/s43017-023-00397-x ). Nevertheless, in the conclusions section, the characterization and analysis is not clearly written: how much of interannual variation can be explained by precipitation (or P-E) and how much of it is due to human decisions on lake outflow. It is appreciated that lake release data is not available, but some quantitative insights based on remote sensing data would add a lot of value and increase the impact of this work on our current state of understanding.

Thank you for this remark. In the revised manuscript we will have more focus on the (geodetic) observations of the water storage compartments in this region and their interpretation rather than the analysis of human versus climate signals. The latter is difficult to disentangle by the integrative observations at hand and may also require a regional hydrological model to simulate the hydrological dynamics with and without human interference which is beyond the scope of this study. Nevertheless, we will additionally use observations of other water storage compartments, namely root zone soil moisture and groundwater, to analyse the contributions to the TWS changes more comprehensively. Also, the results will be discussed in the context of the variability and change of the Monsson system as indicated by the reviewer.

The authors of this manuscript have been part of the international consortium of the "Global Gravity-based Groundwater Product" (G3P) project funded by the EU as a Horizon 2020 project (https://www.g3p.eu/), joining several leading experts in Europe for satellite-based remote sensing of soil moisture (W. Dorigo, TU Wien), glaciers (M. Zemp, Uni Zurich), snow (K. Luojus, FMI) and mass changes with GRACE (A. Güntner, F. Flechtner, GFZ, T. Mayer-Gürr, TU Graz, A. Jäggi, Uni Bern). G3P provides groundwater storage changes as the difference between TWS and surface water storage (SWS), root zone soil moisture (RZSM), snow, and ice. The latest data set version, including all individual storage compartments, is available until 09/2023 (Güntner et al., 2024. While RZSM is satellite-based, the SWS variations are based on simulation results of the hydrological model LISFLOOD (Van der Knijff et al., 2008). However, LISFLOOD simulations of surface water storage changes are considered unreliable in the study region (cf. Prudhomme et al., 2024). In particular, despite similar dynamics and shorter time scales, the modelled SWS does not show the distinct and strong interannual variability we see in the altimetry-derived SWS of the study (see the following figure).

[Figure]

Thus, we computed groundwater storage variations for the present study based on our altimetry-based SWS results as GWS = TWS - RZSM - SWS(altimetry). Snow and ice can be neglected in the study region.

Side note: Our altimetry-based SWS does not include river storage variations. Based on the model's different SWS components of rivers, lakes and reservoirs, we estimate that river SWS explains roughly 10% of the seasonal SWS variations in the study area and does not show large interannual trends.

The following figure shows the area-average time series of TWS, SWS, RZSM, and GWS for the study area.

[Figure]

In the manuscript, we will quantify and discuss the amount of TWS change explained by the different storage compartments and P-E.

Further, we will examine the monsoon's influence on the study region. After revising the manuscript, we will thoroughly revise the abstract and conclusion accordingly.

Güntner, Andreas; Sharifi, Ehsan; Haas, Julian; Boergens, Eva; Dahle, Christoph; Dobslaw, Henryk; Dorigo, Wouter; Dussailant, Inés; Flechtner, Frank; Jäggi, Adrian; Kosmale, Miriam; Luojus, Kari; Mayer-Gürr, Torsten; Meyer, Ulrich; Preimesberger, Wolfgang; Ruz Vargas, Claudia; Zemp, Michael (2024): Global Gravity-based Groundwater Product (G3P). V. 1.12. GFZ Data Services. https://doi.org/10.5880/g3p.2024.001

Van Der Knijff, J. M., Younis, J., & De Roo, A. P. J. (2008). LISFLOOD: a GIS-based distributed model for river basin scale water balance and flood simulation. International Journal of Geographical Information Science, 24(2), 189–212. https://doi.org/10.1080/13658810802549154

Prudhomme, Christel, et al. "Global hydrological reanalyses: The value of river discharge information for world-wide downstream applications–The example of the Global Flood Awareness System GloFAS." Meteorological Applications 31.2 (2024): e2192.

2. Line 8: "separate the TWS signal" -- > "decompose the TWS signal"
3.  Line 10: "study's region" --> study region. This also raises the question if the study region chosen here is the same as East African Rift (EAR)? There are maps of the EAR that differ from the study region obtained via clustering. For example, the Lake Kariba and Lake Malawi (https://www.sciencedirect.com/science/article/pii/S1464343X05001251) are also part of the rift system but outside the study region here. If authors are choosing this name because it is already existing in literature, citing the source would help.

True, we only investigate the northern part of the East African Rift. We chose the name not in view of existing names in the literature but because it appeared to be the best name for the region identified by the clustering in this study. We will make this either clearer in the revised manuscript or consider renaming it.

4. Line 11: The sentence would read better if written as: We observe a decline in TWS un till 2006, followed by a steady increase till 2016, and a sharp increase in 2019 and 2020.
5. Line 13: " large lakes of the region explain large parts" --> "lakes explain large parts"
6. Line 14: "alone contribute up to" --> "alone contributes up to"
7. Line 14: "Satellite altimetry reveals the anthropogenically altered discharge downstream of the dam" : This sentence hurts the coherence of the text. This may be moved to the first or second line in the paragraph.
8. It is already well known that lake water levels and the discharge from the Nile River are Anthropogenic. Authors have cited several papers that also find the same. Hence the last line of the abstract should contain a novel insight from this study.

Will be removed/rephrased

9. Line 21: delete: "cover equally surface and subsurface water storage compartments, i.e., they" (this info is redundant please remove)
10. Line 24: Please rephrase. Either it has to be complementary data or delete "and invaluable complement to all other".
11. Line 25: "tiny" please use a more quantitative adjective such as (micrometer level).
12. Line 25: please rephrase : two twin satellites: language wise it appears that there are 4 of them.
13. Line 25: instead of "trailing each other" it should be "one following the other".
14. Line 26-27: "From collecting these .... derived" --> These intersatellite range measurements over a month are then processed to obtain monthly mean gravity field of the Earth.
15. Line 27: "by computing and comparing .....investigated" --> Changes in the Earth's gravity field are then represented in terms of mass changes near the Earth's surface.
16. Line 35: Rewrite as: Quantifying continental scale terrestrial water storage (TWS) variations has been possible only with GRACE.

17.	Line 46: "These region's lakes ... ecosystems " --> "These lakes have been named in the Global 200 eco ... ecosystems"

18.	Line 52: "monitoring, standardised indices got well established, namely the .. . For example" --> monitoring, well known indices such as SPI and SPEI have been used extensively."

19.	Line 57: "storage variations are by now commonly .. " --> Storage variations are now also monitored.. "

Similar changes are recommended for the rest of the manuscript. A thorough proof reading is essential. I will now only point out spelling mistakes for the manuscript after line 60.

20. Line 72: It is true that the region experienced drought and more water was released. However, then an independent Hydrologic engineer broke this news (https://archive.internationalrivers.org/resources/dams-draining-africa-s-lake-victoria-4117) and the treaty's terms and conditions were enforced which led to a swift recovery. It would be nice to acknowledge that the dam water release was disproportionate and when ensured they were within agreed limits, conditions improved.

Thank you very much for pointing this out. We will acknowledge this in the text.

21. Line 77: The 2018 Rodell paper has termed the TWS increase as "probable natural variability", however, there are studies that also investigate the severity and cause for the trend. In Vishwakarma et al., 2021 (https://iopscience.iop.org/article/10.1088/1748-9326/abd4a9/pdf) the trend observed over the region is found to be "extreme gain" in comparison to the long-term hydrological natural variability. Then by Zhong etal., 2023, these trends have further been attribute to precipitation driven and non-precipitation driven ( https://agupubs.onlinelibrary.wiley.com/doi/full/10.1029/2023WR035817). They found that the trends observed are mostly non-precipitation driven events. This puts the region into "anthropogenic" category, not the "natural variability". Discussion must be added to improve the attribution, or the lack of it.

We will set an extended discussion on anthropogenic versus natural attribution in the revised manuscript also in the context of these previous studies. However, as stated above, we will reduce overall the discussion into anthropogenic vs natural causes.

22. Line 93: Rephrase.

23. Figure 2: this map is not of the full East African Rift region, but a part of it. See the map in (https://www.sciencedirect.com/science/article/pii/S1464343X05001251).

24. Line 139: continues --> continuous.

25. Line 140 – 145: How does this method compares to that given by Wang et al., 2011 for converting lake height to storage?
(https://agupubs.onlinelibrary.wiley.com/doi/10.1029/2011WR010534)

The method used by Wang et al. is based on a parametric description between water level and volume. Thus, it does not allow for more complicated shore profiles. Further, data on both quantities is needed to estimate the power law relationship between water level and volume, which is unavailable in our study region.

Based on your comment and those of the other two reviewers about the surface area estimations, we will revise this part of the manuscript and add more information and discussion.

26. Line 149 –151: Please mention the interpolation technique used

27. Line 158: I am not sure what is meant by "multi-year interannual" Maybe I am wrong, but multi-year variations and interannual variations are synonyms.

28. Line 163: The guidelines for choosing STL parameters are indicative only. For regions such as EAR, where there are two wet seasons and Monsoon system exists (see climatology in Figure 1(d) in https://www.nature.com/articles/s43017-023-00397-x and Figure 5 in https://agupubs.onlinelibrary.wiley.com/doi/full/10.1002/2013WR014350 ), you may also test the seasonal signal to be semiannual. Maybe more investigation is needed to show that these parameters are a good choice.

We tested the parameter. We found the semiannual signal to be less well-fitting to the observed TWS signal than an annual signal. Although precipitation follows a semiannual cycle, TWS does not.

The following investigations will not be fully included in the manuscript but for your information. Nonetheless, we will include more explanations in the text.

We tested for the same example time series one parameter after each other. Thus, all parameters were fixed to the values given in the manuscript except for one variable.

$n_i$: The following figure shows that the inner loop quickly converges. Thus $n_i=1$ is sufficient.

[Figure]

n_o: We cannot assume that TWS data is Gaussian. Thus, n_o has to be larger than 0. For this example, convergence is reached for n_o=5. However, to be safe for all time series, we choose n_o=10.

[Figure]

n_l: Following the reasons given in Cleveland et al. (1990)

n_s: We understood Cleveland et al. (1990) that this parameter should be an integer multiple of the seasonality (here 12) minus 1. Thus, we produced the seasonal diagnostic plot for n_s=11,23,35,47. Here, the seasonal diagnostic plots for n_s=11 and 23 show variations we

interpret as noise rather than signal. With n_s=35, the seasonal signals no longer exhibit such residual noise. Although we expect some changes in the amplitude of the seasonal signal due to climate change, these changes are expected to be slow. Thus, our choice for n_s=35. We will not show the seasonal diagnostic plots due to their large size here.

n_t: Having fixed n_s=35, the choice of n_t would be 19 if following the reasoning given in Cleveland et al. (1990). However, we found that too much noise was still present in the resulting time series with this parameter choice. Thus, after trying out larger values of n_t, we decided to choose n_t=35 (see the figure below).

[Figure]

We will add some of these explanations to the text.

29. Figure 4: The caption appears to be incorrect. Original TWS time-series is red while the data gap is in black. Is blue and green also correctly matched?

The colours were incorrect. Corrected in figure and caption.

30. Line 195: 700 km diameter is good enough to be resolved by GRACE (Vishwakarma et al., 2018: https://www.mdpi.com/2072-4292/10/6/852). Not sure why it was termed "not meaningful" here.

Thank you for this comment. You are entirely correct that a 700km diameter is sufficient for GRACE studies. The reason for not further increasing the number of clusters is that with only one cluster more, ring-shaped clusters will appear (see current Fig. 7). Such shapes will become difficult to interpret. We will correct this in the manuscript.

31. I believe the figure 5 has some interesting time-series. The Cluster analysis divided the region into entities that could also be explained via climatology and human intervention. This

aspect was not explored further for Africa (maybe something in the future), but at least for EAR, there should be some discussion about what makes it unique in terms of climate and why this clustering makes sense.

Exploring the clustering of Africa in more detail is beyond the scope of this study as indicated by the reviewer. Nevertheless, as also commented by reviewer #1, we will add more specific points on the particular features of the study region and how it stands out compared to other regions. We want to point out that the clustering is oriented towards TWS variations, and thus, it is independent of other spatial categorisation schemes such as river basins, precipitation patterns, or others.

32. Section 5: Monthly precipitation and TWS are not directly comparable because of the water budget equation, where P = ET + R + d(S)/dt. To make a fair comparison its recommended that the TWS time-series is differentiated and then compared to the weighted mean Precipitation data (see equation 2 and 3 in Lehmann et al., 2022).

Thank you for this suggestion. We tested the suggested formulas to compare dS/dt with the monthly precipitation. Again, we do not have ET data. The comparison between P and TWSC shows only a weak relationship between these two (correlation = ~35%). This leads to a similar conclusion as drawn with accumulated precipitation. See the following figure:

[Figure]

However, instead of differentiating TWSC from TWS, we can also integrate the other parts of the water budget equation. In the manuscript, we choose to do this by accumulating precipitation and using drought indices (SPI and SPEI) that employ accumulated data. We will add some explanation about this in the manuscript.

Also the concept of accumulated precipitation is not clearly explained. After reading the first paragraph of section 5 three times, I had three interpretations. For example, is the TWS compared with Annual averages of P? Or Is there a moving window of 12 months to compute accumulated P? or the P is accumulated for 12 months and then again for 12 months, which is then added to the last sum of 12 months? It is unclear. Figure 9 has a plot with accumulated P and TWS plotted, which helps me rule out the first and third option, but still not clear.

We applied the same accumulation intervals for P as used for the published SPI and SPEI. Thus, the values for each month are the sum with a moving window. We will clarify this in the text.

33. Figure 9 axis label says precipitation, but the caption says accumulated precipitation. Also, the units of precipitation should be (mm / [time]) and when accumulated (integrated) should become mm. Please revisit this aspect as well.

34. Figure 9: Why is the magnitude of TWS in excess of 4500 mm and there is no negative value. Is TWS also accumulated? The interannual TWS in figure 7 has a range [-200 +400].

Thank you for spotting the mistakes in the figure! We corrected both.

35. Figure 9: Another important observation is that the SPEI (GPCC-based) shows no rise between 2008 to 2019, while there is a rise in TWS as well as SPEI (CRU-based). If there is a lack of trust in precipitation product then how reliable are the conclusions drawn based on them?

The differences between the two SPEI data sets are probably caused by the differences in the precipitation data sets, as we can observe a larger trend since 2008 in the 48 months accumulated CRU observations than in the GPCC observations (GPCC: 5.5mm/year; CRU: 7.3mm/year). This is shown in the figure below:

[Figure]

We will add this analysis and a discussion on the related uncertainties of the precipitation data and the drought analyses to the revised manuscript.

36. Line 228: The decision to choose 48 over 36 needs more thought. The parameters chosen for STL window size could be the guiding light.

Thank you for this suggestion. After revisiting the mentioned part of the manuscript, we decided to shorten the hydrologically trivial (according to reviewer #2) discussion and instead use the STL parameter as a guideline. We found that for both the smoothing of the seasonal and trend components, the minimum length was 35 months. Thus, we also assume that the accumulation period should be at least 3 years, which is also in line with the investigations

currently provided in the manuscript. Unfortunately, SPI is not published for 36 months. Thus, we decided for the sake of comparability between the accumulated precipitation, SPI, and SPEI to use 48 months.

We will revise this part of the manuscript accordingly.

37. Line 250: govern --> governed, despise --> despite

38. Since the signal leakage due to filtering is a problem that will reduce the quality of observations, is it possible to rather use a leakage-correction method for improving GRACE data (such as the forward modelling approach by Chen et al., 2015) instead of filtering the SWS data from altimetry and Lake area?

39. Figure 10: The SWS change slowly while TWS drastically between 2010 and 2017. Precipitation is also not increasing much as seen from Figure 9. The increased lake storage might also interact with the groundwater system. Hence a different rate of change could be attributed to groundwater-recharge (see Figure 5 in https://www.sciencedirect.com/science/article/pii/S0048969721044284 )

As explained above, we plan to introduce groundwater storage change data into the investigation. This will allow us to investigate the yearly storage changes more closely due to the different storage compartments. We will also look into the different behaviour of the storage compartments during dry and wet years.

40. Line 89: In this chapter --> in this section.

41. Section 8, conclusions: Authors claim that there are clear linear trends over Northern India. If one uses STL there is a strong interannual variability over North-west India as well. As we increase the time length, interannual (decadal also) variations start to appear, which is why a longer time-series is needed for climate analysis.

Line 310: courser --> coarser

42. Line 361: this became possible --> this was made possible

 The manuscript is easy to read in parts and requires some effort from the readers in others. A thorough proofreading is required. This is quite an interesting problem and I wish the authors all the best. I hope these comments will be helpful.

Best wishes,

Bramha Dutt Vishwakarma

---

## Author Comment (AC4)

Dear Vagner,

Thank you very much for your valuable comments. We thoroughly revised the manuscript based on them. Please find our response to your comments below in red.

With kind regards,

Eva Boergens (on behalf of the authors)

The study "Interannual Variations of Terrestrial Water Storage in the East African Rift Region" addresses an interesting topic and provides some valuable insights. However, several issues need to be addressed before the manuscript can be considered for publication. I recommend major revisions based on the following main points and some other minor comments presented below.

Main points:

1.    The authors state that "human intervention in the form of dam management at Lake Victoria substantially contributes to the TWS variability" (lines 15-16); however, they didn't provide a clear estimation of the magnitude of this contribution. It would be interesting to see the relative contribution of natural variability and human interventions to the observed TWS fluctuations.

Thank you very much for this comment. Assessing the human versus natural contributions from the available observations alone is difficult due to the integrative nature of gravity observation, which hardly allows for separating different impacts. Instead, hydrological model simulation results could be used to compare simulations with and without human intervention. However, hydrological models explicitly adapted to the study region would be needed for such an investigation, which is unavailable to us. Following your comments and suggestions from the other reviewers, we focus the manuscript more strongly on the observed storage dynamics and reduce the focus on their interpretation regarding the anthropogenic influence.

2.    The study proposes a clustering approach to identify the East African Rift region as having similar interannual TWS dynamics. However, the justification for focusing on this specific region could be further improved by providing a stronger rationale for selecting this study area. The manuscript could highlight the East African Rift region's unique hydrological characteristics, ecological significance, or socio-economic importance.

Africa is the only continent with a net positive TWS trend over the last 22 years. We applied the cluster analysis to better identify the patterns of TWS variations at the sub-continental

scale. The observation that the region around Lake Victoria stands out with the strongest positive linear trend within Africa is causing our interest and provoking further investigations. Partly, this study also continues the work of Kvas et al. (2023) (to which some of the authors of this manuscript contributed), where long-term TWS trends with a higher spatial resolution were investigated, featuring the Lake Victoria region as one prominent example.

We made our reasoning clearer in the revised manuscript and added further evidence for the particular relevance of the study area for both wildlife and humans, e.g., the region's lakes have been named one of the Global 200 eco-regions for conservation by the World Wide Fund for Nature (WWF), emphasising their importance for hydrology and ecosystems (Olson and Dinerstein, 2002). The shores of Lake Victoria are densely populated (one of the world's most populated areas, according to Gridded Population of the World, GPWv4.11). The population heavily relies on the water of lakes for domestic and industrial purposes (Juma et al., 2014).

Following your other comment below, we also added a new section, "Study Region", to collect information about the region in one place.

A Kvas, E Boergens, H Dobslaw, A Eicker, T Mayer-Guerr, A Güntner, Evaluating long-term water storage trends in small catchments and aquifers from a joint inversion of 20 years of GRACE/GRACE-FO mission data, *Geophysical Journal International*, Volume 236, Issue 2, February 2024, Pages 1002–1012, https://doi.org/10.1093/gji/ggad468

Olson, David M., and Eric Dinerstein. "The Global 200: Priority Ecoregions for Global Conservation." Annals of the Missouri Botanical Garden, vol. 89, no. 2, 2002, pp. 199–224. JSTOR, https://doi.org/10.2307/3298564. Accessed 17 June 2024.

Center for International Earth Science Information Network - CIESIN - Columbia University. 2018. Gridded Population of the World, Version 4.11 (GPWv4): Population Count, Revision 11. Palisades, NY: NASA Socioeconomic Data and Applications Center (SEDAC). https://doi.org/10.7927/H4JW8BX5. Accessed 17 06 2024

Juma, D.W., Wang, H. & Li, F. Impacts of population growth and economic development on water quality of a lake: case study of Lake Victoria Kenya water. Environ Sci Pollut Res 21, 5737–5746 (2014). https://doi.org/10.1007/s11356-014-2524-5

3.    Although the study compares TWS variations with precipitation, evapotranspiration, and surface water storage in the major lakes, the analysis of the underlying drivers remains somewhat superficial. The study could provide more information about the potential mechanisms that link these factors to TWS variability in the region (e.g., land use/land cover changes, soil moisture dynamics, groundwater recharge, and human water abstractions). A more comprehensive discussion of these drivers would beef up the interpretations and conclusions of the study.

Thank you for this valuable suggestion, which aligns with the other reviewers' comments. A full analysis of the drivers of TWS variability in the region would go beyond the scope of this

paper as it would need to involve comprehensive hydrological modelling, e.g., assessing the effects of land use/land cover changes. In this study, we focus on geodetic observations to analyse the water storage dynamics in the region. This focus has been made more explicit in the revised version of the manuscript. Nevertheless, to investigate the drivers and contributions of the TWS changes in more detail, we performed further analyses of soil moisture and groundwater storage dynamics in relation to TWS variability and included further results in the revised manuscript (now Section 5.3 Comparison between TWS Signals and Water Storage Compartments).

The authors of this manuscript have been part of the international consortium of the "Global Gravity-based Groundwater Product" (G3P) project funded by the EU as a Horizon 2020 project ([https://www.g3p.eu/](https://www.g3p.eu/)), joining several leading experts in Europe for satellite-based remote sensing of soil moisture (W. Dorigo, TU Wien), glaciers (M. Zemp, Uni Zurich), snow (K. Luojus, FMI) and mass changes with GRACE (A. Güntner, F. Flechtner, GFZ, T. Mayer-Gürr, TU Graz, A. Jäggi, Uni Bern). G3P provides groundwater storage changes as the difference between TWS and surface water storage (SWS), root zone soil moisture (RZSM), snow, and ice. The latest data set version, including all individual storage compartments, is available until 09/2023 (Güntner et al., 2024). While RZSM is satellite-based, the SWS variations are based on simulation results of the hydrological model LISFLOOD (Van der Knijff et al., 2008). However, LISFLOOD simulations of surface water storage changes are considered unreliable in the study region (cf. Prudhomme et al., 2024). In particular, despite similar dynamics and shorter time scales, the modelled SWS does not show the distinct and strong interannual variability we see in the altimetry-derived SWS of the study (see the following figure, not included in the manuscript).

[Figure]

Thus, we computed groundwater storage variations for the present study based on our altimetry-based SWS results as GWS = TWS - RZSM - SWS(altimetry). Snow and ice can be neglected in the study region.

Side note: Our altimetry-based SWS does not include river storage variations. Based on the model's different SWS components of rivers, lakes and reservoirs, we estimate that river SWS explains roughly 10% of the seasonal SWS variations in the study area and does not show large interannual trends.

These additional data sets allowed us to discuss the contributions to the observed TWS signals more thoroughly. The manuscript provides details of the results.

Güntner, Andreas; Sharifi, Ehsan; Haas, Julian; Boergens, Eva; Dahle, Christoph; Dobslaw, Henryk; Dorigo, Wouter; Dussailant, Inés; Flechtner, Frank; Jäggi, Adrian; Kosmale, Miriam; Luojus, Kari; Mayer-Gürr, Torsten; Meyer, Ulrich; Preimesberger, Wolfgang; Ruz Vargas, Claudia; Zemp, Michael (2024): Global Gravity-based Groundwater Product (G3P). V. 1.12. GFZ Data Services. https://doi.org/10.5880/g3p.2024.001

Van Der Knijff, J. M., Younis, J., & De Roo, A. P. J. (2008). LISFLOOD: a GIS-based distributed model for river basin scale water balance and flood simulation. International Journal of Geographical Information Science, 24(2), 189–212. https://doi.org/10.1080/13658810802549154

Prudhomme, Christel, et al. "Global hydrological reanalyses: The value of river discharge information for world-wide downstream applications—The example of the Global Flood Awareness System GloFAS." Meteorological Applications 31.2 (2024): e2192.

4. The study cites some relevant literature; however, it could improve by discussing how the proposed study's findings compare to or advance previous research on TWS variability in the East African Rift region. The paper would benefit from a more thorough synthesis of the existing knowledge and a clearer articulation of this study's novel contributions.

The novelty of this study is the identification and quantification of the strong TWS increase in Eastern Africa and the comprehensive investigation of the different contributions to TWS change (see answer to comment above). Further, we present the first altimetry-based investigation into Lake Victoria and its downstream neighbours to evaluate the Nalubaale dam operation. We clarified this novelty in the introduction and added the comparison to previous studies to the discussions of the results in the respective sections.

5. The study lacks a thorough assessment of the uncertainties associated with the GRACE/GRACE-FO data, the precipitation and evapotranspiration datasets, and the surface water storage estimates. It would be interesting to see a more detailed description of the potential sources of error and their implications for the results. Also, the authors could elaborate more on the limitations, such as the coarse spatial resolution of GRACE data and the lack of ground-based validation data. These limitations could be explicitly acknowledged and discussed.

We agree with this comment, which has been raised similarly by the other reviewers as well. We added more details on the data uncertainties and discussed their implications.

Boergens et al. (2020, 2022) developed a covariance model for TWS data to assess the uncertainties of the used TWS data set. From these, the uncertainties of the STL-derived time series components are derived with the help of a Monte-Carlo simulation.

Although the altimetric water level time series come with an error, these are only formal errors from the Kalman filter estimation. They can only be used for an internal comparison between different time series but not as a measure of uncertainty of the water level observations. Based on literature data, we assume for all altimetric water level time series an uncertainty of 5cm, which is in line with validations against in-situ gauge data. We assume an uncertainty of 5% for the water area extend data. Based on these uncertainties, the uncertainties of the volume time series and SWS are gained through variance propagation.

The newly introduced data set of root zone soil moisture comes with an uncertainty layer, which is used as it is.

For the newly introduced groundwater storage data set, we variance propagate the uncertainties from TWS, SWS, and RZSM.

The precipitation data set is provided without uncertainty assessments.

See more details in the data section of the manuscript.

Further, we added remarks in the manuscript about the unavailability of in-situ observations for water levels and groundwater data and the spatial resolution of TWS data.

Boergens, Eva, et al. "Modelling spatial covariances for terrestrial water storage variations verified with synthetic GRACE-FO data." GEM-International Journal on Geomathematics 11.1 (2020): 24.

Boergens, Eva, et al. "Uncertainties of GRACE-Based Terrestrial Water Storage Anomalies for Arbitrary Averaging Regions." Journal of Geophysical Research: Solid Earth 127.2 (2022): e2021JB022081.

6.    The current conclusion section is somewhat vague and does not fully address the broader implications of the findings for water resources management, ecosystem conservation, or climate change adaptation in the region (conditioned to the rationale for selecting the study area as per comment 2). The authors could elaborate on the potential applications of the study's findings.

Following the large-scale revision of the manuscript, we thoroughly revised the conclusion section. Here we considered all comments the reviewers made regarding the conclusion.

Minor comments:

7.   Between lines 35-40, where it is "Niger Basin in West Africa," it should be "Volta Basin in West Africa" in the context of the sentence.

Changed

8.   Lines 134-137: The description of the water occurrence map processing is unclear. Please provide more details on how the 95% occurrence threshold was determined and how it affects the estimation of lake surface areas.

By visually inspecting the lake's water occurrence maps and polygon outlines, we realised that even in the middle of the lake, pixels show a water occurrence of less than 100% but above 95%. See the following figure for Lake Albert. By ignoring this effect, the surface area would be underestimated.

[Figure]

Also, Pekel et al. (2016) documented that their water detection algorithm misses up to 5% of water pixels. That is due to partial cloud cover during the lake remote sensing observations, a particular problem in regions with regular cloud cover (such as the study region). The 95% threshold was found via a histogram of the water occurrences of pixels located clearly inside Lake Victoria (margin of 20km inside the GLWD polygon outline), see the following figure:

[Figure]

In the manuscript, we clarified the water surface processing and added the two figures above to the appendix.

Pekel, Jean-François, et al. "High-resolution mapping of global surface water and its long-term changes." Nature 540.7633 (2016): 418-422.

9.   Lines 139-143: Please discuss the limitations of the surface water storage analysis based on a simplified relationship between lake level and area changes based on empirical cumulative distribution functions (ECDF). What could be the potential uncertainties it introduces in the storage estimates? For example, the monotonic and continuous relationship between lake level and area might not always be the case in reality. Lakes with complex bathymetry or irregular shorelines may exhibit non-monotonic or discontinuous relationships between level and area. However, the ECDF approach can handle outliers or anomalies in the input data more robustly than a linear regression used by Ferreira et al. (2018).

We found the ECDF approach well-suitable even for complex shorelines, even if it is based on the assumption of a continuous and monotonic relationship. We further assume that the lakes in our study do not show a hysteresis between water level and surface extent. We think this is a reasonable assumption as the lakes are not surrounded by extensive wetlands, which might introduce hysteresis behaviour. Unfortunately, we cannot further investigate

the assumption of a monotonic and continuous relationship between area and water level as we would need time-dependent area data for the lakes.

Overall, we estimate the uncertainty introduced by the ECDF method and its assumptions to be small compared to the uncertainty of altimetric water levels and surface water extent (see comment above).

Following this comment and comments in a similar direction the other two reviewers raised, we added more discussion about the ECDF approach to the manuscript. Further, we included an uncertainty estimation of the SWS data set.

10.    Lines 240-247: The discussion of the differences between the two SPEI datasets seems speculative. Please provide more evidence to support the claim that the divergence after 2008 is caused by differences in precipitation data rather than PET estimation methods.

Thanks for this comment. We accordingly had a more thorough look at the data and revised the presentation in the manuscript: We compared the two precipitation data sets used in the two SPEI data, i.e., GPCC and CRU precipitation, in the figure below (monthly time series of area-average precipitation over the study region).

Differences between the two data sets are clearly visible: monthly maxima tend to be higher in GPCC than in CRU, and in some years, the phase of the annual signal is slightly shifted (CRU being later than GPCC).

[Figure]

Further, we investigate the accumulated time series (following figure) with an accumulation period of 36 months (the same period used for SPEI). The dynamics of the two time series are quite similar, albeit with larger rainfall volumes for GPCC than for CRU.

[Figure]

We cannot rule out that differences in PET used for the two SPEI data sets may also contribute to the SPEI differences. However, the PET data were unavailable for a direct comparison.

We added the figure of the accumulated precipitation to the appendix and added to the text:

*To further investigate the differences between the two variants of SPEI, we looked into their precipitation data used. The comparison of the two precipitation data sets revealed significant differences in the overall volumes of accumulated precipitation but similar interannual dynamics (see Fig B1 in the Appendix). Increasing rainfall trends since 2008 are slightly larger for CRU (4.3 mm/year) than for GPCC (4.0 mm/year) which may partly explain the diverging patterns of the two SPEI data sets after 2008. Nevertheless, differences in the (potential) ET data used for the two SPEI times series may also contribute to the differences, but the (potential) ET data itself were not available to us.*

11.  Lines 290-295: The description of the Nalubaale Dam and its impact on Lake Victoria's water levels is incomplete. Please provide more information on the characteristics of the dam (e.g., operating rules) and downstream effects on the Victoria Nile and other water bodies. A study area section presenting the East African Rift Region would be useful.

After the introduction, we added a section collecting information about the study region, including more on Lake Victoria and the Nalubaale Dam.

12. Lines 314-315: Please provide a more rigorous assessment of the data quality and its impact on the correlation analysis.

As written above, we included an uncertainty assessment for the SWS estimation. However, we assume a constant uncertainty value for the lake altimetry observations (5cm). For rivers, the uncertainty of altimetry observations provided in the literature varies much wider than for lakes. We looked into the formal Kalman filter errors provided with the time series and found these errors around 20 times larger for the river than for the lakes. However, this cannot be transferred one-to-one to the uncertainties of the observations. Further, we cannot use literature values for the uncertainty of the WL of the river, as the spread between different publications and different rivers is much larger than for lakes.

We added to the text:

*Compared to the WL of the lakes, the quality of the time series of the Victoria Nile River is poorer due to the comparatively small size and the challenging topography of the river for satellite altimetry. While literature values for the uncertainty of altimetric WL for lakes are widely available, the uncertainty values for rivers show a significant larger spread. Thus, we do not provide an uncertainty estimate for the WL of the Victoria Nile River.*

13. Lines 367-368: That concluding statement seems too broad and not fully supported by the analysis. Please refine this conclusion and provide a more nuanced interpretation of the relative contributions of natural and anthropogenic factors to TWS variability.

The conclusions have been revised.

14. Please revise the English since there are several issues (e.g., Line 5 shows "region region", Line 92 shows "We analyses…")

A thorough English language check has been performed.

---

## Author Comment (AC5)

Dear Susanna,

Thank you very much for your valuable comments. We thoroughly revised the manuscript based on your suggestions.

Please see our answers below in red.

With kind regards,

Eva Boergens (on behalf of the authors)

General Comments

The study presents an analysis of long-term variations in GRACE total water storage variations (TWS) over the last 22 years for Africa and compares the data to surface water storage (SWS) variations in major lakes derived from satellite altimetry in central Africa. The authors compare the datasets also with meteorological/drought data via time series analysis and statistical methods. They discuss the influence of human and climate on the variability in TWS and surface SWS in Central Africa. They also provide some novel insight into the TWS dataset through a cluster analysis for the continent Africa. The authors have conducted a good work. It provides detailed information on data and methods used and provides very interesting insight into the water storage variations in the study area around Lake Victoria in Africa. I do think, however, that a more structured organization of the manuscript; a quantification of uncertainty of the surface water storage estimates; and, following from that, a more comprehensive discussion and concise conclusion of the results would make the work clearer and more significant.

On organization of the work: In Section 3-7, the authors cover certain topics. For Section 3-5 they combine respective methods, results and discussions into one section. For Section 6, some of the relevant methods are explained in Section 2, then some more method description is added after results and discussion in this section. The authors often jump between results presentations along various figures and corresponding piece-wise discussion and conclusions. It makes it harder for the reader to discriminate objective facts from opinion or suggestions by the authors. Some of this becomes especially a problem in Section 5 and even more so Section 7, which are presented the least clear. A clearer structure should be introduced to the entire manuscript, for example, for example, either the methods, or the discussions should be split off in some way. Also, some of the figure organization need some improvements, for example some legends are incomplete. Introducing panel letters might help to address results in figures with more than two panels more clearly. Some figures might be better suited for a supplement. Further suggestions are given below.

Following your advice, we restructured the manuscript as follows:

1. Introduction

The subsections of section 5 are each be subdivided into results and discussion (in the ordering of text, not with subsubsections). Following your suggestions, we revised the figures and decided which could be moved to the supplement or even omitted altogether. Section 5.3 has been renamed as further storage data sets have been incorporated (see comments below).

On uncertainty of the results: estimates of TWS from GRACE as well as SWS from altimetry and subsequent modeling includes several sources of uncertainty, e.g. measurement errors, parameter uncertainty. These uncertainties should be discussed. But since the authors are quantifying percentage of explained signal variance, a quantification is also suggested, especially for SWS. The conclusions need to be put into perspective of the uncertainties (see also next section).

Thank you very much for this remark, which was raised similarly by the other reviewers. We added uncertainty estimates to all data sets involved as far as possible and included these uncertainties in discussing the results.

Boergens et al. (2020, 2022) developed a covariance model for TWS data to assess the uncertainties of this study's used TWS data set. From these, the uncertainties of the STL-derived time series components are derived with the help of a Monte-Carlo simulation.

Although the altimetric water level time series come with an error, these are only formal errors from the Kalman filter estimation. They can only be used for an internal comparison

between different time series but not as a measure of uncertainty of the water level observations. Based on literature data, we assume for all altimetric water level time series an uncertainty of 5cm, which is in line with validations against in-situ gauge data. We assume an uncertainty of 5% for the water area extend data. Based on these uncertainties, the uncertainties of the volume time series and SWS are gained through variance propagation.

The newly introduced data set of root zone soil moisture comes with an uncertainty layer, which is used as it is.

For the newly introduced data set of groundwater storage, we variance propagate the uncertainties from TWS, SWS, and RZSM.

The precipitation data set is provided without uncertainty assessments.

Boergens, Eva, et al. "Modelling spatial covariances for terrestrial water storage variations verified with synthetic GRACE-FO data." GEM-International Journal on Geomathematics 11.1 (2020): 24.

Boergens, Eva, et al. "Uncertainties of GRACE-Based Terrestrial Water Storage Anomalies for Arbitrary Averaging Regions." Journal of Geophysical Research: Solid Earth 127.2 (2022): e2021JB022081.

On the conclusions: The study's conclusion on the nature of the driver of TWS variations, i.e. whether it is either human or climate during certain temporal periods, is not fully supported by the results and analysis provided. First, this statement is mainly directly addressed in Section 7, where water levels of various lakes and river level are compared, and the impact of dam management is highlighted. There is no direct comparison with SWS and TWS variations provided. Second, a correlation of TWS to drought indicators is not an explanation or proof of climate dominance, as stated in the conclusion (L357-358), because human water use (e.g. of surface or groundwater) itself is typically also heavily influenced by drought conditions and might therefore similarly impact TWS. In addition, in the rest of the manuscript, the authors only analyze the SWS portion of TWS variation but no soil moisture or groundwater, hence, a large portion of TWS variation remains unexplained, and therefore a conclusion on human or climate dominance in TWS remains very speculative. I am also wondering if such a conclusion is even relevant to emphasize on the importance of the work, but rather may take away from the actual interesting quantitative and qualitative findings of the work on the importance of SWS in the region. This could be more highlighted by slightly altering the discussion of the findings.

Following your comment, as well as similar comments by the other reviewers, we decided to shift the focus of the manuscript away from the anthropogenic influence and towards the (geodetic) observations of the variability of the hydrological storage compartments and of TWS. We carefully revised the conclusion accordingly.

In addition, the authors do not comprehensively quantify and discuss why TWS may be rising overall in the Central Africa/Lake Victoria region over the last two decades (they did so only for specific sections of the TWS time series or in relation to P and SWS). It was shown that precipitation plays an important role. However, the P increase (or change in ET) does not indicate if and where the water is stored. (Here, the authors could also make the role of the hydrological processes - flux versus storage - more clear in the work.)

Also, based on a comment by Reviewer #3, we included an explanation about flux vs. storage in Section 5.2 of the manuscript.

Then, is the overall TWS increase mostly due to the accumulation of water in the lakes/reservoirs, or may other storages also play a role? The results the authors show, do suggest that quite some of the increase sources from the lakes. However, since up to 50% of annual variations occur only during very specific times, e.g., dry years (further comments below), and the size of the linear trend is quite different (Figure 10, additional numeric quantification of this overall increase might be helpful to compare SWS and TWS) a large part of the interannual increase is still unexplained by SWS. However, the correlation between the time series (TWS and SWS in Figure 10, bottom) is striking and the overall rise over the last decades very congruent, just the amplitudes are not matching. So, the question is, does the uncertainty of the SWS amplitudes (from sensors and model parameter) (or from TWS) play a role here? Or are maybe other storage components besides SWS equally important for explaining TWS rise in the region? Just as an example (no need to cite), Werth et al. (2017) have suggested groundwater storage increase may play a role for the storage increase in the Niger basin, and the argument was supported by reports of increasing groundwater levels in the region. Since the cluster for Niger and Lake Victoria have some similarity, maybe groundwater might be relevant in your study area as well. Such or similar thoughts could be included in the discussion and conclusions of the work.

Thank you for this very valid and valuable suggestion, which aligns with the other reviewers' comments. We decided to introduce further analyses of soil moisture and groundwater storage change in the manuscript (now Section 5.3 Comparison between TWS Signals and Water Storage Compartments).

The authors of this manuscript have been part of the international consortium of the "Global Gravity-based Groundwater Product" (G3P) project funded by the EU as a Horizon 2020 project (https://www.g3p.eu/), joining several leading experts in Europe for satellite-based remote sensing of soil moisture (W. Dorigo, TU Wien), glaciers (M. Zemp, Uni Zurich), snow (K. Luojus, FMI) and mass changes with GRACE (A. Güntner, F. Flechtner, GFZ, T. Mayer-Gürr, TU Graz, A. Jäggi, Uni Bern). G3P provides groundwater storage changes as the difference between TWS and surface water storage (SWS), root zone soil moisture (RZSM), snow, and ice. The latest data set version, including all individual storage compartments, is available until 09/2023 (Güntner et al., 2024). While RZSM is satellite-based, the SWS variations are

based on simulation results of the hydrological model LISFLOOD (Van der Knijff et al., 2008). However, LISFLOOD simulations of surface water storage changes are considered unreliable in the study region (cf. Prudhomme et al., 2024). In particular, despite similar dynamics and shorter time scales, the modelled SWS does not show the distinct and strong interannual variability we see in the altimetry-derived SWS of the study (see the following figure, not included in the manuscript).

[Figure]

Thus, we computed groundwater storage variations for the present study based on our altimetry-based SWS results as GWS = TWS - RZSM - SWS(altimetry). Snow and ice can be neglected in the study region.

Side note: Our altimetry-based SWS does not include river storage variations. Based on the model's different SWS components of rivers, lakes and reservoirs, we estimate that river SWS explains roughly 10% of the seasonal SWS variations in the study area and does not show large interannual trends.

We now assess the year-wise storage change in TWS, SWS, and GWS. The TWS changes before 2006 are largely explainable with SWS while the changes since 2018 are equally explained by SWS and GWS changes. Most of the annual variation originates from RZSM. We also include uncertainties from all water storage data sets in these investigations. See details of the results in the manuscript.

In addition to the gravity-based groundwater storage estimation, we examined in-situ groundwater observations provided by the Global Groundwater Monitoring Network (GGNM, https://ggis.un-igrac.org/view/ggmn/) to assess how they could be employed in the study. This network has some groundwater time series available in the study region. However, their time frame is very limited. Thus, we decided against introducing them to the study.

Güntner, Andreas; Sharifi, Ehsan; Haas, Julian; Boergens, Eva; Dahle, Christoph; Dobslaw, Henryk; Dorigo, Wouter; Dussaillant, Inés; Flechtner, Frank; Jäggi, Adrian; Kosmale, Miriam; Luojus, Kari; Mayer-Gürr, Torsten; Meyer, Ulrich; Preimesberger, Wolfgang; Ruz Vargas, Claudia; Zemp, Michael (2024): Global Gravity-based Groundwater Product (G3P). V. 1.12. GFZ Data Services. https://doi.org/10.5880/g3p.2024.001

Van Der Knijff, J. M., Younis, J., & De Roo, A. P. J. (2008). LISFLOOD: a GIS-based distributed model for river basin scale water balance and flood simulation. International Journal of Geographical Information Science, 24(2), 189–212. https://doi.org/10.1080/13658810802549154

Prudhomme, Christel, et al. "Global hydrological reanalyses: The value of river discharge information for world-wide downstream applications–The example of the Global Flood Awareness System GloFAS." Meteorological Applications 31.2 (2024): e2192.

In addition, a few clarifications on the methods and discussions are requested in specific comments further below.

Specific Comments

Abstract: The authors state that the study's main objective "determine whether natural variability or human interventions caused these changes" in TWS variations. However, based on the presented results, the authors can only discuss this for SWS, not for TWS, since they do not analyze other storage components (see comment above).

The abstract has been thoroughly revised after incorporating the changes from the revision. The focus is now given as:

*It aims to characterise and analyse the interannual TWS variations compared to meteorological observations and geodetic observations of the water storage compartments surface water, soil moisture, and groundwater.*

Introduction: Clarify why were specifically the interannual variations analyzed and not (also) the seasonal variations?

Our interest in the study region was initially triggered by the strong long-term positive trend of TWS found by Kvas et al. (2023) for the Lake Victoria region. The time series decomposition of STL also allowed us to investigate changes in the amplitude of the seasonal signal. There, we found only minimal changes in the amplitude over time. Thus, we decided to focus on the interannual variations.

We added to the manuscript:

*The TWS variations show a distinct and significant interannual variability but no substantial changes in the seasonal component. Thus, we focus on interannual variability in this study.*

A Kvas, E Boergens, H Dobslaw, A Eicker, T Mayer-Guerr, A Güntner, Evaluating long-term water storage trends in small catchments and aquifers from a joint inversion of 20 years of GRACE/GRACE-

FO mission data, *Geophysical Journal International*, Volume 236, Issue 2, February 2024, Pages 1002–1012, https://doi.org/10.1093/gji/ggad468

L91: SPEI is typically labeled a drought index. On the data website they define it as follows: "The SPEI is a multiscalar drought index based on climatic data."

Changed

L130: Approach to estimate water area bases on optical data. How would the uncertainty of the water occurrence probability due to weather conditions affect the final SWS estimate of the study? Also, this drawback of visual light imagery has been solved by other studies that rely on radar data to detect surface water occurrence, with the advantage that they are not weather-dependent. The authors could include in the discussion, why they have not referred to such data instead, or how application of radar instead of visible light remote sensing images might enhance the accuracy of the method.

We agree that estimating water surface extents from radar imagery is an alternative method, especially during the rainy season in the tropics. However, deriving surface extents from SAR imagery is also challenging.
One limitation is the number of bands available. At least 6 different bands (red, green, blue, near-infrared and 2 shortwave infrared) can be used from optical imagery. SAR imagery provides only one band but with different polarisations (vertical/horizontal). The number of bands available from optical imagery affects the quality of land-water masks. The estimation of land-water masks from optical imagery is more accurate than that of radar images because the processing of SAR images requires speckle filtering of the speckle noise to reduce the noise. The applied thresholding based on a combination of different water indices (MNDWI, NDWI, AWEI, etc.) derived from different optical bands makes the result more accurate. The applied thresholding used for SAR imagery is based on a single band and is, therefore, not as robust as optical imagery.
It should be noted that the JRC water occurrence mask also uses optical imagery containing data gaps caused by clouds, ice cover, etc. However, using several hundred images (even with partial data gaps) since 1984 to estimate the water occurrence mask leads to robust results of the water occurrence mask, even if not all periods can be covered. The increase in temporal resolution over the last few decades has also improved the ability to monitor flood events in our study area.
Finally, we agree that radar imagery could improve the estimation of the land-water mask, especially in the rainy season, but to our knowledge, no datasets equivalent to the JRC dataset based on radar imagery exist. Furthermore, the processing of land-water masks from SAR imagery is beyond the scope of this paper.

We do not think that discussing alternative water area estimates would benefit the manuscript.

L137: cululative > culmulative

Changed

L145: Add a statement to further spell out what your assumption on the lake profile shape for the volume estimation is, e.g. how steep is the pyramid wall inclined?

The pyramid formula assumes linear lake profiles between the heights of two observations. In most cases, the differences between consecutive height observations are as small as a few centimetres, where this assumption is reasonable. However, the differences can be as large as one metre due to data gaps or rapid changes in the water level observed only with sparse altimetry coverage. Here, the assumption is indeed questionable. To investigate the effect, we tested introducing large artificial data gaps in the water level time series and compared the resulting volume time series. The following figure shows both the original volume time series and one computed with the data gaps for Lake Victoria and Lake Bangweulu.

[Figure]

Lake Bangweulu has the most complicated profiles of all lakes in the study area and also the most significant surface extent variations (compared to size). But even for this lake, the effect is very small. For Lake Victoria, on the other end of the spectrum, no change at all is visible.

We added a discussion about this to the manuscript:

*The pyramid formula assumes linear lake profiles between the two WL observations. In most cases, the differences between consecutive height observations are as small as a few centimetres, where this assumption is reasonable. Nevertheless, the differences can be as large as one metre due to data gaps or rapid changes in the water levels observed with temporally sparse altimetry. Thus, we tested the assumption by artificially removing WL observations. We found only very minor differences in the resulting volume time series with*

*and without artificial data gaps. Especially given the uncertainties of the WSA and WL data (see below), these differences are negligible.*

L30ff/L147: Please clarify, if all lakes in the region were included? Or to what percentage are smaller lakes neglected?

No, only those lakes were considered where surface elevation data from altimetry are available. However, this accounts for 94% of the lake area in the study region (according to the outlines of GLWD). This information has been added to the manuscript (Section 2 Study Region):

*All lakes that are accessible with satellite altimetry are included in this study. They account for 94% of the surface water bodies, by area, of the region according to the Global Lake and Wetland Database (GLWD).*

Equation 2) How representative is such a profile for the lakes? This approach probably has some uncertainty because the lake wall angle is likely heterogeneity inclined, for example, shallower near the shore. Can this introduce a significant error to the total surface water storage estimate? And how large is the uncertainty? It would help to provide a reasonable range of uncertainty for this.

See also the answer above. The method's assumption will introduce no significant error.

L151: I appreciate that the authors spatially filter the surface water data to mimic the sensitivity of the GRACE observations to water mass changes. The author's did not, however, clearly state if the applied gaussian filter width of 350 km is comparable to that applied during the GRACE data processing as conducted for the COST-G dataset. A different filter width can significantly alter the amplitude in storage variations. Since the GRACE dataset used is a unified from various datasets, this might be a bit more complex to evaluate. However, a discussion of it is missing. Optionally, this could be included as another source of uncertainty in the surface water storage time series.

The TWS data set is filtered with the time variable anisotropic VDK filter (Horvath et al. 2018). The different data sets in COST-G are combined on an L2 basis; thus, the combined L2 solution is subsequently filtered as one. For clarity, we added both pieces of information to the TWS data section.

In the meantime, for the G3P groundwater product mentioned above, we investigated which filter width of the Gaussian filter best fits the spatial resolution of GRACE-based TWS filtered with VDK. The related publication of Sharifi et al. is being prepared, and we hope to be able to cite it upon publication of this manuscript. So far, we can only provide a technical report

as a reference (Güntner et al., 2023). According to this, a Gaussian filter width of 250km is best suited for data sets of water storage compartments to make them comparable in spatial resolution to VDK-filtered GRACE-based TWS data. We thus changed the SWS filtering to this value.

Of course, using one isotropic filter for all water storage compartments is a simplification. However, we consider it reasonable to allow comparability between the storages.

We added more information about the filter to the manuscript:

*Second, we employ a spatial Gaussian filter with a half-width of 250km to mimic the spatial resolution of TWS. The half-width of 250km has been found by comparing the empirical spatial correlation function of TWS and other water storage compartments filtered with different Gaussian filters (Güntner et al., 2023). That results in the surface water storage (SWS) data set for this study.*

Horvath, A., Murböck, M., Pail, R., Horwath, M., 2018. Decorrelation of GRACE time variable gravity field solutions using full covariance information. Geosciences 8, 323. https://doi.org/10.3390/geosciences8090323

Güntner, A., Sharifi, E., Haas, J., Ruz Vargas, C., and Kidd, R.: Deliverable 4.1 – G3P Product Report – Revision 1, 2023.

151: Please indicate how the filtering was conducted, e.g. in the spatial or frequency (spherical harmonic) domain.

See the answer above.

L154: the term "simple" is vague here. I assume you are referring an assumption for stationarity of the temporal components in the time series, as stated further below in L159? Different approaches available (e.g. fourier based, or others) are not more or less simple, but instead they are potentially better applicable to climate processes. Also, the non-stationarity of climate signals is not only present in seasonal components but also in the inter-annual/trend components, hence, why STL is better applicable for both. Please rephrase to make this clearer.

True, the term "simple" here does not fit. We removed it and added the possible change in the amplitude of the seasonal signal due to climate change:

*A time series decomposition into deterministic periodic (e.g. annual and semiannual) signals and a linear trend is not well suited to characterise the temporal variations of TWS in Africa over the available 21 years. It cannot describe the substantial interannual variability and possible change in the seasonal amplitude due to climate change.*

L160ff: how does the smoothing parameter affect the signal decomposition? What was the criteria for choosing them. I understand this is a trial and error approach, and requires some empirical decision making. However, it would be good to try to write down what you were aiming for, when choosing the parameter.

Please also see our answer to Reviewer #3 on this question. The choice of n_i, n_o, and n_l is straightforward. The most difficult choice is n_s (smoothing length of the seasonal signal), on which n_t (smoothing length of the trend/interannual signal) depends. We chose n_s such that the seasonal sub-cycle (collection of all values of a given month) no longer contains variations that can be interpreted as noise. N_s=35 was the smallest value where we considered the seasonal sub-cycles noise-free. Our reasoning for choosing the parameter values is included in the manuscript:

*The results of the STL decomposition depend on several parameters that govern the smoothness of the interannual and the annual signal. Cleveland et al. (1990) provide guidelines for choosing the parameters, which we used together with empirical testing and visual inspection. That results in the following parameter values. n_p: Length of annual signal, 12 in our case; n_i and n_o: the number of passes through the inner and outer loop, set to 1 and 10, respectively; n_l: the width of the low-pass filter, to be set to the least odd integer larger than n_p, thus set to 13; and n_t and n_s: the trend and seasonal signal smoothing parameter, both set to 35. While the former four parameters are straightforward, n_s requires more considerations. We chose the value n_s=35 in such a way that we consider the interannual variability of the seasonal signal no longer governed by noise. n_t depends on the value of n_s. However, we found that the value provided by the rationale given in Cleveland et al. (1990) (n_t=19) produced a trend component containing still too many short-term variations. Finding the value n_t=35 was done with empirical testing and visual inspection.*

L156/Section 3: Please indicate if the STL is loss-free or not.

STL is loss-free, and we added this to the text.

L171/Figure 4: If I understand this correctly, the black time series (original in a) is corresponding to the blue long-term signal in b (no-data gap)? I wonder if it makes sense to match the color (same in c and d)?

You are correct! We changed the colours in the figure.

Figure 5&6: The clusters are coded two ways, once by colors and once by numbers. It would be easier to if this is limited to either one. Or also add the colors in the titles, behind numbers in figure 5, e.g. cluster 5 (red) and add numbers to colored dendogram in Figure 6.

Following your very good suggestion, we added the colours to the subfigure captions in Fig. 5 for easier recognition. However, we like to keep the numbers as well, as they are very helpful in the text, and not every reader can distinguish colours in figures that well (although we used a colour-blind-friendly colormap). Fig. 6 is removed (see answer below).

Figure 6: I was wondering, if it would be sufficient to have this in a supplement. The additional information is minor, as the time series in Figure 5 already show degree of similarity.

L206: I suggest to add brief explanation: regions with overall positive trend are those located in Central Africa (including blue, yellow, dark green, pink).

L207ff: Here, the authors shift from a 7-cluster analysis to an 8-cluster analysis without a more detailed explanation. This should either be a new paragraph, to make that shift more clear. Alternatively, I am wondering if Figures 5-7 could be combined. For example, why is cluster 8 not also shown in Figure 5?

L207: if I understand it correctly, the sub-clusters in Figure 7 are also appearing in the cluster tree in Figure 6, as the authors emphasis on that here. However, in Figure 6 they are colored all light blue. I was wondering, if it makes sense to mark the purple cluster 8 also in Figure 6, to be more clear.

Answer to the four comments above: We realised that Fig 6 and Fig 7 and the corresponding text were more confusing to the reader than intended. Our original intention in this part of the manuscript was to discuss the choice of cluster number in more detail and interpret the dendrogram shown in Fig. 6 more. Thus, we removed both figures. At the same time, we added some more discussion about the meaning of the clusters and the choice of cluster numbers to the manuscript as follows:

*The numbering of the clusters does not have any further meaning. However, by step-wise increasing the number of regions m the dissimilarity of a found cluster to the other regions can be investigated. If we assume only two clusters (m=2), the algorithm first separates Madagascar Island from mainland Africa due to its spatial disconnection. With m=3, Cluster 0 is already separated from all other regions in mainland Africa, indicating that the TWS signals are most distinct compared to all other TWS signals. The distinction between the TWS signals of the clusters can be measured by the Euclidean distance, on which the algorithm is based, between the mean time series. We found that Cluster 0 has the largest Euclidean distance to all other clusters.*

*The following two regions to be split off are Cluster 5 and Cluster 6, which both have distinct interannual variations, too. As m further increases, the split-off clusters become less distinct and have a larger signal spread within.*

*We decided on the final value for m based on the results, especially the size and shape of the regions. We sought the largest number of clusters while keeping them reasonable for GRACE data interpretation. Here, we found m=8 to be the optimal number.*

*The ninth cluster would be ring-shaped and only about 100km across in the narrowest place. Thus, such a region is no longer meaningfully interpretable with GRACE data.*

L209: … has even larger TWS amplitudes than … > … has a larger TWS amplitude than ...

L210-211: change the word "marked" to "significant","distinct", or "fast"

Both sentences are no longer in the manuscript.

L214/Figure 9: The graph in Figure 9 does not look like the values are accumulated, but rather filtered with some kind of moving-window filter of certain width (or accumulated within a moving window). In case of only accumulating, you would have only values every n months, with n being the accumulation period. Please clarify.

Throughout the manuscript, we use accumulation with a moving window, which is standard for the published SPEI. We clarified this in the text.

L214-215: You compare accumulated precipitation with SST filtered TWS. The two time series are treated with different methods. Are they really comparable this way? Why do the authors not also apply an SST filter (using the same parameter as for TWS) to the precipitation data instead? This would also save them from estimating the correct filter-width for P.

We tested your very interesting suggestion and present the results for the seasonal and trend/interannual signals in the figure below.

[Figure]

The interannual component of the STL decomposition of precipitation (blue line in the trend plot above) is mainly similar to the accumulated precipitation shown in Fig 9 of the manuscript. TWS has only an annual seasonal signal at the seasonal scale, but precipitation also has a semi-annual seasonal signal (lower plot).

If we were only investigating TWS vs. precipitation, using the STL decomposed precipitation time series would indeed be a good idea for further investigation. However, we also investigated the drought index SPEI, where we had to decide on an accumulation period. Thus, we decided to keep processing the meteorological data sets (precipitation and SPEI) as in the original version of the manuscript, as the same procedure could not be applied to all of them.

Figure 8 might also be ok for a supplement, instead of the main manuscript?

L220-2029: I am wondering if this can be shortened, as P becomes less relevant given their concluding that E is missing to better compared to TWS. However, this conclusion is rather trivial from a hydrological perspective.

L218: Maybe add a sentence explaining the purpose of the violin plot. Does the change in width of the blue areas (violins) have any meaning?

Answers to the three comments above. Following the suggestion of Reviewer #3, we used the parameters chosen for the TWS STL decomposition as a guiding light for the correct accumulation period. We found that we need at least a 3-year period to estimate the annual and trend components reliably (see also our answer above to your question regarding the parameters). Thus, we should also choose at least an accumulation period of 3 years for the

meteorological data sets. With the investigations shown in the manuscript (so far), we reached a similar conclusion. But as you correctly state, it is relatively trivial from the hydrological perspective and will be significantly shortened in the revised manuscript. Although we found a higher correlation between TWS and SPEI48, we decided to use 36 months accumulation period in the revised manuscript.

We changed the manuscript accordingly and remove Fig. 8.

L232-233: unclear formulations, please rephrase a bit simpler.

L233-234: unclear formulation, rephrase. "…longterm observation of ?"; also you do not put P-E in relation to TWS, but SPEI

Changed former L232-234 to: *For precipitation minus (potential) evapotranspiration (P-ET), we do not use direct observations but the Standardised Precipitation-Evapotranspiration Index (SPEI) (Vicente-Serrano et al., 2010). This index relates current P-ET observations to the long-term observations since 1955.*

Figure 9: add precipitation to the legend.

Added

L243: do > does

Changed

L253: for the names > for their names

No longer in the manuscript

L256: I cannot see the 50% in Figure 10, the color bar is kind of vague. The top left Figure 10 colors seem saturated given the color bar. What are the maximum value in Figure 10 top row? It looks to me more like 30%, given the time series in Figure 10 bottom.

Figure 10: The red polygon shown in the upper three panels is neither labeled the legend, nor in the caption. I assume it is outline for cluster 7? Please add.

During the large and widespread revision of this section, we decided to remove Fig 10 from the manuscript. With the inclusion of more water storage compartments, namely root zone

soil moisture and groundwater, part of the discussion of SWS was removed. We no longer investigate the spatial pattern of SWS-Lake Victoria separately.

L261: space missing

Removed from manuscript

Figure 11 caption: correct spelling of de-sesonalized

Figure 11: compares PEV and correlation for de-sesonalized SWS and TWS. It would be useful to show the deseasonalized time series somewhere, e.g. add to Figure 11 or Figure 10 bottom?

The caption was incorrect, as we already used the full time series for PEV and correlation before. The figure now also contains PEV and correlation to all WSCs. The new Fig 7 shows the mean time series for all WSCs.

L285: the 50% occur only for years with very low TWS, but not for wetter years. Hence, this feels like an overstatement (also in the abstract). Maybe it would be more representative to also estimate the median or mean of the explained percentage over the years? Or it would be more transparent to discriminate between dry and wet years (see also comment for abstract above)?

We changed the statement to a more moderate one. In median Lake Victoria explains 63% of the SWS change, thus dominating SWS. We did not find a clear relationship between the magnitude of annual SWS change and the explained percentage of Lake Victoria. Further, we also looked into the percentages for GWS and SWS vs TWS and again found no simple relationship (e.g. drying years have more SWS influence, and wetting years have more GWS influence). The discrimination between different years and their drivers is more complicated to discuss, which is now included in the manuscript.

L289: Victoria Nile > Nictoria Nile River

Changed

L291-295: this information might be better suited already in Section 2.3 to provide more detail on the surface water bodies in the region and how they are managed. It would already help for understanding previous sections.

After the introduction, we added a new section, "Study Region", which collects the information in one place.

L311: Can you provide a reference to support this statement?

L235: govern > governed

Changed

L335-335: sentence unclear, reformulate

Changed.

Figure 13: This is not compiled well to support the discussion in Section 7. Maybe presenting the time series in a single or stacked panels and/or in comparison to TWS and/or SWS time series would help the purpose more?

Following your advice, we stacked the time series plots above each other. Plotting them together in one plot (after shifting the time series to a common basis) made reading and understanding the figure even more complicated. We also took care, that the y-axis is in the same resolution, i.e., covering 5m WL for all four plots, to make the amplitude easier to compare.

L363: reformulate sentence, a lake cannot lead, rather results for the lake.

L360-362: I disagree, SWS does not fully explain the steady increase of TWS, as shown in Figure 10, only partially. The value of this multi-year TWS/SWS rise was also not quantified in the manuscript, maybe it would help to add this?

L366: The connection between dam discharge and TWS is not clearly shown in the manuscript.

We thoroughly revised the conclusion, taking the three comments above into account.

References

Werth, S., White, D., & Bliss, D. W. (2017). GRACE Detected Rise of Groundwater in the Sahelian Niger River Basin. Journal of Geophysical Research: Solid Earth, 122(12), 10,459-10,477. https://doi.org/10.1002/2017JB014845

---

## Author Comment (AC6)

Dear Bramha,

Thank you very much for your valuable comments. Your comments and suggestions were considered in revising the manuscript.

Please see our answers below in red.

With kind regards,

Eva Boergens (on behalf of the authors)

Summary: The manuscript uses GRACE(-FO) along with Altimetry, precipitation, and Evaporation datasets to analyse the spatiotemporal behaviour of the East African rift region. In terms of tools, STL and clustering algorithms were used first and then comparisons were made between several variables (lake storage, SPEI, and TWS) to draw conclusions.

General comments: the application of a clustering algorithm to identify regions with similar behaviour is one of the most interesting part of the manuscript, but this is not fully explored. The article has numerous language and grammar errors (from spelling mistakes to redundant and incorrect sentence formations). Authors indicate that they have investigated human vs climate signals, but the analysis in that direction is also weak. They found a good agreement between Altimetry and GRACE & GRACE-FO in general and that remains the most convincing part.

Thank you for this comment. We decided to shift the focus of the manuscript more towards the (geodetic) observations of TWS and of the water storage compartments in this region and their interpretation rather than the analysis of human vs climate signals.

Here are some recommendations/concerns/suggestions:

1. Line 5: here the study claims that it will characterize and analyze the interannual TWS variations over the East African rift region to provide a categorical classification: natural or human. Several important hydrological aspects that represent human and climate have been missed in the analysis: for example, groundwater is not accounted for in the whole analysis. The African Monsoon system has a huge impact on the decadal water resource availability in Eastern Africa, which has not been included in the discussion. Even the Monsson system is evolving with climate (see https://www.nature.com/articles/s43017-023-00397-x ). Nevertheless, in the conclusions section, the characterization and analysis is not clearly written: how much of interannual variation can be explained by precipitation (or P-E) and how much of it is due to human decisions on lake outflow. It is appreciated that lake release data is not available, but some quantitative insights based on remote sensing data would add a lot of value and increase the impact of this work on our current state of understanding.

Thank you for this remark. In the revised manuscript, we focus more on the (geodetic) observations of the water storage compartments in this region and their interpretation rather than the analysis of human versus climate signals. The latter is difficult to disentangle by the integrative observations at hand and may also require a regional hydrological model to simulate the hydrological dynamics with and without human interference, which is beyond the scope of this study. Nevertheless, we used additional observations of other water storage compartments, namely root zone soil moisture and groundwater, to comprehensively analyse the contributions to the TWS changes (Section 5.3 Comparison between TWS Signals and Water Storage Compartments).

The authors of this manuscript have been part of the international consortium of the "Global Gravity-based Groundwater Product" (G3P) project funded by the EU as a Horizon 2020 project (https://www.g3p.eu/), joining several leading experts in Europe for satellite-based remote sensing of soil moisture (W. Dorigo, TU Wien), glaciers (M. Zemp, Uni Zurich), snow (K. Luojus, FMI) and mass changes with GRACE (A. Güntner, F. Flechtner, GFZ, T. Mayer-Gürr, TU Graz, A. Jäggi, Uni Bern). G3P provides groundwater storage changes as the difference between TWS and surface water storage (SWS), root zone soil moisture (RZSM), snow, and ice. The latest data set version, including all individual storage compartments, is available until 09/2023 (Güntner et al., 2024). While RZSM is satellite-based, the SWS variations are based on simulation results of the hydrological model LISFLOOD (Van der Knijff et al., 2008). However, LISFLOOD simulations of surface water storage changes are considered unreliable in the study region (cf. Prudhomme et al., 2024). In particular, despite similar dynamics and shorter time scales, the modelled SWS does not show the distinct and strong interannual variability we see in the altimetry-derived SWS of the study (see the following figure, not included in the manuscript).

[Figure]

Thus, we computed groundwater storage variations for the present study based on our altimetry-based SWS results as GWS = TWS - RZSM - SWS(altimetry). Snow and ice can be neglected in the study region.

Side note: Our altimetry-based SWS does not include river storage variations. Based on the model's different SWS components of rivers, lakes and reservoirs, we estimate that river SWS explains roughly 10% of the seasonal SWS variations in the study area and does not show large interannual trends.

In the manuscript, we now quantify and discuss the amount of TWS change explained by the different storage compartments and P-E.

After revising the manuscript, we thoroughly revised the abstract and conclusion accordingly.

Güntner, Andreas; Sharifi, Ehsan; Haas, Julian; Boergens, Eva; Dahle, Christoph; Dobslaw, Henryk; Dorigo, Wouter; Dussailant, Inés; Flechtner, Frank; Jäggi, Adrian; Kosmale, Miriam; Luojus, Kari; Mayer-Gürr, Torsten; Meyer, Ulrich; Preimesberger, Wolfgang; Ruz Vargas, Claudia; Zemp, Michael (2024): Global Gravity-based Groundwater Product (G3P). V. 1.12. GFZ Data Services. https://doi.org/10.5880/g3p.2024.001

Van Der Knijff, J. M., Younis, J., & De Roo, A. P. J. (2008). LISFLOOD: a GIS-based distributed model for river basin scale water balance and flood simulation. International Journal of Geographical Information Science, 24(2), 189–212. https://doi.org/10.1080/13658810802549154

Prudhomme, Christel, et al. "Global hydrological reanalyses: The value of river discharge information for world-wide downstream applications–The example of the Global Flood Awareness System GloFAS." Meteorological Applications 31.2 (2024): e2192.

Looking into the monsoon system and its temporal changes in East Africa did provide further insights into our study. Thus, we did not elaborate on it.

2. Line 8: "separate the TWS signal" -- > "decompose the TWS signal"

Changed

3. Line 10: "study's region" --> study region. This also raises the question if the study region chosen here is the same as East African Rift (EAR)? There are maps of the EAR that differ from the study region obtained via clustering. For example, the Lake Kariba and Lake Malawi (https://www.sciencedirect.com/science/article/pii/S1464343X05001251) are also part of the rift system but outside the study region here. If authors are choosing this name because it is already existing in literature, citing the source would help.

True, we only investigate the northern part of the East African Rift. We chose the name not because of existing names in the literature but because it appeared to be the best name for the region identified by the clustering in this study. We decided to change the name of the study region to the Northern East African Rift (NEAR) region (but not in the title).

4. Line 11: The sentence would read better if written as: We observe a decline in TWS un
   till 2006, followed by a steady increase till 2016, and a sharp increase in 2019 and 2020.

*Changed*

5. Line 13: " large lakes of the region explain large parts" --> "lakes explain large parts"

*Changed*

6. Line 14: "alone contribute up to" --> "alone contributes up to"

*Changed*

7. Line 14: "Satellite altimetry reveals the anthropogenically altered discharge downstream
   of the dam": This sentence hurts the coherence of the text. This may be moved to the
   first or second line in the paragraph.

*Rephrased*

8. It is already well known that lake water levels and the discharge from the Nile River are
   Anthropogenic. Authors have cited several papers that also find the same. Hence the
   last line of the abstract should contain a novel insight from this study.

*All the cited literature was focused on the drought conditions prior to 2006. Here, we found
evidence in support of these findings. However, the floods of 2019/2020 have not been
investigated in the same way before, and here, we do not find any clear indication of the
anthropogenic influence of the Nalubaale dam on TWS variability. Now, the last sentences of
the abstract are:*

*The Nalubaale Dam regulates Lake Victoria's outflow. Water level observations from satellite
altimetry reveal the impact of dam operation on downstream discharge and on TWS decrease in
the drought years before 2006. On the other hand, we do not find evidence for an impact of the
Nalubaale Dam regulations on the the strong TWS increase after 2019.*

9. Line 21: delete: "cover equally surface and subsurface water storage compartments, i.e.,
   they" (this info is redundant please remove)

*Removed*

10. Line 24: Please rephrase. Either it has to be complementary data or delete "and invaluable complement to all other".

Removed

11. Line 25: "tiny" please use a more quantitative adjective such as (micrometer level).

Changed

12. Line 25: please rephrase : two twin satellites: language wise it appears that there are 4 of them.

Changed

13. Line 25: instead of "trailing each other" it should be "one following the other".

Changed

14. Line 26-27: "From collecting these .... derived" --> These intersatellite range measurements over a month are then processed to obtain monthly mean gravity field of the Earth.

Changed

15. Line 27: "by computing and comparing .....investigated" --> Changes in the Earth's gravity field are then represented in terms of mass changes near the Earth's surface.

Changed

16. Line 35: Rewrite as: Quantifying continental scale terrestrial water storage (TWS) variations has been possible only with GRACE.

Not changed as it would alter the meaning of the sentence.

17. Line 46: "These region's lakes ... ecosystems " --> "These lakes have been named in the Global 200 eco ... ecosystems"

Changed

18. Line 52: "monitoring, standardised indices got well established, namely the .. . For example" --> monitoring, well known indices such as SPI and SPEI have been used extensively."

Changed

19. Line 57: "storage variations are by now commonly .. " --> Storage variations are now also monitored.. "

Changed

Similar changes are recommended for the rest of the manuscript. A thorough proof reading is essential. I will now only point out spelling mistakes for the manuscript after line 60.

20. Line 72: It is true that the region experienced drought and more water was released. However, then an independent Hydrologic engineer broke this news (https://archive.internationalrivers.org/resources/dams-draining-africa-s-lake-victoria-4117) and the treaty's terms and conditions were enforced which led to a swift recovery. It would be nice to acknowledge that the dam water release was disproportionate and when ensured they were within agreed limits, conditions improved.

Thank you very much for pointing this out. We acknowledged this in the text.

21. Line 77: The 2018 Rodell paper has termed the TWS increase as "probable natural variability", however, there are studies that also investigate the severity and cause for the trend. In Vishwakarma et al., 2021 (https://iopscience.iop.org/article/10.1088/1748-9326/abd4a9/pdf) the trend observed over the region is found to be "extreme gain" in comparison to the long-term hydrological natural variability. Then by Zhong etal., 2023, these trends have further been attribute to precipitation driven and non-precipitation driven ( https://agupubs.onlinelibrary.wiley.com/doi/full/10.1029/2023WR035817). They found that the trends observed are mostly non-precipitation driven events. This puts the region into "anthropogenic" category, not the "natural variability". Discussion must be added to improve the attribution, or the lack of it.

Thank you very much for pointing out these two publications. We have added them to the introduction. However, as stated above, we will reduce the focus on discussing anthropogenic vs natural causes in the manuscript.

22. Line 93: Rephrase.

Rephrased.

23. Figure 2: this map is not of the full East African Rift region, but a part of it. See the map in (https://www.sciencedirect.com/science/article/pii/S1464343X05001251).

See our answer above. We changed the name to Northern East African Rift (NEAR).

24. Line 139: continues --> continuous.

Changed

25. Line 140 – 145: How does this method compares to that given by Wang et al., 2011 for converting lake height to storage? (https://agupubs.onlinelibrary.wiley.com/doi/10.1029/2011WR010534)

The method used by Wang et al. is based on a parametric description between water level and volume. Thus, it does not allow for more complicated shore profiles. Further, data on both quantities is needed to estimate the power law relationship between water level and volume, which is unavailable in our study region.

We added to the text:

*Here, we assume a monotonic but non-parametric relationship between WL and WSA. Thus, the ECDF method is more flexible for complicated terrains than methods fitting a parametric curve through the WL -- WSA relationship (e.g. Wang et al, 2011).*

26. Line 149 –151: Please mention the interpolation technique used

Added that we use a linear interpolation.

27. Line 158: I am not sure what is meant by "multi-year interannual" Maybe I am wrong, but multi-year variations and interannual variations are synonyms.

Removed "multi-year"

28. Line 163: The guidelines for choosing STL parameters are indicative only. For regions such as EAR, where there are two wet seasons and Monsoon system exists (see climatology in Figure 1(d) in https://www.nature.com/articles/s43017-023-00397-x and Figure 5 in https://agupubs.onlinelibrary.wiley.com/doi/full/10.1002/2013WR014350 ), you may also

test the seasonal signal to be semiannual. Maybe more investigation is needed to show that these parameters are a good choice.

We tested the parameter. We found the semiannual signal to be less well-fitting to the observed TWS signal than an annual signal. Although precipitation follows a semiannual cycle, TWS does not.

The following investigations will not be fully included in the manuscript but are included for your information, as we know you are an expert using STL. Nonetheless, we included more explanations in the text.

We tested for the same example time series one parameter after each other. Thus, all parameters were fixed to the values given in the manuscript except for one variable.

$n_i$: The following figure shows that the inner loop quickly converges. Thus $n_i=1$ is sufficient.

[Figure]

$n_o$: We cannot assume that TWS data is Gaussian. Thus, $n_o$ has to be larger than 0. For this example, convergence is reached for $n_o=5$. However, to be safe for all time series, we choose $n_o=10$.

[Figure]

n_l: Following the reasons given in Cleveland et al. (1990)

n_s: We understood Cleveland et al. (1990) that this parameter should be an integer multiple of the seasonality (here 12) minus 1. Thus, we produced the seasonal diagnostic plot for n_s=11, 23, 35, 47. Here, the seasonal diagnostic plots for n_s=11 and 23 show variations we interpret as noise rather than signal. With n_s=35, the seasonal signals no longer exhibit such residual noise. Although we expect some changes in the amplitude of the seasonal signal due to climate change, these changes are expected to be slow. Thus, our choice for n_s=35. We will not show the seasonal diagnostic plots due to their large size here.

n_t: Having fixed n_s=35, the choice of n_t would be 19 if following the reasoning given in Cleveland et al. (1990). However, we found that too much noise was still present in the resulting time series with this parameter choice. Thus, after trying out larger values of n_t, we decided to choose n_t=35 (see the figure below).

[Figure]

We added to the text:

*The results of the STL decomposition depend on several parameters that govern the smoothness of the interannual and the annual signal. Cleveland et al. (1990) provide guidelines for choosing the parameters, which we used together with empirical testing and visual inspection. That results in the following parameter values. $n_p$: Length of annual signal, 12 in our case; $n_i$ and $n_o$: the number of passes through the inner and outer loop, set to 1 and 10, respectively; $n_l$: the width of the low-pass filter, to be set to the least odd integer larger than $n_p$, thus set to 13; and $n_t$ and $n_s$: the trend and seasonal signal smoothing parameter, both set to 35. While the former four parameters are straightforward, $n_s$ requires more considerations. We chose the value $n_s=35$ in such a way that we consider the interannual variability of the seasonal signal no longer governed by noise. $n_t$ depends on the value of $n_s$. However, we found that the value provided by the rationale given in Cleveland et al. (1990) ($n_t=19$) produced a trend component containing still too many short-term variations. Finding the value $n_t=35$ was done with empirical testing and visual inspection.*

29. Figure 4: The caption appears to be incorrect. Original TWS time-series is red while the data gap is in black. Is blue and green also correctly matched?

The colours were incorrect. Corrected in figure and caption.

30. Line 195: 700 km diameter is good enough to be resolved by GRACE (Vishwakarma et al., 2018: https://www.mdpi.com/2072-4292/10/6/852). Not sure why it was termed "not meaningful" here.

Thank you for this comment. You are entirely correct that a 700km diameter is sufficient for GRACE studies. The reason for not further increasing the number of clusters is that with only one cluster more, ring-shaped clusters will appear (see current Fig. 7, removed in the revision). Such shapes will become difficult to interpret.

We changed the text to (m is the number of final clusters):

*We decided on the final value for m based on the results, especially the size and shape of the regions. We sought the largest number of clusters while keeping them reasonable for GRACE data interpretation. Here, we found m=8 to be the optimal number.*

*The ninth cluster would be ring-shaped and only about 100km across in the narrowest place. Thus, such a region is no longer meaningfully interpretable with GRACE data.*

31. I believe the figure 5 has some interesting time-series. The Cluster analysis divided the region into entities that could also be explained via climatology and human intervention. This aspect was not explored further for Africa (maybe something in the future), but at least for EAR, there should be some discussion about what makes it unique in terms of climate and why this clustering makes sense.

Exploring Africa's clustering in more detail is beyond the scope of this study, but we invite you to contribute to future work on this!

We want to point out that the clustering is oriented towards TWS variations and thus independent of other spatial categorisation schemes such as river basins, precipitation patterns, or others. As the study aims to investigate the drivers for the observed TWS changes, we do not think it necessary to anticipate discussions about this already in the clustering results section.

32. Section 5: Monthly precipitation and TWS are not directly comparable because of the water budget equation, where P = ET + R + d(S)/dt. To make a fair comparison its recommended that the TWS time-series is differentiated and then compared to the weighted mean Precipitation data (see equation 2 and 3 in Lehmann et al., 2022).

Thank you for this suggestion. We tested the suggested formulas to compare dS/dt with the monthly precipitation. Again, we do not have ET data. The comparison between P and TWSC shows only a weak relationship between these two (correlation = ~35%). See the following figure:

[Figure]

These findings do not improve the interpretations of TWS vs precipitation.

However, instead of differentiating TWSC from TWS, we can also integrate the other parts of the water budget equation. In the manuscript, we choose to do this by accumulating precipitation and using the drought index SPEI that employs accumulated data.

We added to the manuscript at the beginning of now Sec 5.2:

*TWS and precipitation are not directly comparable but linked to each other via the water budget equation: d (TWS)/dt = P - ET - R, with R being the runoff. We decided against differentiating TWS for the comparison but to temporally integrate P and P-ET to evaluate similarities. We call the temporally integrated precipitation ``accumulated precipitation''.*

Also the concept of accumulated precipitation is not clearly explained. After reading the first paragraph of section 5 three times, I had three interpretations. For example, is the TWS compared with Annual averages of P? Or Is there a moving window of 12 months to compute accumulated P? or the P is accumulated for 12 months and then again for 12 months, which is then added to the last sum of 12 months? It is unclear. Figure 9 has a plot with accumulated P and TWS plotted, which helps me rule out the first and third option, but still not clear.

We applied the same accumulation intervals for P as used for the published SPEI. Thus, the values for each month are the sum with a moving window.

We clarified the text:

*Instead of monthly precipitation, we consider time series of accumulated precipitation. To this end, the accumulated precipitation value for each month is the sum over the preceding n months, with n taking integer values between 1 and 48.*

33. Figure 9 axis label says precipitation, but the caption says accumulated precipitation. Also, the units of precipitation should be (mm / [time]) and when accumulated (integrated) should become mm. Please revisit this aspect as well.

34. Figure 9: Why is the magnitude of TWS in excess of 4500 mm and there is no negative value. Is TWS also accumulated? The interannual TWS in figure 7 has a range [-200 +400].

Thank you for spotting the mistakes in the figure! We corrected both.

35. Figure 9: Another important observation is that the SPEI (GPCC-based) shows no rise between 2008 to 2019, while there is a rise in TWS as well as SPEI (CRU-based). If there is a lack of trust in precipitation product then how reliable are the conclusions drawn based on them?

We investigated the accumulated precipitation time series for GPCC and CRU (following figure) with an accumulation period of 36 months (the same period used for SPEI).

[Figure]

The amplitude and the trends since 2008 differ between the two data sets.

We cannot rule out that differences in PET used for the two SPEI data sets may also contribute to the SPEI differences. However, the PET data were unavailable for a direct comparison.

We added the figure of the accumulated precipitation to the appendix and added to the text:

*The comparison of the two precipitation data sets revealed significant differences in the overall volumes of accumulated precipitation but similar interannual dynamics (see B1 in the Appendix). Increasing rainfall trends since 2008 are slightly larger for CRU (4.3 mm/year) than for GPCC (4.0 mm/year) which may partly explain the diverging patterns of the two SPEI data sets after 2008.  Nevertheless, differences in the (potential) ET data used for the two SPEI times series may also contribute to the differences, but the (potential) ET data itself were not available to us.*

Further, we added at the end of the section:

*Further, the discussed differences between the two SPEI data sets and their input precipitation data sets limit the explanatory power of the comparison between SPEI and TWS. Hence, we will investigate the different storage compartments of TWS in the next section to identify further drivers of TWS variability.*

36. Line 228: The decision to choose 48 over 36 needs more thought. The parameters chosen for STL window size could be the guiding light.

Thank you for this suggestion. After revisiting the mentioned part of the manuscript, we decided to shorten the hydrologically trivial (according to reviewer #2) discussion and instead use the STL parameter as a guideline. We found that for both the smoothing of the seasonal and trend components, the minimum length was 35 months. Thus, we also assume that the accumulation period should be at least 3 years, which is also in line with the investigations currently provided in the manuscript. We decided to change the accumulation period of precipitation and SPEI to 36 in the study and revised the part of the manuscript accordingly.

37. Line 250: govern --> governed, despise --> despite

Changed

38. Since the signal leakage due to filtering is a problem that will reduce the quality of observations, is it possible to rather use a leakage-correction method for improving GRACE data (such as the forward modelling approach by Chen et al., 2015) instead of filtering the SWS data from altimetry and Lake area?

We decided against using the leakage correction provided together with the used GravIS TWS data, as this correction is only suitable for regional means. Although large parts of the study are done on the regional mean time series, we also investigate spatial patterns. We think it better to have a common data basis for all parts of the study and not switch between leakage-corrected time series and uncorrected spatial patterns.

For applying another leakage approximation based on forward modelling (like Chen et al., 2015) or hydrological models, we decided against this as we do not trust the hydrological models sufficiently in the region (see answer above about LISFLOOD) as input to this method.

Additionally, we found only small differences between the region's leakage-corrected and uncorrected time series. Thus, we felt that leakage is not a big issue for this study.

39. Figure 10: The SWS change slowly while TWS drastically between 2010 and 2017. Precipitation is also not increasing much as seen from Figure 9. The increased lake storage

might also interact with the groundwater system. Hence a different rate of change could be attributed to groundwater-recharge (see Figure 5 in https://www.sciencedirect.com/science/article/pii/S0048969721044284 )

As explained above, we introduced groundwater storage change data into the investigation. This allowed us to investigate the yearly storage changes more closely due to the different storage compartments. We also looked into the different behaviour of the storage compartments during dry and wet years but found no clear correlation.

40. Line 89: In this chapter --> in this section.

Changed

41. Section 8, conclusions: Authors claim that there are clear linear trends over Northern India. If one uses STL there is a strong interannual variability over North-west India as well. As we increase the time length, interannual (decadal also) variations start to appear, which is why a longer time-series is needed for climate analysis.

Line 310: courser --> coarser

42. Line 361: this became possible --> this was made possible

Three comments above were taken into account during the revision of the conclusion.

The manuscript is easy to read in parts and requires some effort from the readers in others. A thorough proofreading is required. This is quite an interesting problem and I wish the authors all the best. I hope these comments will be helpful.

Best wishes,

Bramha Dutt Vishwakarma